# Shared and unique brain network features predict cognitive, personality, and mental health scores in the ABCD study

Jianzhong Chen[1,2,3,4,20], Angela Tam [1,2,3,4,20], Valeria Kebets[1,2,3,4], Csaba Orban [1,2,3,4],
Leon Qi Rong Ooi [1,2,3,4,5], Christopher L. Asplund [2,3,4,6,7,8], Scott Marek[9], Nico U. F. Dosenbach [10,11,12,13],
Simon B. Eickhoff[14,15], Danilo Bzdok [16,17], Avram J. Holmes [18] & B. T. Thomas Yeo [1,2,3,4,5,19 ✉]

How individual differences in brain network organization track behavioral variability is a fundamental question in systems neuroscience. Recent work suggests that resting-state and task-state functional connectivity can predict specific traits at the individual level. However, most studies focus on single behavioral traits, thus not capturing broader relationships across behaviors. In a large sample of 1858 typically developing children from the Adolescent Brain Cognitive Development (ABCD) study, we show that predictive network features are distinct across the domains of cognitive performance, personality scores and mental health assessments. On the other hand, traits within each behavioral domain are predicted by similar network features. Predictive network features and models generalize to other behavioral measures within the same behavioral domain. Although tasks are known to modulate the functional connectome, predictive network features are similar between resting and task states. Overall, our findings reveal shared brain network features that account for individual variation within broad domains of behavior in childhood.

[1] Department of Electrical and Computer Engineering, National University of Singapore, Singapore, Singapore. [2] Centre for Sleep and Cognition, National University of Singapore, Singapore, Singapore. [3] Centre for Translational MR Research, National University of Singapore, Singapore, Singapore. [4] N.1 Institute for Health & Institute for Digital Medicine (WisDM), National University of Singapore, Singapore, Singapore. [5] Integrative Sciences and Engineering Programme (ISEP), National University of Singapore, Singapore, Singapore. [6] Division of Social Sciences, Yale-NUS College, Singapore, Singapore. [7] Department of Psychology, National University of Singapore, Singapore, Singapore. [8] Duke-NUS Medical School, Singapore, Singapore. [9] Department of Psychiatry, Washington University School of Medicine, St. Louis, MO, USA. [10] Department of Neurology, Washington University School of Medicine, St. Louis, MO, USA. [11] Department of Radiology, Washington University School of Medicine, St. Louis, MO, USA. [12] Department of Biomedical Engineering, Washington University School of Medicine, St. Louis, MO, USA. [13] Department of Pediatrics, Washington University School of Medicine, St. Louis, MO, USA. [14] Institute for Systems Neuroscience, Medical Faculty, Heinrich-Heine University Düsseldorf, Düsseldorf, Germany. [15] Institute of Neuroscience and Medicine, Brain & Behaviours (INM-7), Research Center Jülich, Jülich, Germany. [16] Department of Biomedical Engineering, Montreal Neurological Institute, McGill University, Montreal, QC, Canada. [17] Mila - Quebec AI Institute, Montreal, QC, Canada. [18] Yale University, Departments of Psychology and Psychiatry, New Haven, CT, USA. [19] Martinos Center for Biomedical Imaging, Massachusetts General Hospital, Charlestown, MA, USA. [20] These authors contributed equally: Jianzhong Chen, Angela Tam. ✉email: thomas.yeo@nus.edu.sg

A central question in systems neuroscience is how brain-network architecture supports a wide repertoire of human behavior across the lifespan. Childhood is a period of rapid neural development and behavioral changes across cognition, personality, and mental health[1–3]. Consequently, there is particular interest in understanding the nature of brain-behavior relationships instantiated early in the lifespan[4,5]. Here, we utilized a large-scale dataset of typically developing 9- to 10-year-old children[6] to quantitatively characterize functional network organization that supports individual-level prediction of cognitive performance, impulsivity-related personality scores, and mental health assessments across resting and task states.

Whole-brain connectome-wide neurodevelopmental studies have found associations between resting-state functional network organization and behavioral traits[7–10]. However, clinical decisions are made at the individual level[11,12]. As such, there is an increasing shift from associational analyses to individual-level prediction[13–17]. Using machine-learning algorithms, we can exploit interindividual heterogeneity in functional connectomes to make predictions about a single person's behavior[14]. Consequently, neurodevelopmental prediction studies have used resting-state functional connectivity (resting FC) to predict individual differences in cognitive performance[18,19], impulsivity scores[20] and autism symptoms[21,22].

Recent studies have further suggested that task-state functional connectivity (task FC) yields better prediction of cognitive performance over resting FC[23–25], with additional improvements from combining task FC and resting FC[26,27]. Therefore, one might hypothesize that the functional connections predictive of individual-level cognition (i.e., predictive network features) might differ between rest and task states. However, other studies have shown that the brain functional network architecture is broadly similar during rest and task[28–30]. While task contexts reliably modulate functional network organization[31–33], task modulation of the functional connectome within individuals is much smaller than differences between individuals[34]. Therefore, an alternative hypothesis is that predictive-network features are similar across brain states. We seek to investigate the two competing hypotheses in this study.

Furthermore, most previous connectome-based prediction studies have focused on specific behavioral traits[17,19,21,23–25,35]. Yet, the human brain has evolved to execute a diverse range of behaviors, so focusing on single behavioral traits might miss the forest for the trees[36]. More specifically, it remains unclear whether predictive-network features are similar or different across behavioral measures. For example, specialized brain networks support distinct cognitive processes, such as attention, language, or episodic memory[37–39]. Thus, one might hypothesize that distinct network features support prediction of different cognitive traits. Conversely, many studies have emphasized information integration across specialized brain networks[40–42]. FC studies of cognition[14,19,23] and mental disorders[43–46] have also suggested the importance of default, control, and salience/ventral attention networks. Therefore, an alternate hypothesis is that a common set of predictive-network features might explain individual differences in cognition, or even across cognition and mental health. To systematically investigate the two hypotheses, we considered the prediction of a variety of behavioral measures. This population neuroscience approach allowed us to estimate the degree of overlap in predictive-network features across different behavioral domains (cognitive performance, personality scores, and mental health assessments), as well as across phenotypes within the same behavioral domain.

In the present study, we utilized the Adolescent Brain Cognitive Development (ABCD) study, a unique dataset with a large sample of children and a diverse set of behavioral measures[6]. We used resting FC and task FC to predict a wide range of cognitive, impulsivity-related personality, and mental health measures. We also investigated whether combining resting FC and task FC can improve behavioral prediction. Most importantly, we explored the existence of shared and unique predictive-network features within and across behavioral domains, as well as across brain (resting and task) states.

## Results

We used resting fMRI and task fMRI from 11875 children (ABCD 2.0.1 release). There were three tasks: monetary-incentive delay (MID), stop-signal task (SST), and N-back. We also considered all available dimensional neurocognitive[47] and mental health[48] assessments, yielding 16 cognitive, 9 (impulsivity-related) personality, and 11 mental health measures (Supplementary Tables 1 and 2). Mental health measures included assessments from Achenbach Child Behavior Check List[49], Parent General Behavior Inventory[50], and Pediatric Psychosis Questionnaire—Brief Version[51]. After quality control (QC) and considering only participants with complete resting fMRI, task fMRI, and behavioral data, our main analyses utilized data from 1858 unrelated children (Fig. 1A).

**Task FC outperforms resting FC for predicting cognition, but not personality or mental health.** We computed FC (Pearson's correlations) among the average time courses of 400 cortical[52] and 19 subcortical[53] regions (Fig. 1B, C), yielding a $419 \times 419$ FC matrix for each brain state (rest, MID, SST, and N-back). We used kernel regression to predict each behavioral measure based on resting FC, MID-FC, SST-FC, and N-back FC separately. We have previously demonstrated that kernel regression is a powerful approach for resting-FC behavioral prediction[54]. The idea behind kernel regression is that participants with more similar FC matrices would exhibit more similar behavior.

To evaluate the kernel regression performance, we utilized an inner-loop (nested) cross-validation procedure in which participants were repeatedly divided into training and test sets. The regression model was fitted on the training set and used to predict behavior in the test set. Care was taken so that participants from the same site were not split between training and test sets. This cross-validation procedure was repeated 120 times to ensure stability[55]. See "Methods" for more details.

Figure 2A shows the prediction performance averaged within each behavioral domain. Each behavioral domain was predicted better than chance (false-discovery rate FDR $q < 0.05$) with $p < 0.001$ across all brain states for cognition, (impulsivity-related) personality, and mental health, respectively.

Consistent with previous studies[23], we found that MID-FC and N-back FC outperformed resting FC ($p < 0.001$) in predicting cognition. SST-FC had worse performance than resting FC ($p = 0.008$), but we note that resting FC had about 50% more timepoints than SST-FC, which could explain the difference. Interestingly, N-back FC performed the best with the least amount of timepoints. In the cases of personality and mental health, there was no statistical difference between resting FC and any task state. Thus, task FC appeared to improve prediction performance for cognition, but not personality or mental health.

**Combining task FC and resting FC improves prediction, particularly for cognition.** Previous studies have suggested that combining task FC and resting FC can improve prediction of fluid intelligence test performance[26,27] and reading comprehension[24]. We extended the previous studies by performing multikernel ridge regression using resting FC, MID-FC, SST-FC, and N-back FC

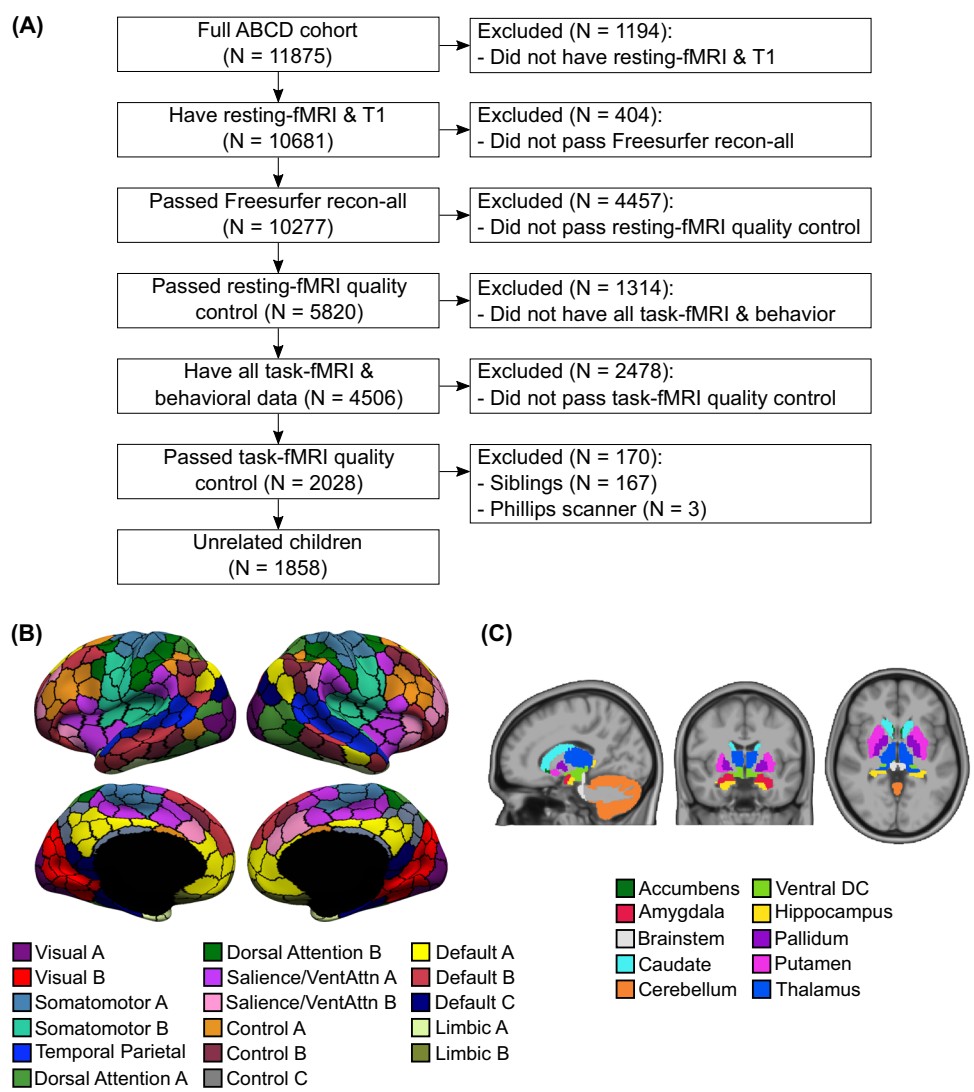

**Fig. 1 Overview of preprocessing workflow. A** Flowchart illustrating inclusion/exclusion criteria. **B** Cortical parcellation of 400 regions[52]. Parcel colors are assigned according to 17 large-scale networks[152]. Image reproduced under a CC BY 4.0 license, credit: https://doi.org/10.6084/m9.figshare.10062482.v1 (**C**) Nineteen subcortical regions[53]. Image reproduced under a CC BY 4.0 license, credit: https://doi.org/10.6084/m9.figshare.10063016.v1.

jointly to predict a broader range of cognitive measures as well as noncognitive (personality and mental health) measures.

Figure 2 shows the multikernel prediction performance averaged within each behavioral domain. Since N-back performed the best among the single-kernel regression for all behavioral domains (Fig. 2A), we compared multikernel FC with N-back FC (Fig. 2B). We found that multikernel FC performed better than N-back FC for cognitive ($p < 0.001$) and personality ($p = 0.022$), but not mental health ($p = 0.124$).

Figure 3 shows the prediction performance of multikernel FC for all individual behaviors. As can be seen, the prediction performance varies widely across behavioral measures. All 16 cognitive and 9 personality measures were significantly predicted better than chance, while 6 of the 11 mental health measures were significantly predicted. On average, across behavioral measures that were predicted better than chance, the correlation between observed and predicted values for cognition was $0.316 \pm 0.126$ (mean ± std), personality was $0.103 \pm 0.044$, and mental health was $0.132 \pm 0.053$.

Thus, prediction performance was better for cognition than personality or mental health. For example, the best predicted cognitive measure was crystallized cognition with an accuracy of

$r = 0.530$, while the best predicted personality measure was positive urgency with an accuracy of 0.143 and the best predicted mental health measure was total psychosis symptoms with an accuracy of 0.184. Henceforth, we will focus on the 31 behavioral measures that were significantly predicted by multikernel FC.

**Predictive brain-network features cluster together within behavioral domains.** Most previous studies have focused on predicting a small number of behavioral measures. By considering a large number of behavioral measures across multiple behavioral domains, we were able to explore the question of whether predictive brain-network features were shared or unique across behavioral measures. The multikernel regression models were inverted[56], yielding a 419 × 419 predictive-feature matrix for each brain state (rest, MID, SST, and N-back) and each behavioral measure. Haufe's inversion approach yields a positive (or negative) predictive-feature value for an edge, indicating that higher FC for the edge was associated with predicting greater (or lower) behavioral values.

Most previous studies have interpreted the regression weights[19,24] or selected features[14,23] of predictive models, which

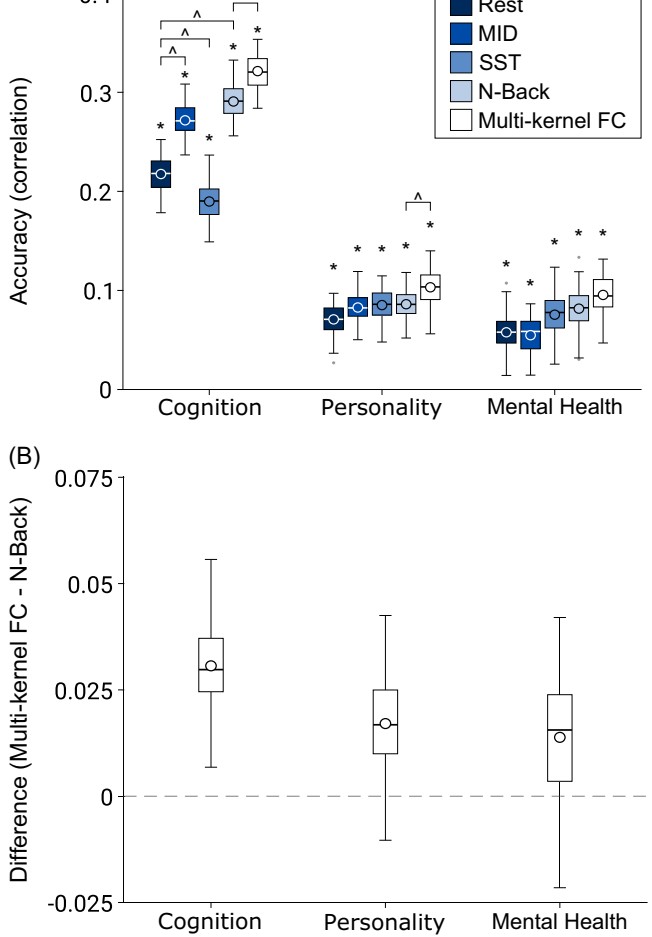

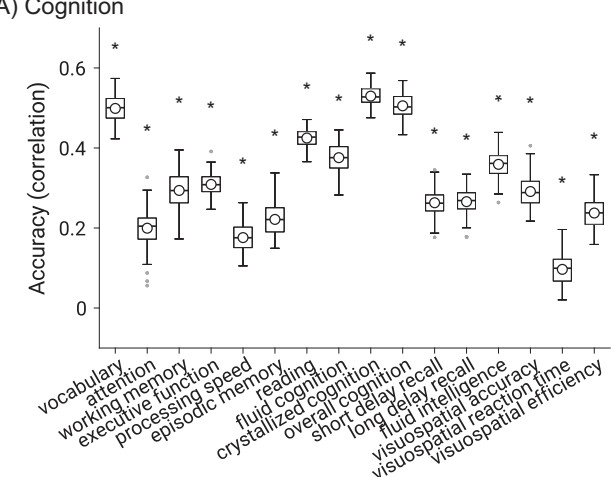

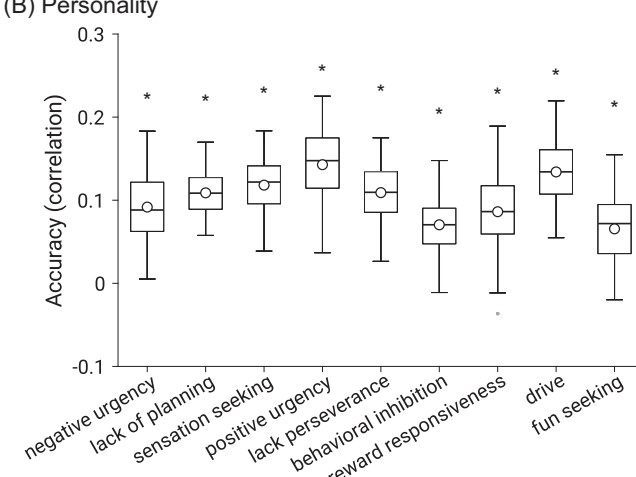

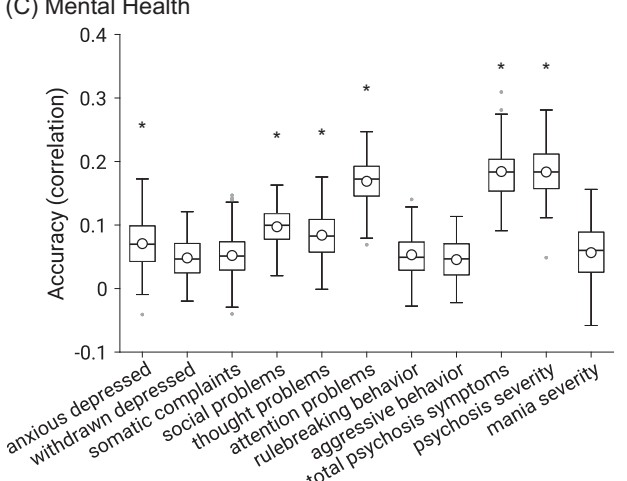

**Fig. 2 Cross-validated prediction performance. A** Prediction performance (Pearson's correlation between observed and predicted values) using kernel ridge regression for resting state and task states (MID, SST, and N-back). Multikernel FC utilized FC from all 4 brain states for prediction. * denotes above-chance prediction after correction for multiple comparisons (FDR $q < 0.05$). ^ denotes statistically significant difference between approaches after correction for multiple comparisons (FDR $q < 0.05$). Note that we only compared multikernel FC with N-back, the best single-kernel regression performer. For the single-kernel results, we only compared resting state with each of the three task states. The boxplots show the average accuracy within each behavioral domain across 120 replications. The cognition domain comprises measures such as fluid cognition and working memory. The (impulsivity-related) personality domain comprises measures such as sensation seeking and behavioral inhibition. The mental health domain comprises measures such as thought problems and psychosis severity. For each boxplot, the horizontal line indicates the median and the circle indicates the mean. The bottom and top edges of the box indicate the 25th and 75th percentiles, respectively. Outliers are defined as data points beyond 1.5 times the interquartile range. The whiskers extend to the most extreme data points not considered outliers. Task FC appeared to only improve prediction performance for cognition, but not (impulsivity-related) personality or mental health. Multikernel FC improved prediction performance for cognition and personality, but not mental health. Similar conclusions were obtained using coefficient of determination (COD) instead of Pearson's correlation as a measure of prediction performance (Supplementary Fig. 1). MID: monetary-incentive delay; SST: stop-signal task. **B** The average difference in accuracy (Pearson's correlation between observed and predicted values) between the multikernel FC and N-back models across 120 replications. Source data are provided as a Source Data file.

**Fig. 3 Cross-validated prediction performance (Pearson's correlation between observed and predicted values) using multikernel ridge regression by exploiting resting FC, MID-FC, SST-FC, and N-back FC jointly. A** Cognitive measures. **B** (Impulsivity-related) Personality measures. **C** Mental health measures. * denotes above-chance prediction after correcting for multiple comparisons (FDR $q < 0.05$). The boxplots show the accuracy across 120 replications. Note the different scales across the three panels. The same set of behavioral measures were predicted better than chance when using coefficient of determination (COD) instead of Pearson's correlation as a measure of prediction performance (Supplementary Fig. 2). Boxplot convention is the same as Fig. 2. Source data are provided as a Source Data file.

can be highly misleading[56]. For example, suppose we seek to predict the target variable $y$ (e.g., fluid cognition) from the FC of two edges ($FC1$ and $FC2$). In this hypothetical example, let us further assume that $FC1 = y - motion$, and $FC2 = motion$. Then a prediction model with 100% accuracy would be $1 \times FC1 + 1 \times FC2$. The regression weights of this model are both one for $FC1$ and $FC2$. Therefore, if we interpreted the weights of the regression model, we would conclude that both $FC1$ and $FC2$ are strongly related to the target variable $y$. Haufe's inversion resolves this issue by computing the covariance between the predicted target variable and the FC of the two edges. Using the Haufe approach, $FC2$ will be assigned a weight of zero, consistent with the intuition that $FC2$ is not related to the target variable even though it is helpful for predicting the target variable. As will be shown in additional control analyses, the predictive features were more robust across regression models with Haufe's approach, further underlining the importance of this inversion process.

Supplementary Figs. 3–6 show the predictive-feature matrices of all 31 significantly predicted behaviors for all brain states. The predictive features were similar within each behavioral domain, but there was a number of notable exceptions. For example, within the somatomotor network, resting-state predictive-network features were positive for reward responsiveness, but negative for sensation seeking. To quantify the degree of similarity in predictive-network features across behavioral measures, predictive-feature matrices for each behavioral measure were concatenated across brain states and correlated between behaviors, yielding a $31 \times 31$ matrix (Fig. 4A). Behavioral measures are ordered based on ABCD's classification of these measures into cognition, personality, and mental health behavioral domains, so we refer to this ordering as "hypothesis-driven". If a pair of behavioral measures exhibited a high value (green) in the matrix (Fig. 4A), then this indicates that the two behavioral measures are predicted by highly similar network features.

The predictive-feature matrices were highly similar within each behavioral domain (Fig. 4A). For each pair of behavioral measures, we computed the proportion of network blocks for which the predictive-network features exhibited consistent directionality (positive or negative) across the pair of behavioral measures (Supplementary Fig. 7). Among all predictive-feature matrices, 49.3% of network blocks were positive, while 50.7% of network blocks were negative, so the chance level of the predictive-feature values having the same sign was 50.01%. Within each behavioral domain, the proportion of consistent predictive-network features across behavioral measures was significantly greater than chance: 74% for cognition ($p < 0.001$), 58% for personality ($p < 0.001$), and 67% for mental health ($p < 0.001$). Each within-domain proportion was also significantly greater than the corresponding between-domain proportions ($p < 0.015$). The exception was the relatively high between-domain proportion for mental health and personality, consistent with Fig. 4A.

Instead of ordering the behavioral measures in a hypothesis-driven fashion (Fig. 4A), we also reordered the behavioral measures by hierarchical clustering of the predictive-feature matrices (Fig. 5A). The hierarchical clustering yielded three data-driven behavioral clusters (Fig. 5A) that were highly similar to the hypothesis-driven behavioral domains (Fig. 4A). The predictive-feature matrices were again much more similar within each data-driven behavioral domain.

**Predictive brain-network features were similar across brain states.** Given that predictive-network features were similar within

a behavioral domain, the predictive-feature matrices were averaged across behaviors, yielding a predictive-feature matrix for each behavioral domain and each brain state (Fig. 6). The predictive features were similar across different brain states, but there were a number of notable exceptions. For example, within the somatomotor network, predictive-network features were negative for cognition in the resting state, but positive for cognition in the N-back condition. To quantify the degree of similarity in predictive-network features across brain states, the 12 predictive-feature matrices (Fig. 6) were correlated with each other, yielding a $12 \times 12$ similarity matrix (Fig. 4B).

The predictive-feature matrices were similar across brain states within each behavioral domain (Fig. 4B). Performing the same analyses using the data-driven behavioral clusters yielded similar results (Fig. 5B and Supplementary Fig. 8). For each pair of brain states, we computed the proportion of network blocks for which the predictive-network features exhibited consistent directionality (positive or negative) across the pair of brain states. Within each behavioral domain, the proportion of consistent predictive-network features across brain states was significantly greater than chance: 63% for cognition ($p < 0.001$), 70% for personality ($p < 0.001$), and 68% for mental health ($p < 0.001$).

Overall, these results suggest that predictive-network features were more similar within behavioral domains (cognition, personality, and mental health) than across behavioral domains. Furthermore, predictive-network features were similar across brain states. Critically, the similarity in predictive-network features cannot be completely explained by similarity among the actual behavioral measures themselves (Supplementary Fig. 9). For example, "lacking of planning" and "sensation seeking" shared predictive features with cognitive measures (Fig. 5A), although the behavioral measures themselves were more correlated with other mental health and personality measures (Supplementary Fig. 9). As another example, the average correlations of predictive-network features within the cognition, personality, and mental health domains were 0.68 ± 0.19 (mean ± std), 0.21 ± 0.33, and 0.46 ± 0.27 respectively. On the other hand, the average correlations among the raw behavioral scores with the cognition, personality, and mental health domains were 0.29 ± 0.22, 0.17 ± 0.16, and 0.34 ± 0.27 respectively.

**Predictive models and predictive-network features generalize to other behavioral measures within the same behavioral domain.** Given that predictive-network features were similar within behavioral domains, we further performed a cross-behavior prediction analysis where each of the 31 significantly predicted behaviors was predicted using the prediction models of *other* behavioral measures from the same behavioral domain (or different domains). We found that cross-behavior predictions for all behavioral domains were significantly better than chance using models from the same domain ($p < 0.001$ for all 3 behavioral domains, Fig. 7A). Within-domain cognitive and mental health models also predicted better than personality models applied to either cognitive or mental health measures ($p < 0.011$). There was no significant difference between other within-domain and between-domain predictions.

In a second analysis, for each behavior, we selected the top predictive features of *other* behavioral measures from the same behavioral domain (or different domains). The top FC edges were then aggregated and used to predict the behavior. As shown in Fig. 7C, we found that top features from the same behavioral domain significantly predicted cognition ($p < 0.001$), personality ($p = 0.002$), and mental health ($p < 0.001$). When predicting cognition, within-domain cognitive features performed better than features from personality or mental health models

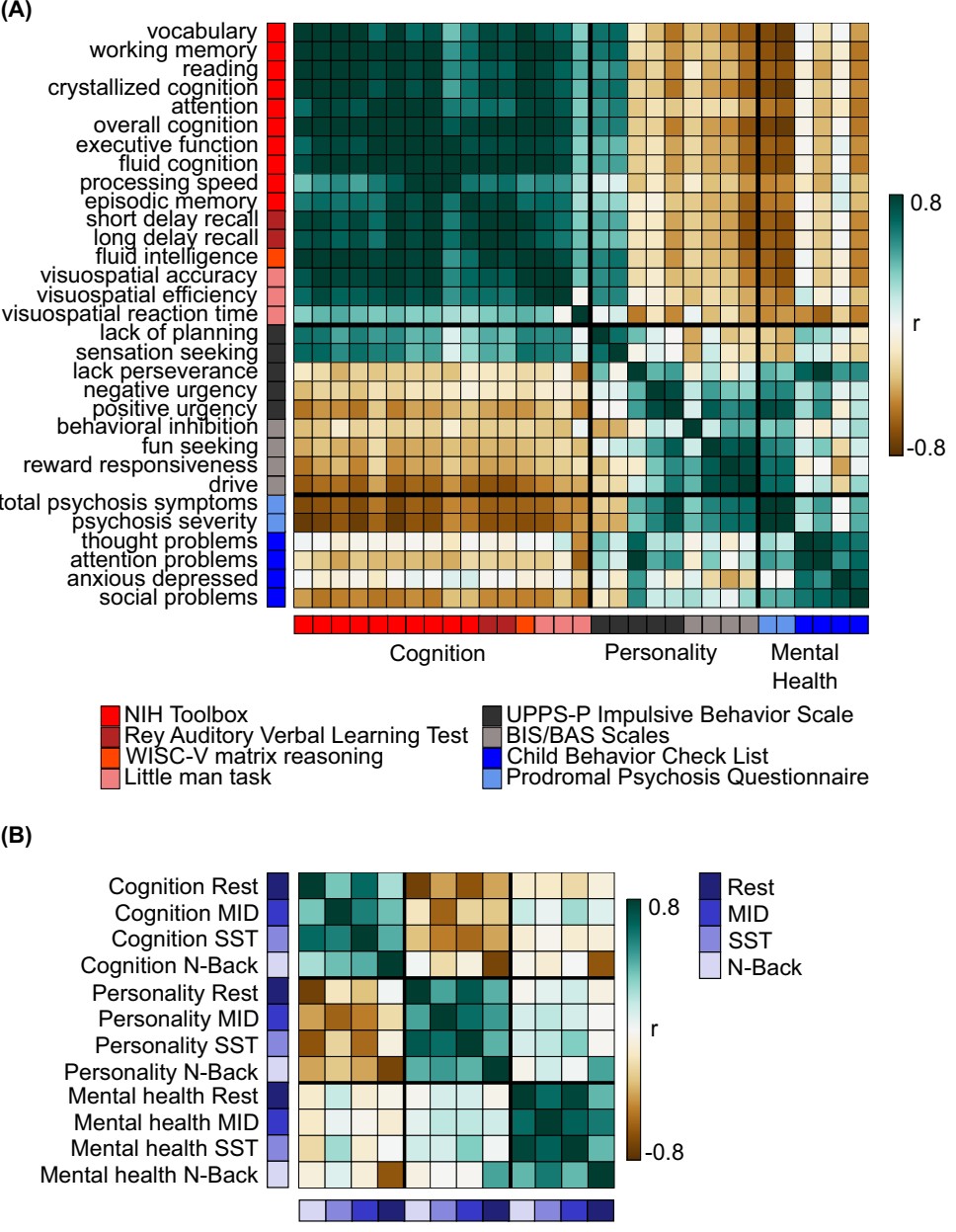

**Fig. 4 Predictive-network features are similar within hypothesis-driven behavioral domains and across brain states. A** Correlations of predictive-feature matrices (Supplementary Figs. 3–6) across behavioral measures. The predictive-feature matrices were concatenated across brain states and correlated across behavioral measures. If a pair of behavioral measures exhibited a high value (green), then this indicates that the two behavioral measures are predicted by highly similar network features. **B** Correlations of predictive-feature matrices across brain states. Predictive-feature matrices were averaged within each behavioral domain and correlated across brain states. The behavioral measures were ordered and categorized based on ABCD's classification of these measures into cognition, personality, and mental health behavioral domains, so we referred to this ordering as "hypothesis-driven". Supplementary Fig. 10 shows the analog of this figure, but without collapsing across either dimension of brain state or behavior. MID: monetary-incentive delay; SST: stop-signal task.

($p < 0.001$). There was no significant difference between other within-domain and between-domain predictions.

Similar conclusions were obtained using data-driven behavioral domains (Fig. 7B, D). We note that this analysis might overestimate the domain specificity of feature/model transferability because the data-driven behavioral domains were defined based on the predictive-feature matrices. However, we note that the hypothesis-driven results might underestimate the potential of feature/model transferability, so both hypothesis-driven and data-driven results were shown for completeness.

Overall, we found that predictive models and predictive-network features generalized to other behavioral measures within the same

behavioral domain. Within-domain generalizations were often significantly better than between-domain generalizations.

**Distinct brain-network features support the prediction of cognition, personality, and mental health.** Having established that predictive-network features were similar within behavioral domains and across brain states, we investigated the topography of predictive-network features that were shared across states within each behavioral domain. Predictive-feature matrices were averaged within each hypothesis-driven behavioral domain, yielding 12 predictive-feature matrices (one for each behavioral

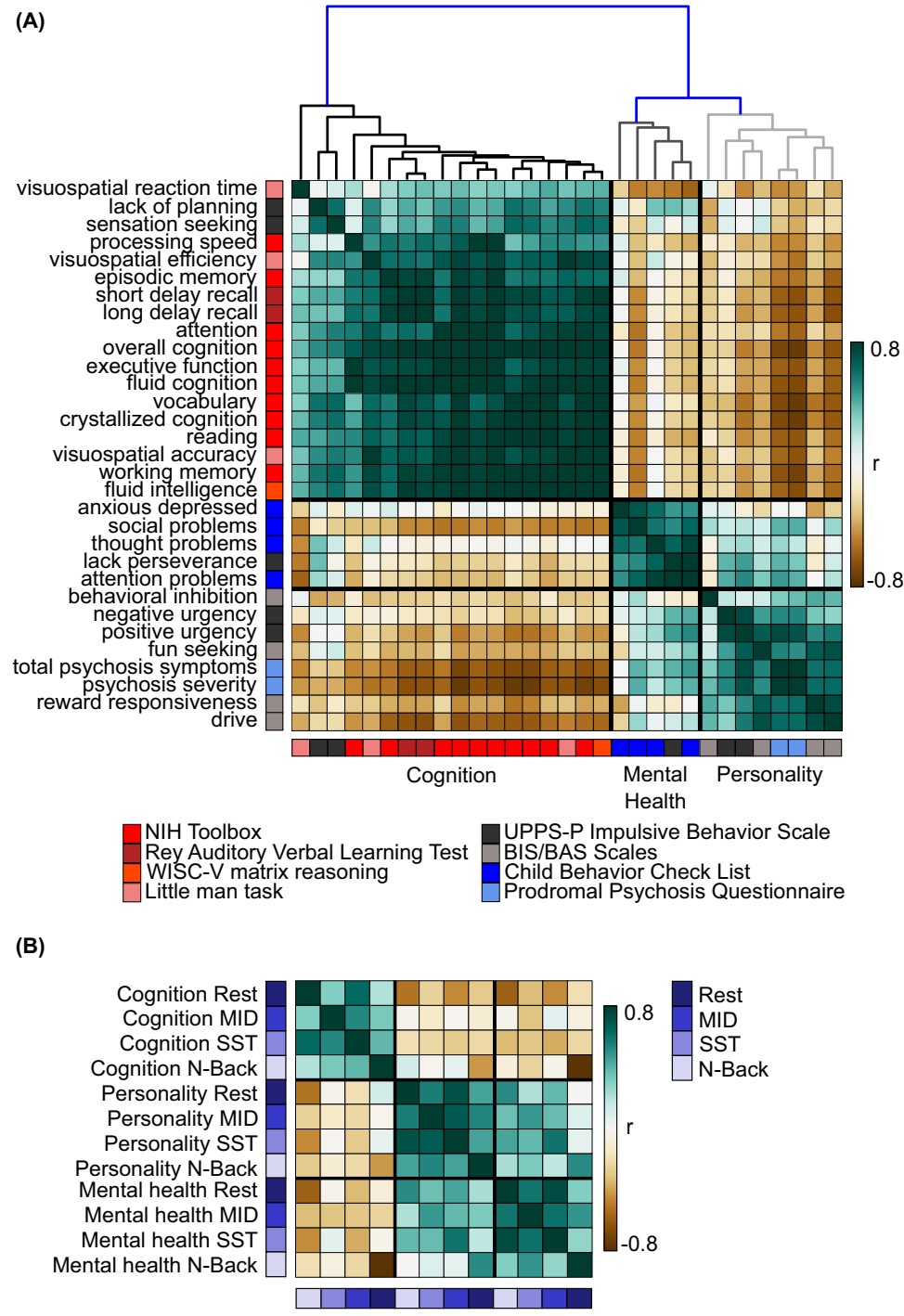

**Fig. 5 Predictive-network features are similar within data-driven behavioral domains and across brain states.** Both panels (**A**) and (**B**) are the same as Fig. 4, except that behavioral measures are ordered and categorized based on the data-driven clusters of cognition, personality, and mental health. These data-driven clusters were obtained by hierarchical clustering of the predictive-feature matrices (Supplementary Figs. 3–6) as indicated by the dendrogram in panel A. Clustering was performed using hierarchical agglomerative average linkage (UPGMA) clustering as implemented in scipy 1.2.1[153]. Supplementary Fig. 11 shows the analog of this figure, but without collapsing across either dimension of brain state or behavior. MID: monetary-incentive delay; SST: stop-signal task.

domain and each brain state, Fig. 6). To limit the number of multiple comparisons, permutation tests were performed for each within-network and between-network block by averaging predictive-feature values within and between 18 networks (FDR $q < 0.05$, Supplementary Fig. 12).

To examine predictive features common across brain states, we averaged the predictive-feature matrices across all brain states,

considering only network blocks that were significant and exhibited the same directionality across states (Fig. 8A). This conjunction thus highlights predictive-network features that are shared across brain states and across behavioral measures within a behavioral domain. Figure 8B illustrates the connectivity strength obtained from averaging within each significant block. Figure 8C, D illustrate the predictability of each cortical region

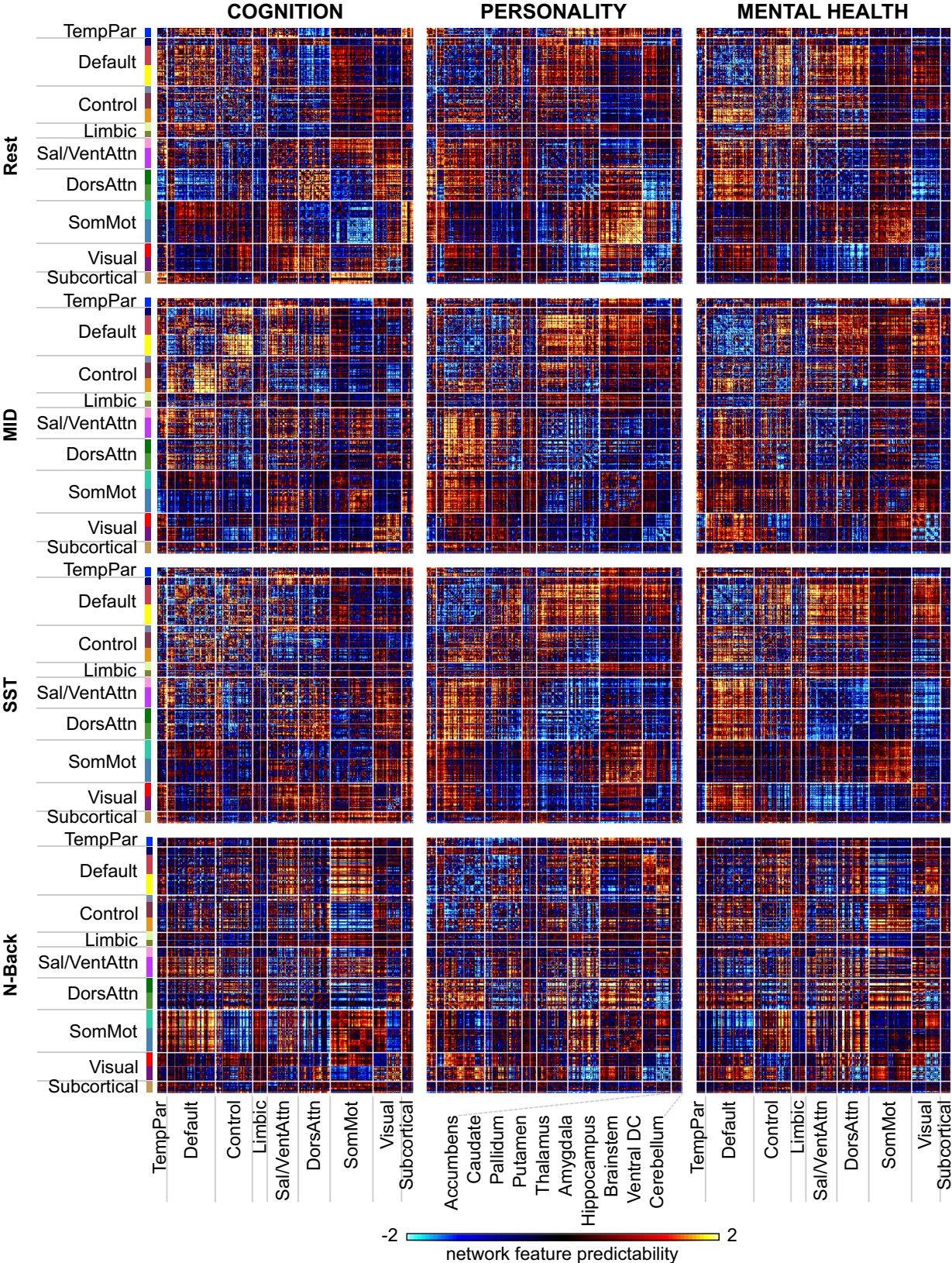

**Fig. 6 Predictive-feature matrices for each brain state (rest, MID, SST, and N-back) averaged across all behavioral measures within each hypothesis-driven behavioral domain (cognition, personality, and mental health).** For visualization, the values within each matrix were divided by their standard deviations (across all entries in the matrix).

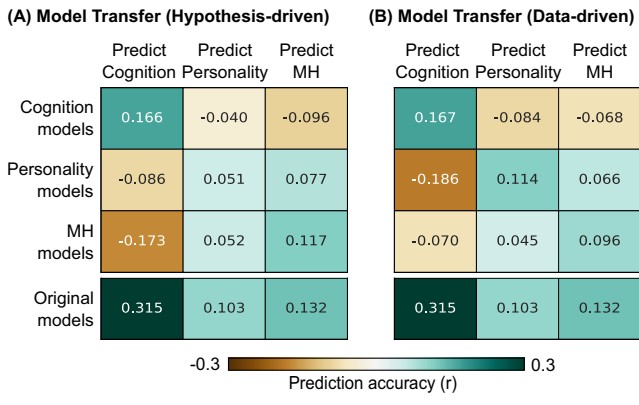

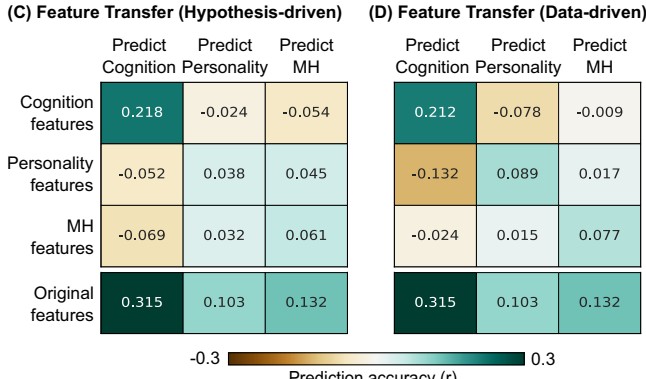

**Fig. 7 Predictive models and predictive-network features generalize to other behavioral measures within the same behavioral domain. A** Cross-behavior predictive performance averaged within each behavioral domain of cognition, personality, and mental health (MH) is shown in the first three rows. Each behavioral measure was predicted by averaging predictive models of other behavioral measures in the same domain (diagonal cells) or different domains (off-diagonal cells). As a reference, average prediction performance of behavioral measures (predicted better than chance) from the original multikernel FC models (Fig. 3) is shown in the fourth row. **B** Same as (**A**), but using data-driven behavioral domains (Fig. 6). **C** Cross-behavior predictive performance averaged within each behavioral domain of cognition, personality, and mental health (MH) is shown in the first three rows. Each behavioral measure was predicted by averaging top predictive-network features of other behavioral measures in the same domain (diagonal cells) or different domains (off-diagonal cells). As a reference, average prediction performance of behavioral measures (predicted better than chance) from the original multikernel FC models (Fig. 3) is shown in the fourth row. **D** Same as (**C**), but using data-driven behavioral domains.

obtained by summing the rows of Fig. 8A for positive and negative predictive-feature values separately (see subcortical regions in Supplementary Fig. 13A).

Consistent with the results in previous sections (Figs. 5 and 6), the patterns of predictive-network features were distinct across the three behavioral domains (Fig. 8 and Supplementary Fig. 13A). Out of the 171 unique blocks of large-scale network features (Fig. 8A), 15 included significant predictions for cognition, 25 for personality, and 22 for mental health. Critically, cognition shared only 2 blocks with personality and 1 block with mental health (<14%), suggestive of largely distinct predictive networks. Mental health and personality shared 8 blocks (~35%), consistent with greater predictive-network similarity, despite general distinctiveness.

Cognitive performance of individual participants was predicted by a distributed set of large-scale network features (Fig. 8A, B) with default C, control A, somatomotor B, and salience/ventral attention

A networks being particularly prominent (Fig. 8C, D and Supplementary Fig. 14). For example, greater connectivity of somatomotor network B with subcortical and default network A regions was predictive of higher cognitive scores (i.e., better cognition). As another example, greater connectivity between salience/ventral attention network A and default network C, as well as lower connectivity between salience/ventral attention network A and control networks, were predictive of better cognition.

Personality measures of individual participants were predicted by a distributed set of large-scale network features (Fig. 8A, B) with default A/B and dorsal attentional A/B networks being particularly prominent (Fig. 8C, D and Supplementary Fig. 4). For example, greater connectivity between default networks A/B and dorsal attention networks A/B was predictive of greater personality scores (i.e., greater impulsivity and sensitivity to reward/punishment). On the other hand, lower connectivity within default networks A/B was predictive of greater impulsivity-related traits.

Mental health of individual participants was predicted by a distributed set of large-scale network features (Fig. 8A, B) with default A/B and control A networks being particularly prominent (Fig. 8C, D and Supplementary Fig. 14). For example, greater connectivity between default network B and control network A was predictive of larger mental health scores (i.e., worse mental health). On the other hand, lower connectivity within default networks A/B was predictive of worse mental health.

As a control analysis, we utilized the previously derived data-driven clusters of cognition, personality, and mental health (Fig. 5) to perform the same analyses, yielding highly similar results (Supplementary Figs. 13B, 15 and 16). Average correlations between the hypothesis-driven and data-driven predictive-feature matrices (before thresholding for significant network blocks) were $r = 0.99$ (cognition), 0.84 (personality), and 0.90 (mental health).

**Control analyses**. We performed several additional control analyses to ensure robustness of our results. First, we regressed age and sex (in addition to FD/DVARS) from the behavioral variables before prediction, which only decreased the prediction performance slightly (Supplementary Fig. 17).

Second, instead of multikernel FC prediction, we averaged functional connectivity across all brain states[26] and utilized the resulting mean FC for kernel regression. We found that mean FC yielded statistically worse prediction performance for cognition compared with multikernel regression (Supplementary Fig. 18), but not personality and mental health. Interestingly, mean FC was also numerically (but not statistically) worse than n-back FC for predicting cognition ($r = 0.28$ vs. $r = 0.29$). Overall, this suggests that although multikernel approach might have benefited from more data per participant, more data in itself did not improve prediction performance. Instead, the multikernel approach was able to make better use of more data across different brain states to improve behavioral prediction.

Third, to ensure our results were robust to the regression model, we also performed linear ridge regression. We obtained similar prediction performance, but linear regression achieved worse COD (Supplementary Fig. 19). Remarkably, the predictive-feature matrices were highly similar for both linear regression and kernel regression (average $r = 0.99$), suggesting that the predictive-feature matrices are robust to the choice of regression algorithm. We note that if we interpreted the regression weights directly without Haufe model inversion, then the agreement between kernel regression and linear regression "only" achieved an average correlation of $r = 0.66$. This observation confirms the importance of inverting the regression models[56].

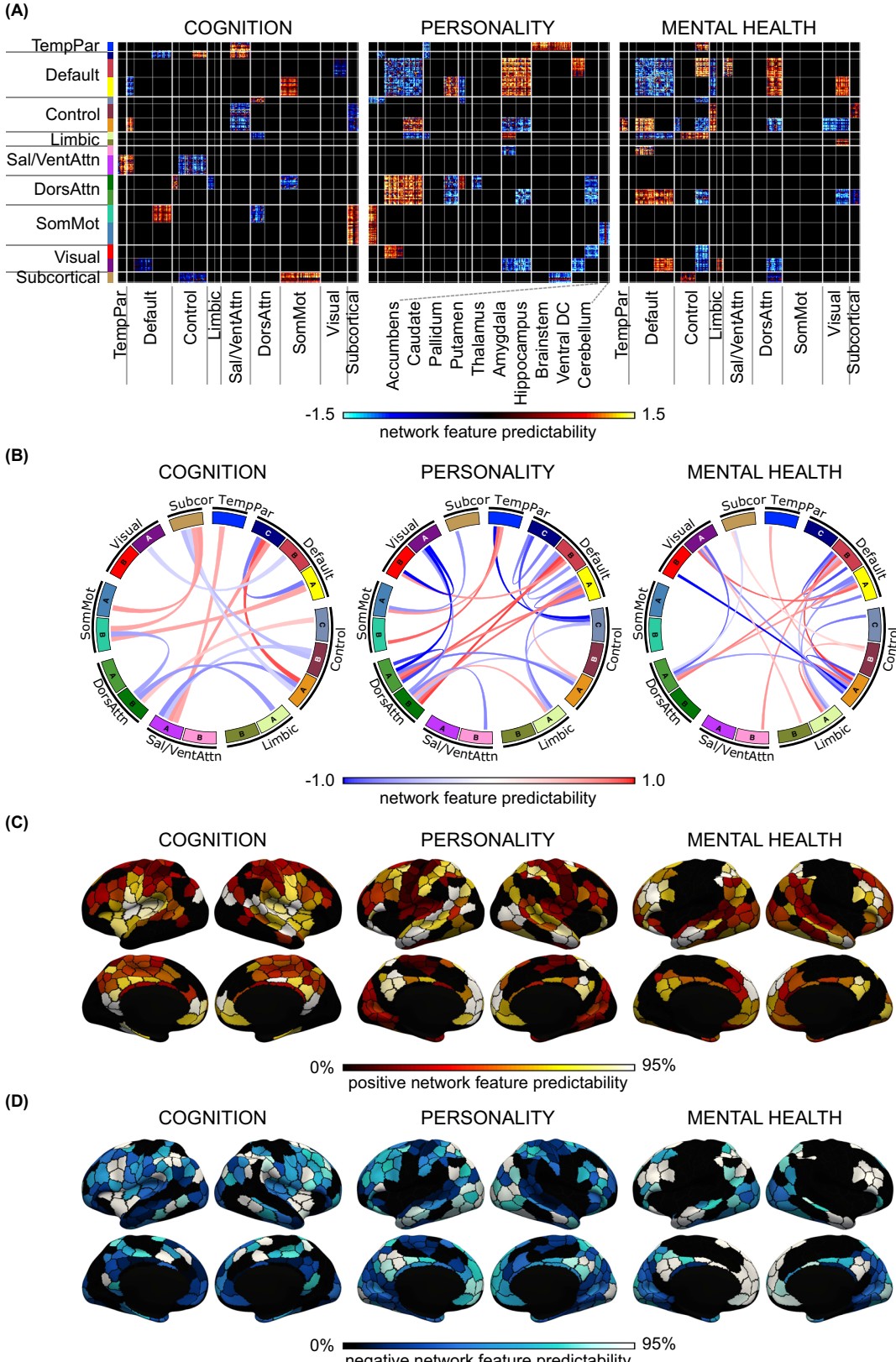

Fourth, many participants were excluded due to image-quality issues (Fig. 1A). The resulting sample had a higher proportion of female and White participants with higher family income than excluded participants (Supplementary Table 3). Image-quality issues were also correlated with various behavioral measures (Supplementary Tables 4–7). To increase confidence that our

results were applicable to the broader population, we performed three analyses. In the first analysis, we applied a bandstop filter to remove respiratory pseudomotion, thus retaining 21.7% more participants ($N = 2262$). In the second analysis, we additionally loosened the image QC criteria, thus retaining double the participants ($N = 3744$). In the third analysis, for each behavioral

**Fig. 8 Brain-network features that support individual-level prediction of cognition, personality, and mental health. A** Predictive-feature matrices averaged across brain states, considering only within-network and between-network blocks that were significant across all four brain states (rest, MID, SST, and N-back). **B** Predictive-network connections obtained by averaging the matrices in panel (**A**) within each between-network and within-network block. **C** Positive predictive features obtained by summing positive predictive-feature values across the rows of panel (**A**). A higher value for a brain region indicates that stronger connectivity yielded a higher prediction for the behavioral measure. **D** Negative predictive features obtained by summing negative predictive-feature values across the rows of panel (**A**). A higher value for a brain region indicates that weaker connectivity yielded a greater prediction for the behavioral measure. In both panels (**C**) and (**D**), the color of each parcel corresponds to the percentile of predictive-feature values among 400 parcels. See Supplementary Fig. 13A for the subcortical maps. For visualization, the values within each predictive-feature matrix in panel A were divided by their standard deviations across all entries in the predictive-feature matrix. The current figure utilized hypothesis-driven behavioral domains. Conclusions were highly similar using data-driven behavioral clusters (Supplementary Figs. 15 and 16), as well as other control analyses (Supplementary Figs. 21–23).

measure, we selected subsets of participants from the main sample ($N = 1858$) that matched the age, sex, household income, racial composition, and behavioral distributions of the full ABCD sample, yielding an average of 921 participants (min = 563, max = 1073) for each behavioral measure (Supplementary Table 9). There was no statistical difference between the included and excluded participants for the 5 characteristics (age, sex, household income, racial composition, and behavioral distributions) for the 36 behavioral measures after correction for multiple comparisons with FDR $q < 0.05$. By construction, these participants have low motion/imaging artifacts (since participants are a subset of the main sample) and are representative of the entire ABCD sample (due to our matching procedure). The predictive-feature matrices remained highly similar to the original matrices: $r = 0.93$ (first analysis, Supplementary Fig. 21), $r = 0.85$ (second analysis, Supplementary Fig. 22), and $r = 0.85$ (third analysis, Supplementary Fig. 23).

Lastly, we computed the predictive-feature matrices based on the single-kernel regression models and found that the results were highly similar to the predictive-feature matrices of the multikernel regression model (average $r = 0.95$).

## Discussion

In a large sample of typically developing children, we found that compared with resting FC, task FC of certain tasks improves prediction of cognition, but not (impulsivity-related) personality or mental health. Integrating resting FC and task FC further improves prediction of cognition and personality, but not mental health. By considering a large number of measures across cognition, personality, and mental health, we found that these behavioral domains were predicted by largely distinct patterns of brain-network features. However, within a behavioral domain (e.g., cognition) and across brain states, the predictive-network features were similar, suggesting the potential existence of shared neural mechanisms explaining individual variation within each behavioral domain.

**Predictive brain-network features cluster together within behavioral domains.** Previous task-FC behavioral prediction studies have typically focused on specific cognitive traits, such as fluid intelligence test performance[23], attention[25] or reading comprehension[27]. By exploring a wide range of behavioral measures, we gained insights into shared and unique predictive-network features across traits within the same domain and across domains, as well as across brain states (rest and task). While there were differences among predictive-network features within a behavioral domain (Supplementary Figs. 3–6), the similarity was striking (Figs. 4 and 5). This was especially the case for the cognitive domain (Figs. 4 and 5 and Supplementary Figs. 3–6).

Decades of studies, ranging from lesion to functional neuroimaging studies, have suggested the existence of brain networks that are specialized for specific cognitive functions[57–61].

For example, language tasks activate a specific network of brain regions[62–64]. Another example is the specific loss of episodic memory but not language after medial temporal lobe lesions[65,66]. Of course, the networks that preferentially underpin aspects of behavior do not work in isolation, and many studies have also emphasized information integration across specialized brain networks[40,42,67,68]. Lesion studies have also suggested that damage to connector hubs leads to deficits in multiple functional domains[69]. Thus, while we did not expect predictive-network features to be completely different across cognitive measures, we did not anticipate such strong similarity.

Similarly, in the case of mental health measures, while diagnostically distinct psychiatric disorders are likely the result of differentially disrupted brain systems, there is significant comorbidity among disorders and overlap in clinical symptoms[70–72]. Certain brain circuits have also been disproportionately reported to be transdiagnostically aberrant across multiple psychiatric and neurological disorders[43,73–76]. For instance, there is evidence for the central role of frontoparietal network disruptions across psychiatric diagnosis[77,78]. Therefore, similarly to cognition, we did not expect predictive-network features to be completely different across mental health measures, but the degree of similarity was still surprising. These findings underscore the importance of studying multiple facets of psychopathology simultaneously to better characterize covariation among symptoms and to redefine psychiatric nosologies[79,80].

One possibility is that the regression models might be predicting a broad behavior rather than the specific behaviors they were trained on. For example, in the case of cognition, perhaps the network features were partially predicting the $g$ factor, a general cognitive ability that can account for half of the variance of cognitive test scores[81]. In the case of mental health, the network features might be partially predicting the $p$ factor, a general psychopathology factor that reflects individuals' susceptibility to develop psychopathologies[82]. The similarity in predictive-network features across the personality measures was less surprising since the personality measures we considered were mostly impulsivity-related. Thus, the regression models might be partially predicting an overall impulsivity trait[83].

Indeed, behavioral measures are thought to be supported by a combination of shared and distinct factors[84–87]. However, it is unlikely that FC is equally sensitive to all behavioral factors. In particular, FC might be more sensitive to certain shared and distinct factors, while being insensitive to other shared and distinct factors[88]. Given that the predictive-network features were more strongly correlated (within a behavioral domain) than the behavioral measures themselves, this suggests that the relative contributions of shared FC-sensitive factors were larger than shared non-FC-sensitive factors.

**Distinct brain-network features support the prediction of cognition, personality, and mental health.** We found that cognitive performance was predicted by distributed network features

with default C, control A, salience/ventral attention A, and somatomotor B networks being particularly prominent. The overrepresentation of default, control, and salience/ventral attention networks was consistent with previous studies[24,25,89]. The prominent role of the somatomotor network was more surprising, although somatomotor regions have been associated with fluid intelligence test performance[23], attention[25], and general cognitive dysfunction[74].

Mental health measures were predicted by distributed network features with default A/B and control A networks being particularly prominent. Previous studies have linked the default and control networks to multiple psychiatric disorders and symptom profiles[43,90,91]. Interestingly, while the predictive-network features for cognition and mental health both involved the default network, different subnetworks were involved in the two behavioral domains: default network C in cognition and default networks B/C in mental health.

Finally, (impulsivity-related) personality measures were predicted by distributed network features with default A/B and dorsal attention A/B networks being particularly prominent. Most impulsivity studies have typically highlighted fronto-striatal circuits[92–96]. However, very few studies have investigated functional connectivity at the whole-brain level across all networks and these studies only focused on adults[97]. Therefore, our study complements the literature by providing evidence for the importance of default and attentional networks in predicting impulsivity in children.

**Resting and task-network organization**. A surprising result is that the predictive-network features were similar across brain states (rest, MID, SST, and N-back) for all behavioral domains, particularly in the case of personality and mental health. On the one hand, task-network reorganization has been shown to influence cognitive performance[32,98]. On the other hand, our results are consistent with studies showing that task states only modestly influence functional connectivity[28,29,99] with inter-individual differences dominating task modulation[34].

We note that a previous study[27] suggested that the regression models utilized different network features for prediction across different brain states, while another study[23] suggested that there was substantial overlap in predictive-network features across resting FC and task FC. These discrepancies might arise because previous studies only interpreted the most salient edges selected for prediction, which might yield unstable results. Here, we followed the elegant approach of Haufe and colleagues (2014) to invert the prediction models, leading to highly consistent predictive-network features across two regression models (kernel regression and linear regression). Omitting the inversion step leads to weaker agreement between the two models.

Consistent with previous studies[23,100,101], we found that task FC outperforms resting FC for the prediction of cognitive performance, at least in the case of N-back and MID. Although resting FC was better than SST-FC for predicting cognition (Fig. 2), we note that there was more resting-fMRI data than SST-fMRI data, which might explain the gap in performance. Here, we did not control for fMRI duration because our goal was to maximize prediction performance and to quantitatively characterize the predictive-network features. Similarly, the prediction improvement from integrating information across brain states (multikernel regression) partly comes from the use of more fMRI data per child, but at least in the case of cognition, the improvement was not entirely due to more data (Supplementary Fig. 18).

Consistent with previous studies[24,26,27], we found that combining resting FC and task FC improved prediction of cognition.

Extending upon this work, we demonstrated that combining resting FC and task FC modestly improved prediction of personality, but not mental health. We also found that regardless of using resting FC, task FC, or both resting FC and task FC, greater performance was achieved for predicting cognition than personality or mental health. This is again consistent with previous studies relating resting fMRI with interindividual variation in multiple behavioral domains[16,89,102,103].

**Strengths and limitations**. One strength of our study is the use of a whole-brain connectomics approach to predict a wide range of behavioral traits. Many neurodevelopmental studies have focused on specific brain circuits[104–111]. Yet, the human brain comprises functional modules that interact as a unified whole to support behavior[112–114]. Therefore, whole-brain network-level approaches could provide critical insights into neurodevelopment that might be missed by studies focusing on specific networks. Our results were also robust across brain states, across simple and more advanced predictive algorithms, and across recruitment sites.

However, since the ABCD cohort comprises typically developing children, it is unclear how our results, especially those pertaining to mental health, might generalize to groups with clinical diagnoses. Furthermore, the cross-sectional nature of our study and the limited age range of the participants prevented us from thoroughly examining neurodevelopmental changes across time or age. Whole-brain neurodevelopmental studies have shown that functional networks become more distributed throughout adolescence[115–117]. As such, it remains to be seen how the predictive-network features from our study might be similarly affected by the developmental process.

In our study, we used resting FC and task FC to jointly predict behavioral measures. Therefore, a participant was included only if the imaging data of every brain state survived QC. This resulted in a much smaller sample size than if we analyzed each brain state independently ([118], Supplementary Fig. 24). The resulting main sample ($N = 1858$) was also less representative of the full sample in terms of age, sex, racial composition, and household income. In a control analysis, we subsampled participants from the main sample to match characteristics of the full sample, thus simultaneously addressing issues about data quality and representativeness of the sample. However, this control analysis further reduced sample sizes. Thus, it remains an open question how to maximize sample sizes, while maintaining data quality and representativeness.

Furthermore, to match processing across resting and task states, task activations were not regressed from the task-state data. Therefore, the prediction improvement of certain task FC over resting FC in the cognitive domain might be partially due to task activation[119]. However, we note that a previous study has suggested that task-induced changes in FC might predict phenotypes independent of task activation[120]. Future work will benefit from further differentiation of task activation from task-induced FC changes.

Finally, although most behavioral measures were predicted better than chance, the prediction accuracies were low, especially for personality and mental health measures[121]. Further improvement will be necessary for clinical utility. From the imaging perspective, improved imaging acquisition and modeling, as well as aggregation across larger sample size might improve prediction performance[122–124]. From the behavioral perspective, improvement might be achieved by using composite measures[89] or more innovative digital approaches allowing for greater sampling frequency and thus better reliability[125–127]. Lastly, the use of nonimaging features could further enrich our predictive models[128].

## Methods

**Participants**. We considered data from 11875 children from the ABCD 2.0.1 release. All data used in this paper were collected by the ABCD study. The Institutional Review Board (IRB) at the University of California, San Diego, approved all aspects of the ABCD study[129]. Parents or guardians provided written consent, while the child provided written assent[130]. After strict preprocessing quality control (QC) and considering only participants with complete rest fMRI, task fMRI, and behavioral data, our main analyses utilized 1858 unrelated children (Fig. 1A). See further details below.

**Imaging acquisition and processing**. Images were acquired across 21 sites in the United States with harmonized imaging protocols for GE, Philips, and Siemens scanners[131]. We used structural T1, resting fMRI, and task fMRI from three tasks: monetary-incentive delay (MID), N-back, and stop-signal task (SST). For each participant, there were four resting-fMRI runs. Each resting-fMRI run was 300-secs long. For each participant, there were two runs for each fMRI task. Each MID run was 322.4-secs long. Each N-back run was 289.6-secs long. Each SST run was 349.6-secs long. See Supplemental Methods S1 for details.

Minimally preprocessed T1 data were used[132]. The structural data were further processed using FreeSurfer 5.3.0[133–138], which generated accurate cortical surface meshes for each individual. Individuals' cortical surface meshes were registered to a common spherical coordinate system[135,136]. Individuals who did not pass recon-all QC[132] were removed.

Minimally preprocessed fMRI data[132] were further processed with the following steps: (1) removal of initial frames, number of frames removed depended on the type of scanner;[132] and (2) alignment with the T1 images using boundary-based registration[139] with FsFast (http://surfer.nmr.mgh.harvard.edu/fswiki/FsFast). Functional runs with boundary-based registration (BBR) costs greater than 0.6 were excluded. Framewise displacement (FD)[140] and voxel-wise differentiated signal variance (DVARS)[141] were computed using fsl_motion_outliers. Volumes with FD > 0.3 mm or DVARS > 50, along with one volume before and two volumes after, were marked as outliers and subsequently censored. Uncensored segments of data containing fewer than five contiguous volumes were also censored[16,142]. Functional runs with over half of their volumes censored and/or max FD > 5 mm were removed. We also excluded individuals who did not have at least 4 min for each fMRI state (rest, MID, N-back, and SST) from further analysis. Supplementary Fig. 20 shows the distribution of censored frames and runs.

The following nuisance covariates were regressed out of the fMRI time series: global signal, six motion correction parameters, averaged ventricular signal, averaged white matter signal, and their temporal derivatives (18 regressors in total). Regression coefficients were estimated from the noncensored volumes. We chose to regress the global signal because we were interested in behavioral prediction and global signal regression has been shown to improve behavioral prediction performance[23,143]. The brain scans were interpolated across censored frames using least-squares spectral estimation[144], band-pass filtered (0.009 Hz ≤ f ≤ 0.08 Hz), and projected onto FreeSurfer fsaverage6 surface space and smoothed using a 6-mm full-width half-maximum kernel.

**Functional connectivity**. We used a whole-brain parcellation comprising 400 cortical regions of interest (ROIs)[52] (Fig. 1B) and 19 subcortical ROIs[53] (Fig. 1C). For each participant and each fMRI run, functional connectivity (FC) was computed as Pearson's correlations between the average time series of each pair of ROIs. FC matrices were averaged across runs from each state, yielding a $419 \times 419$ FC matrix for each fMRI state (rest, MID, N-back, and SST). We note that correlation values were converted to z-scores using Fisher's r-to-z transformation prior to averaging and converted back to correlation values after averaging. Censored frames were ignored when computing FC. To match processing across resting and task states, task activations were not regressed from the task-state data.

**Behavioral data**. We analyzed data from all available dimensional neurocognitive[47] and mental health[48] assessments, yielding 16 cognitive, 11 mental health, and 9 impulsivity-related personality measures. The cognitive measures were vocabulary, attention, working memory, executive function, processing speed, episodic memory, reading, fluid cognition, crystallized cognition, overall cognition, short delay recall, long delay recall, fluid intelligence, visuospatial accuracy, visuospatial reaction time, and visuospatial efficiency. The mental health measures were anxious depressed, withdrawn depressed, somatic complaints, social problems, thought problems, attention problems, rule-breaking behavior, aggressive behavior, total psychosis symptoms, psychosis severity, and mania. The impulsivity-related personality measures were negative urgency, lack of planning, sensation seeking, positive urgency, lack of perseverance, behavioral inhibition, reward responsiveness, drive, and fun seeking. See Supplemental Methods S2 for more details.

Participants who did not have all behavioral measures were excluded from further analysis. As recommended by the ABCD consortium, individuals from Philips scanners were also excluded due to incorrect preprocessing. Finally, by excluding siblings, the main analysis utilized data from 1858 unrelated children (Fig. 1A). For these 1858 children, the length of fMRI data remaining after censoring was 897 ± 232 secs (mean ± std) for resting state, 531 ± 107 secs for MID, 579 ± 126 secs for SST, and 482 ± 86 secs for N-back.

Supplementary Table 3 shows the demographic information of included and excluded participants. Supplementary Table 4 shows the means and standard deviations of the behavioral measures of included and excluded participants. Supplementary Tables 5–7 show the correlation between QC measures and behavioral scores in the included participants. Consistent with previous studies[145], there were associations between behavioral measures and various QC measures. For example, participants with worse cognition had lower-quality data.

**Single fMRI-state prediction**. We used kernel ridge regression to predict each behavioral measure based on resting FC, MID-FC, N-back FC, and SST-FC, separately. We chose kernel regression because of its strong prediction performance in resting-FC-based behavioral prediction[54]. Briefly, let $y_i$ and $FC_i$ be the behavioral measure and FC of training individual $i$. Let $y_t$ and $FC_t$ be the behavioral measure and FC of a test individual. Then, kernel regression would predict the test individual's behavior as the weighted average of the training individuals' behavior, i.e., $y_t \approx \sum_{i \in \text{training set}} \text{Similarity}(FC_i, FC_t) y_i$, where $\text{Similarity}(FC_i, FC_t)$ was defined as the Pearson's correlation between $FC_i$ and $FC_t$. Thus, kernel regression assumed that individuals with more similar FC exhibit similar behavior. To reduce overfitting, an $l_2$-regularization term was included[16,54,143]. Details of this approach can be found elsewhere[16,54,143].

Kernel regression was performed within an inner-loop (nested) cross-validation procedure. More specifically, there were 22 ABCD sites. To reduce sample-size variability across sites, we combined sites together to create 10 "site-clusters", each containing at least 150 individuals (Supplementary Table 8). Thus, participants within a site are in the same site cluster.

We performed leave-3-site-clusters-out nested cross-validation for each behavioral measure with 120 replications. For each fold, a different set of 3 site clusters was chosen as the test set. Kernel ridge regression parameters were estimated from the remaining 7 site clusters using cross-validation. For model selection, the regularization parameter was estimated within the "inner-loop" of the inner-loop (nested) cross-validation procedure. For model evaluation, the trained kernel regression model was applied to all unseen participants from the test site clusters.

Head motion (mean FD and DVARS) was regressed from each behavioral measure before the cross-validation procedure. More specifically, regression coefficients were estimated from the 7 training site clusters and applied to the 3 test site clusters. This regression procedure was repeated for each split of the data into 7 training site clusters and 3 test site clusters.

Prediction performance was measured by correlating predicted and actual measures[14]. When averaging prediction accuracies (correlations) across behavioral measures, the correlations underwent Fisher-r-to-z transformation before averaging and converted back to correlation values after averaging. We also computed coefficients of determinations, which yielded similar conclusions.

**Multistate prediction**. To explore whether combining resting FC and task FC would result in better prediction accuracy, we utilized FC matrices from all four brain states (rest, MID, SST, and N-back) for prediction using a multikernel framework (Supplemental Methods S3). Similarly to single-kernel regression, multikernel regression assumed that participants with similar FC exhibit similar behavioral scores. However, instead of taking into account FC from one fMRI state, here we utilized FC from all four fMRI states.

**Statistical tests of prediction accuracy**. To test whether a model achieved better-than-chance accuracy, we performed permutation tests by shuffling behavioral measures across participants within each site and repeating the entire leave-3-site-clusters-out nested cross-validation procedure. To compare two models, a permutation test was not valid, so the corrected resampled t-test was utilized[146,147]. The resampled t-test corrected for the fact that accuracies of test folds were not independent. The resampled t-test assumed that the performance difference between two models was Gaussian distributed. This assumption was validated with the Kolmogorov–Smirnov test. We corrected for multiple comparisons using FDR ($q < 0.05$). All $p$-values were calculated as two-tailed $p$-values.

**Model generalization across behaviors**. We tested whether prediction models could be generalized across behavioral measures. More specifically, for each of the 31 significantly predicted behavioral measures (Fig. 3), we averaged the predictions of the multikernel FC models from all other behavioral measures from the same behavioral domain (or different behavioral domain). The average prediction was compared with the actual behavioral value to compute cross-behavior prediction accuracy. For example, in the case of generalization within the same behavioral domain, a participant's fluid cognition was predicted as follows. The resting FC and task FC of the participant were fed into the 15 multikernel predictive models (of the other 15 cognitive measures), yielding 15 prediction values. These 15 prediction values were averaged, yielding a final prediction of fluid cognition for the participant. We emphasize that the average prediction did not include the target behavioral variable and was performed independently for each of the 120 cross-validation folds.

**Feature generalization across behaviors**. We tested whether top predictive-network features could be generalized across behavioral measures. More specifically, for each of the 31 significantly predicted behavioral measures, we averaged the predictive-feature matrices across all other behavioral measures from the same behavioral domain (or different behavioral domain) and across all brain states. The top 10% of edges with the highest absolute values in the average predictive-feature matrices were then selected. For each participant, an aggregate FC value was computed as the average FC of selected edges with positive predictive-feature values minus the average FC of selected edges with negative predictive-feature values. The aggregate FC value was used as the prediction for the participant. The prediction was compared with the actual behavioral value to compute cross-behavior prediction performance. We note that scale differences between the aggregate FC value and behavioral measures were not an issue since Pearson's correlation is invariant to scale. We emphasize that the selection procedure did not include the target behavioral variable, was performed independently for each of the 120 cross-validation folds, and no test data was used for the selection.

**Model interpretation**. As can be seen, multikernel FC yielded the best prediction performance. Models estimated for prediction can be challenging to interpret[148]. Here, we utilized the approach from Haufe and colleagues (2014), yielding a $419 \times 419$ predictive-feature matrix for each FC state and each behavioral measure (Supplemental Methods S4). A positive (or negative) predictive-feature value indicates that higher FC was associated with predicting greater (or lower) behavioral values.

To compare similarity between a given pair of predictive-feature matrices, we computed the proportion of network blocks for which the predictive-network features exhibited consistent directionality (positive or negative) between the pair of predictive-feature matrices. More specifically, for each predictive-feature matrix, predictive-feature values were averaged within and between 18 groups comprising 17 cortical networks and subcortical structures (Fig. 1B), yielding an $18 \times 18$ symmetric matrix. For each of $18 \times 18$ network blocks and a given pair of predictive-feature matrices, the predictive-feature values were considered consistent if they were both positive or both negative. Among all predictive-feature matrices, 49.3% of network blocks were positive, while 50.7% of network blocks were negative, so the chance level of the predictive-feature values having the same sign was 50.01%. The proportion of consistent predictive-network blocks was computed for each of 120 cross-validation folds. Statistical significance was tested using the corrected resampled t-test against the chance level of 50.01%[146,147].

The predictive-feature matrices were more similar among behavioral measures within the same behavioral domain (cognition, mental health, and personality) than across domains. Thus, we averaged the predictive-feature matrices within the same behavioral domain (cognitive, mental health, and personality) considering only behavioral measures that were successfully predicted by multikernel FC regression. This yielded a $419 \times 419$ predictive-feature matrix for each fMRI state and each behavioral domain.

Statistical significance of the predictive-feature values was tested using a permutation test (2000 permutations). To limit the number of multiple comparisons, tests were performed for each within-network and between-network block by averaging predictive-feature values within and between 18 networks (Fig. 8B, C). We corrected for multiple comparisons using FDR ($q < 0.05$).

**Multiple-comparison correction**. We performed FDR correction ($q < 0.05$) on all statistical tests that did not involve predictive-feature matrices to determine which behaviors were significantly predicted and would therefore be utilized in subsequent analyses. There was a total of 105 tests and the $p$-value threshold was 0.041. A separate FDR correction ($q < 0.05$) was performed on all tests involving predictive-network features. There was a total of 4149 tests and the $p$-value threshold was 0.029. FDR correction was also performed separately for Supplementary Table 4 and Supplementary Table 9. Supplementary Table 10 provides more details about the statistical tests performed. When reporting $p$-values, the nominal $p$-values were reported.

**Control analyses**. Because the multikernel model contained more input data compared with the single-kernel models, we explored the potential effect of the amount of input data on model performance. To this end, we performed a single-kernel ridge regression on a general functional connectivity matrix created by averaging the functional connectivity across all fMRI conditions (rest + MID + N-back + SST) to predict behaviors, which we called Mean FC. We then compared the performance of the Mean FC model with the best single-kernel fMRI model (e.g., N-back only) and the multikernel model. To assess the impact of age and sex on model performance, we performed kernel ridge regression to predict behaviors after regressing out age and sex, in addition to head motion (mean FD and DVARS).

In another set of three analyses, we investigated whether our results are robust to the QC criteria. In the first analysis, we used the same QC thresholds as the original preprocessing, but applied a bandstop filter (0.31–0.43 Hz) to remove respiratory pseudomotion from the motion estimates[149–151], yielding 21.7% more participants ($N = 2262$). In the second analysis, in addition to respiratory pseudomotion filtering, we loosened the motion thresholds, so only frames with

FD > 0.5 and DVARS > 75 were censored. We also loosened the boundary-based registration (BBR) QC criterion, so only runs with BBR cost >0.7 were discarded. The resulting sample size was doubled ($N = 3744$). In the third analysis, we performed stratified subsampling to obtain subsets of participants from the main sample ($N = 1858$) that matched the age, sex, household income, racial composition, and behavioral distributions of the full ABCD sample. This procedure was performed for each behavioral measure separately. Briefly, we divided each characteristic (age, sex, household income, racial composition, and behavioral distribution) into a number of bins. There were 2 bins for sexes, 5 bins for racial groups, and 4 bins for income levels. In the case of age and behavioral measures, the number of bins varied, depending on the behavioral measures. On average, there were 2.14 bins for age (min = 2, max = 3, median = 2) and 3.64 bins for behavioral measures (min = 3, max = 7, median = 3). For example, if there were 2 bins for age and 3 bins for a particular behavioral measure, then there were 2 (sex) × 5 (race) × 4 (income) × 2 (age) × 3 (behavior) = 240 bins. The target percentage of participants in a bin was the percentage of participants from the full sample in the bin. We also set a target number of participants M. For example, suppose M = 1000, target percentage for a bin is 12% and there were 200 (of the 1858) participants in the bin. Then, we randomly selected 12% × 1000 = 120 participants out of the 200 participants. The procedure was repeated for all bins. However, the procedure might fail if M was too big, in which case, the target number of participants for a bin is much greater than the subset of 1858 participants in the bin. In this scenario, we altered the number of bins and/or change the target M. For all three analyses, we compared the predictive-feature matrices with the matrices from the main analysis.

**Reporting summary**. Further information on research design is available in the Nature Research Reporting Summary linked to this article.

## Data availability
The ABCD data are publicly available via the NIMH Data Archive (NDA). Processed data from this study (including the predictive-network features and FC matrices) have been uploaded to the NDA. Researchers with access to the ABCD data will be able to download the data: https://nda.nih.gov/study.html?id=824. Source data are provided with paper. Source data are provided with this paper.

## Code availability
Preprocessing utilized previously published pipelines:[16,143] https://github.com/ThomasYeoLab/CBIG/tree/master/stable_projects/preprocessing/CBIG_fMRI_Preproc2016 Preprocessing code specific to this study can be found here: https://github.com/ThomasYeoLab/ABCD_scripts Analysis code specific to this study was deposited in Zenodo database under accession code 5908961. The code was reviewed by one co-author (L.Q.R.O.) to reduce the chance of coding errors. The software dependencies were Freesurfer (5.3.0), FSL (5.0.8), MATLAB (2018b), and Python (3.6). From time to time, the code might be updated. The most updated version of the code can be found on GitHub (https://github.com/ThomasYeoLab/CBIG/tree/master/stable_projects/predict_phenotypes/ChenTam2022_TRBPC)

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

## Acknowledgements

Our research is currently supported by the Singapore National Research Foundation (NRF) Fellowship (Class of 2017), the NUS Yong Loo Lin School of Medicine (NUHSRO/2020/124/TMR/LOA), the Singapore National Medical Research Council (NMRC) LCG (OFLCG19May-0035), NMRC STaR (STaR20nov-0003), and the USA NIH (R01MH120080). Our computational work was partially performed on resources of the National Supercomputing Centre, Singapore (https://www.nscc.sg). Any opinions, findings and conclusions, or recommendations expressed in this material are those of the authors and do not reflect the views of the Singapore NRF or the Singapore NMRC. Data used in the preparation of this article were obtained from the Adolescent Brain Cognitive Development[SM] (ABCD) Study (https://abcdstudy.org), held in the NIMH Data Archive (NDA). This is a multisite, longitudinal study designed to recruit more than 10,000 children aged 9–10 and follow them over 10 years into early adulthood. The ABCD Study® is supported by the National Institutes of Health and additional federal partners under award numbers U01DA041048, U01DA050989, U01DA051016, U01DA041022, U01DA051018, U01DA051037, U01DA050987, U01DA041174, U01DA041106, U01DA041117, U01DA041028, U01DA041134, U01DA050988, U01DA051039, U01DA041156, U01DA041025, U01DA041120, U01DA051038, U01DA041148, U01DA041093, U01DA041089, U24DA041123, U24DA041147. A full list of supporters is available at https://abcdstudy.org/federal-partners.html. A listing of participating sites and a complete listing of the study investigators can be found at https://abcdstudy.org/consortium_members/. ABCD consortium investigators designed and implemented the study and/or provided data, but did not necessarily participate in the analysis or writing of this report. This paper reflects the views of the authors and may not reflect the opinions or views of the NIH or ABCD consortium investigators. The ABCD data repository grows and changes over time. The ABCD data used in this report came from https://doi.org/10.15154/1504041.

## Author contributions

J.C., A.T., V.K., C.O., C.L.A., S.M., N.U.F.D., S.E., D.B., A.J.H., and B.T.T.Y. designed the research. J.C. and A.T. conducted the research. J.C., A.T., C.O. and B.T.T.Y. analyzed and interpreted the results. J.C., A.T., and B.T.T.Y. wrote the paper and made figures. J.C. and A.T. analyzed the data. J.C., A.T., and L.Q.R.O. reviewed and published the code. All authors provided analytic support. All authors edited the paper.

## Competing interests

The authors declare no competing interests.

## Additional information

**Peer-review information** *Nature Communications* thanks Michael Harms and the other anonymous reviewer(s) for their contribution to the peer review of this work. Peer reviewer reports are available.

