## [Peer Review File · Nature Communications]

REVIEWER COMMENTS

Reviewer #1 (Remarks to the Author):

Thank you for inviting me to review this manuscript by Chen and colleagues, in which the authors analysed a very large dataset (>1800 subjects) to determine whether edge-weights from functional connectivity matrices from either resting state or task fMRI analyses were more predictive of cognitive function and a variety of different mental health-related measures.

Overall, the manuscript was clear, well-written and technically exemplary.

I felt at times that many analyses were presented without a strong explanation of why the analysis was critical for testing the authors' primary hypothesis. For this reason, many of the results took on a somewhat exploratory quality. For instance, there were no strong a priori hypotheses stated about which (if any) edges should be differentially predictive of specific cognitive/mental health signatures (with a particular emphasis on cognitive signatures). This lack of specificity ended up detracting somewhat from my enthusiasm for the major results of the manuscript.

Apologies if I overlooked a detail, but I couldn't find any information about how task regressors were accounted for in the analysis. This could be an important factor in determining the task-based functional connectivity measures, though I do appreciate that it's also important to ensure that measures calculated during rest and task were of equivalent character.

Reviewer #2 (Remarks to the Author):

The manuscript "Shared and unique brain network features predict cognition, personality and mental health in childhood", by Chen et al., seeks to provide a comprehensive evaluation of the extent to which functional connectivity (FC) can predict a variety of traits at the individual level in generally healthy adolescents across 3 broad domains – namely cognitive, impulsivity-related personality, and mental health behavioral measures. It does this using FC derived in both resting and task-states, which additionally allows the manuscript to probe the impact of "brain-state" on the connectivity/behavioral relationships. The data is from the large-scale ABCD Study. Overall, the manuscript is well-written and builds nicely on previous (and rigorous) predictive work by this group in other datasets. The extension of that work to a broad variety of behavioral measures from ABCD is timely and highly relevant to the important goal of understanding the ability of FC to predict individual behavioral differences. As such, this is likely to be an important and impactful manuscript in the rapidly expanding literature on predictive modeling using FC. However, I have a couple of concerns about the manuscript and analysis in its present form, especially (i) the highly selected subset of ABCD participants that made it into the analysis cohort and the potential impact of that selection on the results, and (ii) the impact of computing FC from task data with no attempt to remove the direct effect of BOLD activation from the data.

1. Starting from the full ABCD cohort of 11875 individuals, the final study sample is drastically reduced to just 1858 individuals. The number of individuals excluded for each of the various criteria is nicely (and transparently) laid out in Figure 1, which makes clear that nearly 70% of the excluded individuals were excluded due to not passing fMRI “quality control” (6935 of 10017). Based on the Methods, this exclusion presumably arises from fMRI runs that had over half of their volumes “censored” (removed) due to the chosen thresholds for movement (FD_rms) and DVARS. However, no information is provided about the specific distribution of quality control problems (FD_rms or DVARS), or critically, the extent to which those variables (e.g., high motion) themselves are correlated with the behavioral variables under investigation. I have two particular concerns about this:

A) As a likely influential study using ABCD data, this study has the potential to establish an analysis template in which it is considered appropriate and acceptable to work with just a highly restricted subset of ABCD participants.

B) More importantly, it is very likely that a number of the behavioral variables are themselves correlated with the quality control metrics, and thus the QC/exclusion process may be fundamentally biasing the reported relationships between FC and behavior. It seems likely that with 70% of individuals excluded due to fMRI QC, the resulting subset is no longer reflective of a normal, healthy pre-adolescent population, but rather one that is highly-sampled to remove individuals prone to move in the scanner. The result is likely a much more homogenous sample, which doesn’t fully sample the behavioral variance of traits that happen to be correlated with motion. Evidence of the potential biasing effect of the exclusion criteria is evident in Table S3, which shows that the included individuals are appreciably more White, female, and from a family with higher income than the excluded individuals.

Thus, I think it is important that some additional analyses be performed to get a better handle on the extent to which the fMRI QC exclusion criteria may be biasing the results. One option would be to repeat the analyses using much more liberal exclusion criteria for FD_rms and DVARS, so that only a small subset of subjects with the worst data problems (i.e., consistently very high motion) are excluded. A second option would be to clean the data using a “softer” approach such as ICA-FIX or AROMA that adapts to structured artifacts in the data in a continuous fashion, and thus does not require an arbitrary threshold for data inclusion vs. exclusion.

2. Since it wasn’t mentioned, I assume that a model of the effect of the task on the BOLD activation was not regressed out of the task data prior to computing their FC matrices. In my mind, this results in a serious interpretational problem for “functional connectivity” in task-states (which is present also in some other literature on this topic), as no attempt is made to remove the direct effect of the task itself from the FC estimate. In that regard, it doesn’t seem all that surprising that FC computed during a cognitive task would be more predictive of cognitive behavioral variables. This issue should be addressed explicitly, including at least a discussion of the issue and its potential impact on the results. (Preferably, the task FC results would be computed on residualized task data for comparison; however, if such data is not readily available, a highlighting and discussion of the issue would be sufficient for this particular manuscript).

3. The Abstract and Discussion express “surprise” about the “strong similarity” of the predictive network features within a given behavioral domain. However, even for the cognitive domain, which had the highest average correlation between measures of its predictive-feature matrices, the average correlation was “only” 0.68, which leaves plenty of variance to potentially explain some of the differences within a behavioral domain. As a hypothetical, what if the average correlation for cognition was 0.5 – would that still qualify as strongly similar? Given that there is no generative model or quantitative hypotheses that would make explicit predictions about the degree of similarity, the framing of the results in this fashion seems somewhat arbitrary. As a further example, to my eyes, the predictive-feature matrices in Figure S7 actually differ in appreciable ways across brain-states within a given behavioral domain. Are those matrices “strongly similar”? In some respects they are, and in others, they are clearly not (e.g., within-network SomMot, to give just one example).

4. Similarly, the interpretation of the topography of the predictive network features feels somewhat arbitrary in that certain within- and between-network connections have been highlighted, while others are ignored.

5. Given that the manuscript broadly covers a wide range of behavioral variables, I think it would be beneficial if a portion of the Discussion was dedicated to the issue of what sort of role FC can broadly play in predicting behavior (at least in adolescents), especially for personality and mental health measures, for which the predictive power of FC was rather minimal. What are the implications of these generally small effect sizes for the goal of using FC in personalized medicine?

6. Will the predictive-feature matrices be made publicly available? It is great that they are displayed in the Supplement for visual comparison, but they contain a lot of potentially valuable detailed information that can only be harnessed if the matrices themselves are publicly available.

Minor issues:

7. It would be helpful if the caption for Figure 2 included some indication of what the cognition, personality, and mental health domains represent, since at that point in the manuscript, it isn’t yet clear. i.e., the average correlation of X, Y, Z individual measures for each domain.

8. Given that boxplots can use different algorithms for setting the whisker length and defining outliers, it would be helpful if the details for these particular boxplots were defined somewhere in the manuscript.

9. What was the basis for choosing positive and negative urgency as the behavioral measures for which to display the predictive features matrix in the main text, especially given that personality was rather poorly predicted as a whole? Why not two of the cognitive measures instead (such as crystallized cognition and visuospatial reaction time – i.e., the two cognitive measures with the highest and lowest predictive accuracy)?

10. The text related to Figure 5B implies that the 12 predictive-feature matrices after averaging within each behavioral domain (Figure S7) were correlated both across behavioral domains and brain states. The expected result for Figure 5B would be a 12 x 12 correlation matrix. Instead, Figure 5B just shows 3

separate (4 x 4) correlation matrices for each behavioral domain. The full 12 x 12 matrix would be more informative, and more analogous to Figure 5A.

11. The text indicates that the average correlations between the hypothesis-driven and data-driven predictive-feature matrices were high (p. 15). Were those correlations computed before or after thresholding the matrices for significant blocks? If after, why? (Using the thresholded matrices would likely inflate the correlations, if those are what was used).

12. Several places in the manuscript refer to the importance of inverting the regression models (Haufe et al., 2014) to derive the predictive-feature matrices. It would be helpful if some intuition could be provided as to why the model inversion proposed by Haufe et al. is the “right” thing to do.

13. The scaling/normalization of the predictive-feature matrices would benefit from better explanation. It is unclear in S4 whether each subject’s FC matrix is individually normalized and the resulting values extracted to form $FC_{\{ab\}}$ for each edge, or whether the normalization occurs on $FC_{\{ab\}}$ itself. Relatedly, the equation for $p_{\{ab\}}$ in S4 indicates that $\text{variance}(\hat{y})$ was used as a normalizing factor, which makes sense given that the numerator is a covariance. But this doesn’t seem consistent with the statement in various places that the predictive-feature matrices were divided by their standard deviations for visualization. I suspect these are two different normalizations, but this is confusing and would benefit from clarification. Also, in Figure 7, feature predictability is specified on both a normalized and percentage scale. For the normalized scale (“divided by their standard deviations”), what was the std computed across (the matrix)? As for the percentage scale, no explanation is provided as to what that represents.

14. The manuscript provides the average correlation of the predictive-feature matrices for cognition (0.68), but not for personality and mental health. For completeness, the average for those should be provided as well.

15. The length of the fMRI runs should be provided as part of the manuscript. Given that the MID, SST, and N-Back all involve similar runs lengths, and that Rest involves twice as much data (in a complete subject), it seems likely that the Rest results in Figure 2 would have been worse if using a matched amount of data.

16. It is fairly standard to use a Fisher r-to-z transform prior to averaging correlations, which is not mentioned anywhere in the manuscript, and thus presumably wasn’t done for either averaging the FC matrices across runs, or when averaging the predictive correlations across measures within a behavioral domain (e.g., in Figure 2). Some explanation and justification for the choice to average correlation values directly should be provided.

17. It is unclear to me, are the p-values reported in the manuscript “FDR-corrected p-values” (i.e., the minimum q-level that would still reject the null-hypothesis), or are they the nominal p-value (with the actual p-value threshold that satisfies $q=0.05$ not reported)?

18. Given the possibility of site-specific influences on the behavioral measures, did the permutation

testing shuffle behavioral measures only within site, or did it shuffle randomly across all sites? If the latter, justification for that choice should be provided.

19. Section S4 contains an incomplete sentence: "A positive value (or negative) predictive-feature value for an edge, indicating that higher FC between brain regions and was associated with predicting greater (or lower) behavioral values."

Reviewer #3 (Remarks to the Author):

In this manuscript, Chen and colleagues analyzed data from 1858 children ages 9-10 from the ABCD dataset. The authors examine whole-brain functional connectivity and use these measures to predict individual differences in a host of behavioral measures focused on cognition, personality (impulsivity dimensions), and mental health. The authors contrast prediction from resting and task state data and across different behavioral measures to look for commonalities and differences. Major findings include that (a) there was significant prediction of all three classes of measures, with cognitive measures predicting best, (b) prediction and predictive features were similar for task and rest states, with some evidence that task states (or a combo of task and rest) slightly outperformed rest state predictions for the cognitive measures and some benefit to combining data across states, and (c) predictive features were very similar across behavioral measures within a domain although they differed across domains.

This manuscript adds an important contribution to a growing literature on predicting behavioral measures from fMRI functional connectivity data. The paper was well written and easy to follow. The analyses were logical and thoughtfully implemented. The broad scope of this paper, in terms of both behavioral measures analyzed and whole-brain functional connectivity approach, was impressive and directly resulted in findings that would have difficult to appreciate in papers with a narrower approach, i.e., regarding the general similarity of predictive features across many measures. I have a few questions about methodological controls and the interpretation of these findings, outlined below.

Major:

1. One of the most novel findings of this manuscript is that predictive features were very similar across diverse measures within a behavioral domain. The authors discuss that this may have been related to predicting common factors across cognition or psychopathology (e.g., g or p), but also point out that the neural measures were significantly more correlated than the behavioral measures themselves. How do the authors reconcile/interpret this observation?
2. Moreover, the interpretation of features in prediction is quite fraught (as the authors themselves acknowledge). This may be especially true when features are correlated amongst themselves as they are likely to be in functional connectivity data. I think the arguments made here would be greatly strengthened if the authors included a few stronger tests of the commonalities across feature sets: e.g., could the authors use the model based on one behavioral measure to predict a different behavioral measure within the same domain? Is prediction from these top features sufficient/necessary for prediction for all behaviors within a domain? What about across domains?
3. The authors conclude that task data (or a combination of task and rest data) may be better than rest

data at predicting cognitive measures, but it was not clear to me that these measures were well matched. First, as far as I could tell, task measures still included task activations which may inflate functional connectivity and cross-subject correspondence (e.g., Cole et al., 2019). This may be acceptable from the perspective of clinical applications, but should be accounted for in the interpretation of the data (i.e., the interpretation could be that task activations help to improve predictions, rather than, or in addition to, variations in task functional connectivity). Secondly, the amount of fMRI data that went into any given correlation matrix may vary substantially across different tasks and rest (as well as across subjects). This will be exacerbated in the multi-task comparisons which contains matrices based on ~4x as much data as any single measure. Given that functional connectivity reliability increases systematically with data quantity (e.g., Laumann et al., 2015; Elliott et al., 2019; Noble et al., 2017), these differences may be important and underlie improvements in prediction. If I understood correctly, the authors included a control for the additional features of the multi-kernel FC by averaging the correlation matrices from different states into a single final matrix – but this still does not address the issue that this final matrix is now based on a very different quantity of fMRI data than the matrices from individual tasks. The authors could address this issue by sub-sampling the fMRI data from each state to match the amount of time that is used for functional connectivity analysis and then re-conduct their analyses.

Minor:

4. Related to pt. 3 above, I did not see mention of the length of the task fMRI scans in the methods. It would be helpful to include this information, along with information about how much data was retained (average and range) for participants after frame censoring in each scan type.
5. Many subjects (~80%) were excluded from the current analyses, most due to not passing task or resting fMRI QC. Can the authors say a little more on which specific QC criteria led to exclusion in these datasets? Was there any difference in behavioral measures between included and excluded participants? I was surprised, given the multiband nature of this dataset, that the FD criteria was not filtered to address respiratory artifacts (e.g., Fair et al., 2019; Power et al., 2019).
6. While prediction was significant, it was still fairly low (correlations between true and predicted measures peaked around 0.3 for the cognitive domain, and around 0.1 for personality and mental health domains). I think it would be helpful to discuss this point in the limitation section, as this level of prediction dampens enthusiasm for using these methods in clinical domains (at least at present/with these exact methods). Perhaps the authors could (briefly) acknowledge potential imaging or behavioral factors that may limit prediction.
7. Pg. 4, first paragraph includes a typo: "...distinct cognitive processes such as attention, language or attention..."
8. Typo (maybe?) on pg 15: "Mental health of individual participants was predicted by a distributed set of large-scale network features (Figures 7A & 7B) with default and frontoparietal control networks being particularly prominent (Figures 7C & 7D). For example, greater connectivity between default networks A/B and dorsal attention networks A/B were predictive of larger mental health scores (i.e., worse mental health)." Did the authors mean default and frontoparietal? It looks like both are implicated.

Cole, M. W., Ito, T., Schultz, D., Mill, R., Chen, R., & Cocuzza, C. (2019). Task activations produce spurious but systematic inflation of task functional connectivity estimates. *NeuroImage*, 189, 1-18.

Elliott, M. L., Knodt, A. R., Cooke, M., Kim, M. J., Melzer, T. R., Keenan, R., et al. (2019). General functional connectivity: Shared features of resting-state and task fMRI drive reliable and heritable individual differences in functional brain networks. *NeuroImage*, 189, 516-532.

Fair, D. A., Miranda-Dominguez, O., Snyder, A. Z., Perrone, A., Earl, E. A., Van, A. N., et al. (2020). Correction of respiratory artifacts in MRI head motion estimates. *Neuroimage*, 208, 116400.

Laumann, T. O., Gordon, E. M., Adeyemo, B., Snyder, A. Z., Joo, S. J., Chen, M. Y., et al. (2015). Functional system and areal organization of a highly sampled individual human brain. *Neuron*, 87(3), 657-670.

Noble, S., Spann, M. N., Tokoglu, F., Shen, X., Constable, R. T., & Scheinost, D. (2017). Influences on the test–retest reliability of functional connectivity MRI and its relationship with behavioral utility. *Cerebral Cortex*, 27(11), 5415-5429

Power, J. D., Lynch, C. J., Silver, B. M., Dubin, M. J., Martin, A., & Jones, R. M. (2019). Distinctions among real and apparent respiratory motions in human fMRI data. *NeuroImage*, 201, 116041.

Response to reviewers NCOMMS-20-20465-T “Shared and unique brain network features predict cognition, personality and mental health in childhood”

We are pleased to see a high level of enthusiasm for our work. We thank the reviewers for their close read of this manuscript and insightful comments. Several important suggestions were made for improvement. We have considered each suggestion carefully and revised accordingly. Please find detailed responses (in blue) to the reviewer comments (in italics). For convenience, changes to the manuscript are quoted verbatim (normal font) when appropriate. We believe the manuscript is much improved and hope it is now suitable for publication.

Reviewer #1:

(R1Q1) Thank you for inviting me to review this manuscript by Chen and colleagues, in which the authors analysed a very large dataset (>1800 subjects) to determine whether edge-weights from functional connectivity matrices from either resting state or task fMRI analyses were more predictive of cognitive function and a variety of different mental health-related measures. Overall, the manuscript was clear, well-written and technically exemplary.

We thank the reviewer for the positive comments.

(R1Q2) I felt at times that many analyses were presented without a strong explanation of why the analysis was critical for testing the authors' primary hypothesis. For this reason, many of the results took on a somewhat exploratory quality. For instance, there were no strong a priori hypotheses stated about which (if any) edges should be differentially predictive of specific cognitive/mental health signatures (with a particular emphasis on cognitive signatures). This lack of specificity ended up detracting somewhat from my enthusiasm for the major results of the manuscript.

We apologize for not being clearer about our hypotheses. In this manuscript, we hope to answer two main questions.

Our first question is whether predictive network features are similar across behavioral measures. One possible hypothesis regarding this question is that specialized brain networks are known to support distinct neural processes, so one might expect distinct predictive network features across behavioral measures. A second hypothesis is that studies have shown that multiple networks coordinate to give rise to behavior, so one might also expect similar predictive network features across behavioral measures. Our analyses (Figures 4A & 5A) suggest that predictive network features are similar within behavioral domains, providing support for the second hypothesis. For your convenience, Figure 4A is repeated below. We have made this more explicit in the current manuscript.

pg 3 (Introduction)

Thus, one might hypothesize that distinct network features support prediction of different cognitive traits... Therefore, an alternate hypothesis is that a common set of predictive network features might explain individual differences in cognition, or even across cognition and mental health. To systematically investigate the two hypotheses, we considered the prediction of a variety of behavioral measures.

In the current manuscript, we also evaluated the two competing hypotheses in a more quantitative fashion. More specifically, we have the following observation: when we visually compare two predictive-feature matrices, we look at each network block and check whether the predictive features are both positive (red) or negative (blue). To encode this intuition when comparing a given pair of predictive-feature matrices, we computed the proportion of network blocks for which the predictive network features exhibited consistent directionality (positive or negative) between the pair of predictive-feature matrices. Among all predictive

feature matrices, 49.3% of network blocks were positive while 50.7% of network blocks were negative, so the chance level of the predictive-feature values having the same sign was 50.01% (note that this does not assume the blocks are independent). Statistical significance was tested using the corrected resampled t-test against the chance level of 50.01% (Nadeau and Bengio 2003, Bouckaert and Frank 2004).

We found that within each behavioral domain, the proportion of consistent predictive network features *across behavioral measures* was significantly better than chance, providing support for the second hypothesis that predictive network features are similar within behavioral domains.

We have updated the manuscript as follows.

Figure 4. Predictive network features are similar within hypothesis-driven behavioral domains and across brain states. (A) Correlations of predictive-feature matrices (Figures S3 to S6) across behavioral measures. The predictive-feature matrices were concatenated across brain states and correlated across behavioral measures. If a pair of behavioral measures exhibited a high value (green), then this indicates that the two behavioral measures are predicted by highly similar network features.

pg 11 (Results)

For each pair of behavioral measures, we computed the proportion of network blocks for which the predictive network features exhibited consistent directionality (positive or negative) across the pair of behavioral measures (Figure S7). Among all predictive feature matrices, 49.3% of network blocks were positive while 50.7% of network blocks were negative, so the chance level of the predictive-feature values having the same sign was 50.01%. Within each behavioral domain, the proportion of consistent predictive network features across behavioral measures was significantly greater

than chance: 74% for cognition ($p=6e-45$), 58% for personality ($p=1e-10$) and 67% for mental health ($p=4e-14$). Each within-domain proportion was also significantly greater than the corresponding between-domain proportions ($p < 0.015$). The sole exception was the relatively high between-domain proportion for mental health and personality, consistent with Figure 4A.

pg 30 (Methods)

To compare similarity between a given pair of predictive-feature matrices, we computed the proportion of network blocks for which the predictive network features exhibited consistent directionality (positive or negative) between the pair of predictive-feature matrices. More specifically, for each predictive-feature matrix, predictive-feature values were averaged within and between 18 groups comprising 17 cortical networks and subcortical structures (Figure 1B), yielding an 18 x 18 symmetric matrix. For each of 18 x 18 network blocks and a given pair of predictive-feature matrices, the predictive-feature values were considered consistent if they were both positive or both negative. Among all predictive feature matrices, 49.3% of network blocks were positive while 50.7% of network blocks were negative, so the chance level of the predictive-feature values having the same sign was 50.01%. The proportion of consistent predictive-network blocks were computed for each of 120 cross-validation folds. Statistical significance was tested using the corrected resampled t-test against the chance level of 50.01% (Nadeau and Bengio 2003, Bouckaert and Frank 2004).

Our second question is whether predictive network features are similar across brain states. Again, there are two competing hypotheses. The first hypothesis is that task-FC predicts cognition better than resting-FC (e.g., Greene et al., 2018), so predictive network features are distinct across brain states. The second hypothesis is that functional network architecture is broadly similar during rest and task, so predictive network features are similar across brain states (e.g., Gratton et al., 2018). We have made this more explicit in the current manuscript.

pg 3 (Introduction)

Therefore, one might hypothesize that the functional connections predictive of individual-level cognition (i.e., predictive network features) might differ between rest and task states... Therefore, an alternative hypothesis is that predictive network features are similar across brain states. We seek to investigate the two competing hypotheses in this study.

Our analyses (Figures 4B & 5B) suggest that predictive network features are similar within behavioral domains, providing support for the second hypothesis. For your convenience, Figure 4B is repeated below. In the current manuscript, we also evaluated the two competing hypotheses in a more quantitative fashion by computing the proportion of consistent predictive network features across brain states. We found that within each behavioral domain, the proportion of consistent predictive network across brain states was significantly better than chance, thus providing support for the second hypothesis. We have updated the manuscript as follows.

pg 15 (Results)

For each pair of brain states, we computed the proportion of network blocks for which the predictive network features exhibited consistent directionality (positive or negative) across the pair of brain states. Within each behavioral domain, the proportion of consistent predictive network features across brain states was significantly greater than chance: 63% for cognition ($p=8e-32$), 70% for personality ($p=3e-43$) and 68% for mental health ($p=4e-35$).

(B)

Figure 4. Predictive network features are similar within hypothesis-driven behavioral domains and across brain states. (B) Correlations of predictive-feature matrices across brain states. Predictive-feature matrices were averaged within each behavioral domain and correlated across brain states.

Finally, the reviewer expressed concern that “there were no strong a priori hypotheses stated about which (if any) edges should be differentially predictive of specific cognitive/mental health signatures (with a particular emphasis on cognitive signatures).”

As discussed above, our two primary questions did not involve whether certain specific edges should be predictive of specific behavioral measures. Generating hypotheses about specific edges based on previous studies is not trivial. First, most studies perform association analyses (instead of prediction), so it’s unclear whether their results would inform our current study. This is especially the case for mental health and personality measures. Second, most prediction studies interpret their regression models based on the weights of the regression models or the selected features, which is problematic (Haufe et al., 2014). Indeed, our own results in this study show that the Haufe inversion utilized in this study generates much more consistent predictive network features across regression models, compared with regression weights ($r = 0.99$ versus $r = 0.66$). As such, we are not confident that previous studies could provide strong hypotheses for our study.

Nevertheless, given the reviewer’s request, we have performed a detailed literature search to provide a more detailed account of previous studies and the networks implicated by these studies.

1) In the case of cognition, we considered studies that perform prediction, have large sample sizes (>100), and perform whole-brain FC analysis for all networks. We found 13 studies summarized below. As can be seen, most studies focus on a single cognitive measure. Furthermore, different parcellations are used in the literature, so we have tried our best to interpret their findings using our network nomenclature based on visual inspection (Yeo et al., 2011).

In the following table, DN denotes the default network, CON denotes the control network. LIM denotes the limbic network. SAL denotes the salience/ventral attention network. DAN denotes the dorsal attention network. SMN denotes the somatomotor network. VIS denotes the visual network. SUB denotes subcortical regions.

Behavioral Measure	Study	Imaging Modality	Sample Size	Implicated Networks
-------	------------------	-------------	---------------------

General cognition	Sripada 2019	rest-FC	2013 children	DN, CON, SAL, DAN
	Sripada 2020	rest-FC	5937 children	DN, CON, SAL, DAN, VIS
	Dubois 2018	rest-FC	884 adults	DN, CON, SAL, VIS
	Cai 2019	rest-FC & task-FC	267 mixed	CON, VIS
Fluid Intelligence	Finn 2015	rest-FC	118 adults	DN, SAL, CON, DN
	Greene 2018	rest-FC & task-FC	1086 mixed	SMN, VIS
	Gao 2019	rest-FC & task-FC	1086 mixed	DN, VIS, SAL
	Li 2019	rest-FC	677 adults	DN, CON, SAL, SMN
Picture vocabulary	Jiang 2020	rest-FC & task-FC	463 adults	DN, CON, VIS
Oral reading recognition	Jiang 2020	rest-FC & task-FC	463 adults	DN, CON, SAL, VIS, SUB
Working memory	Yamashita 2018	rest-FC & task-FC	982 mixed	CON, SAL, SMN
	Avery 2020	rest-FC & task-FC	659 adults	DN, CON, SUB, SMN
Cognitive flexibility	Chén 2019	rest-FC	783 adults	DN, CON, SAL, VIS, SUB, SMN
Attention	Rosenberg 2016	rest-FC	138 mixed	DN, CON, SAL, SUB, SMN, VIS

The following table summarizes the percentage of studies suggesting which networks are involved in predicting various cognitive measures.

DN	CON	SAL	DAN	VIS	SMN	LIM	SUB
79%	86%	71%	14%	64%	43%	0%	29%

References

- Avery, E.W., Yoo, K., Rosenberg, M.D., Greene, A.S., Gao, S., Na, D.L., Scheinost, D., Constable, T.R. and Chun, M.M., 2020. Distributed patterns of functional connectivity predict working memory performance in novel healthy and memory-impaired individuals. *Journal of cognitive neuroscience*, 32(2), pp.241-255.
- Cai, B., Zhang, G., Hu, W., Zhang, A., Zille, P., Zhang, Y., Stephen, J.M., Wilson, T.W., Calhoun, V.D. and Wang, Y.P., 2019. Refined measure of functional connectomes for improved identifiability and prediction. *Human brain mapping*, 40(16), pp.4843-4858.
- Chén, O.Y., Cao, H., Reinen, J.M., Qian, T., Gou, J., Phan, H., De Vos, M. and Cannon, T.D., 2019. Resting-state brain information flow predicts cognitive flexibility in humans. *Scientific reports*, 9(1), pp.1-16.
- Dubois, J., Galdi, P., Paul, L.K. and Adolphs, R., 2018. A distributed brain network predicts general intelligence from resting-state human neuroimaging data. *Philosophical Transactions of the Royal Society B: Biological Sciences*, 373(1756), p.20170284.
- Finn, E.S., Shen, X., Scheinost, D., Rosenberg, M.D., Huang, J., Chun, M.M., Papademetris, X. and Constable, R.T., 2015. Functional connectome fingerprinting: identifying individuals using patterns of brain connectivity. *Nature neuroscience*, 18(11), pp.1664-1671.
- Gao, S., Greene, A.S., Constable, R.T. and Scheinost, D., 2019. Combining multiple connectomes improves predictive modeling of phenotypic measures. *Neuroimage*, 201, p.116038.
- Greene, A.S., Gao, S., Scheinost, D. and Constable, R.T., 2018. Task-induced brain state manipulation improves prediction of individual traits. *Nature communications*, 9(1), pp.1-13.
- Jiang, R., Zuo, N., Ford, J.M., Qi, S., Zhi, D., Zhuo, C., Xu, Y., Fu, Z., Bustillo, J., Turner, J.A. and Calhoun, V.D., 2020. Task-induced brain connectivity promotes the detection of individual differences in brain-behavior relationships. *NeuroImage*, 207, p.116370.

Li, M., Wang, D., Ren, J., Langs, G., Stoecklein, S., Brennan, B.P., Lu, J., Chen, H. and Liu, H., 2019. Performing group-level functional image analyses based on homologous functional regions mapped in individuals. *PLoS biology*, 17(3), p.e2007032.

Rosenberg, M.D., Finn, E.S., Scheinost, D., Papademetris, X., Shen, X., Constable, R.T. and Chun, M.M., 2016. A neuromarker of sustained attention from whole-brain functional connectivity. *Nature neuroscience*, 19(1), pp.165-171.

Sripada, C., Rutherford, S., Angstadt, M., Thompson, W.K., Luciana, M., Weigard, A., Hyde, L.H. and Heitzeg, M., 2019. Prediction of neurocognition in youth from resting state fMRI. *Molecular psychiatry*, pp.1-9.

Sripada, C., Angstad, M., Rutherford, S., Taxali, A., Clark, D.A., Greathouse, T., Weigard, A., Hyde, L. and Heitzeg, M., 2020. Brain connectivity patterns in children linked to neurocognitive abilities. *bioRxiv*.

Yamashita, M., Yoshihara, Y., Hashimoto, R., Yahata, N., Ichikawa, N., Sakai, Y., Yamada, T., Matsukawa, N., Okada, G., Tanaka, S.C. and Kasai, K., 2018. A prediction model of working memory across health and psychiatric disease using whole-brain functional connectivity. *Elife*, 7, p.e38844.

2) In the case of mental health, we considered papers that have large samples sizes (>100) and perform whole-brain FC analysis for all networks. We found 20 studies summarized below. As can be seen, the majority of studies focus on a single mental disorder. Furthermore, because relatively few studies performing prediction in mental health domain, the studies were largely association studies with only 5 of the 20 studies performing prediction.

Disorder	Study	Imaging Modality	Sample Size	Implicated Networks
p-factor	Karcher 2021	rest-FC	7581 children	DN
	Lees 2021	rest-FC	9074 children	DN, CON, SAL, DAN, SUB
	Elliott 2018	rest-FC	605 adults	DN, CON, VIS, SMN
	Xia 2018	rest-FC	999 mixed	DN, CON, SAL
	Sripada 2020	rest-FC	5880 children	DN, CON
	Kebets 2019	rest-FC	224 adults	SAL, DAN, SMN, VIS, SUB
Anxiety	Takagi 2018	rest-FC & task-FC	889 adults	DN, CON, SAL, SUB, SMN
Depression	Yan 2019	rest-FC	1642 adults	DN, VIS, SMN, DAN
	Drysdale 2017	rest-FC	1188 adults	SUB, LIM
Autism	Di Martino 2014	rest-FC	763 mixed	DN
	Plitt 2015	rest-FC	296 mixed	DN, CON
	Tang 2020	rest-FC	970 mixed	DN
ADHD	Tomasi 2012	rest-FC	551 children	DN, DAN, SUB
	Sripada 2014	rest-FC	576 mixed	DN, CON, SAL
	Mostert 2016	rest-FC	212 adults	SAL
Schizophrenia / Schizoaffective / Psychotic bipolar	Baker 2014	rest-FC	200 adults	CON
Psychotic	Wang 2018	rest-FC	158	DN, CON, SAL, SMN, LIM, DAN, VIS
Psychotic / affective psychosis /	Baker 2019	rest-FC	1010 adults	DN, CON

affective				
Schizophrenia	Chen 2021	rest-FC	226 adults	DN

The following table summarizes the percentage of studies suggesting which networks are involved in various mental disorders.

DN	CON	SAL	DAN	VIS	SMN	LIM	SUB
79%	53%	37%	26%	21%	26%	11%	26%

References

- Baker, J.T., Holmes, A.J., Masters, G.A., Yeo, B.T., Krienen, F., Buckner, R.L. and Öngür, D., 2014. Disruption of cortical association networks in schizophrenia and psychotic bipolar disorder. *JAMA psychiatry*, 71(2), pp.109-118.
- Baker, J.T., Dillon, D.G., Patrick, L.M., Roffman, J.L., Brady, R.O., Pizzagalli, D.A., Öngür, D. and Holmes, A.J., 2019. Functional connectomics of affective and psychotic pathology. *Proceedings of the National Academy of Sciences*, 116(18), pp.9050-9059.
- Chen, J., Müller, V.I., Dukart, J., Hoffstaedter, F., Baker, J.T., Holmes, A.J., Vatansever, D., Nickl-Jockschat, T., Liu, X., Derntl, B. and Kogler, L., 2021. Intrinsic connectivity patterns of task-defined brain networks allow individual prediction of cognitive symptom dimension of schizophrenia and are linked to molecular architecture. *Biological psychiatry*, 89(3), pp.308-319.
- Di Martino, A., Yan, C.G., Li, Q., Denio, E., Castellanos, F.X., Alaerts, K., Anderson, J.S., Assaf, M., Bookheimer, S.Y., Dapretto, M. and Deen, B., 2014. The autism brain imaging data exchange: towards a large-scale evaluation of the intrinsic brain architecture in autism. *Molecular psychiatry*, 19(6), pp.659-667.
- Drysdale, A.T., Grosenick, L., Downar, J., Dunlop, K., Mansouri, F., Meng, Y., Fetcho, R.N., Zebley, B., Oathes, D.J., Etkin, A. and Schatzberg, A.F., 2017. Resting-state connectivity biomarkers define neurophysiological subtypes of depression. *Nature medicine*, 23(1), pp.28-38.
- Elliott, M.L., Romer, A., Knodt, A.R. and Hariri, A.R., 2018. A connectome-wide functional signature of transdiagnostic risk for mental illness. *Biological psychiatry*, 84(6), pp.452-459.
- Karcher, N.R., Michelini, G., Kotov, R. and Barch, D.M., 2021. Associations between resting-state functional connectivity and a hierarchical dimensional structure of psychopathology in middle childhood. *Biological Psychiatry: Cognitive Neuroscience and Neuroimaging*, 6(5), pp.508-517.
- Kebets, V., Holmes, A.J., Orban, C., Tang, S., Li, J., Sun, N., Kong, R., Poldrack, R.A. and Yeo, B.T., 2019. Somatosensory-motor dysconnectivity spans multiple transdiagnostic dimensions of psychopathology. *Biological psychiatry*, 86(10), pp.779-791.
- Lees, B., Squeglia, L.M., McTeague, L.M., Forbes, M.K., Krueger, R.F., Sunderland, M., Baillie, A.J., Koch, F., Teesson, M. and Mewton, L., 2021. Altered neurocognitive functional connectivity and activation patterns underlie psychopathology in preadolescence. *Biological Psychiatry: Cognitive Neuroscience and Neuroimaging*, 6(4), pp.387-398.
- Mostert, J.C., Shumskaya, E., Mennes, M., Onnink, A.M.H., Hoogman, M., Kan, C.C., Vasquez, A.A., Buitelaar, J., Franke, B. and Norris, D.G., 2016. Characterising resting-state functional connectivity in a large sample of adults with ADHD. *Progress in Neuro-Psychopharmacology and Biological Psychiatry*, 67, pp.82-91.
- Plitt, M., Barnes, K.A. and Martin, A., 2015. Functional connectivity classification of autism identifies highly predictive brain features but falls short of biomarker standards. *NeuroImage: Clinical*, 7, pp.359-366.
- Sripada, C.S., Kessler, D. and Angstadt, M., 2014. Lag in maturation of the brain's intrinsic functional architecture in attention-deficit/hyperactivity disorder. *Proceedings of the National Academy of Sciences*, 111(39), pp.14259-14264.
- Sripada, C., Angstadt, M., Rutherford, S., Taxali, A., Greathouse, T., Clark, D.A., Hyde, L., Weigard, A., Brislin, S., Hicks, B. and Heitzeg, M., 2020. Connectomic Alterations Linked to Transdiagnostic Risk for Psychopathology at the Transition to Adolescence. *BioRxiv*.
- Takagi, Y., Sakai, Y., Abe, Y., Nishida, S., Harrison, B.J., Martínez-Zalacaín, I., Soriano-Mas, C., Narumoto, J. and Tanaka, S.C., 2018. A common brain network among state, trait, and pathological anxiety from whole-brain functional connectivity. *Neuroimage*, 172, pp.506-516.
- Tang, S., Sun, N., Floris, D.L., Zhang, X., Di Martino, A. and Yeo, B.T., 2020. Reconciling dimensional and categorical models of autism heterogeneity: a brain connectomics and behavioral study. *Biological psychiatry*, 87(12), pp.1071-1082.

Tomasi, D. and Volkow, N.D., 2012. Abnormal functional connectivity in children with attention-deficit/hyperactivity disorder. *Biological psychiatry*, 71(5), pp.443-450.

Wang, D., Li, M., Wang, M., Schoeppe, F., Ren, J., Chen, H., Öngür, D., Brady, R.O., Baker, J.T. and Liu, H., 2020. Individual-specific functional connectivity markers track dimensional and categorical features of psychotic illness. *Molecular psychiatry*, 25(9), pp.2119-2129.

Xia, C.H., Ma, Z., Ciric, R., Gu, S., Betzel, R.F., Kaczkurkin, A.N., Calkins, M.E., Cook, P.A., de la Garza, A.G., Vandekar, S.N. and Cui, Z., 2018. Linked dimensions of psychopathology and connectivity in functional brain networks. *Nature communications*, 9(1), pp.1-14.

Yan, C.G., Chen, X., Li, L., Castellanos, F.X., Bai, T.J., Bo, Q.J., Cao, J., Chen, G.M., Chen, N.X., Chen, W. and Cheng, C., 2019. Reduced default mode network functional connectivity in patients with recurrent major depressive disorder. *Proceedings of the National Academy of Sciences*, 116(18), pp.9078-9083.

3) In the case of impulsivity-related personality, most studies focus on seed-based analysis of frontostriatal circuits. We could only find 5 studies (below) performing whole-brain FC analysis for all networks. Of these, only 1 study involved more than 100 participants. Consequently, we did not feel that we can generate reasonable hypotheses from the current literature.

Behavioral measure	Study	Imaging Modality	Sample Size	Implicated Networks
Delay-discounting	Cai 2020	Rest-FC	809 adults	DN, SUB, SMN, DAN, VIS
	Chen 2018	Rest-FC	61 adults	DN, CON, SAL
Barratt Impulsiveness Scale	Herman 2020	Rest-FC	30 adults	SMN
Balloon Analogue Risk Task / Monetary Choice Questionnaire / Iowa Gambling Task	Hobkirk 2019	Rest-FC	72 adults	CON, SAL, SUB
Delay-discounting / UPPS-P Impulsive Scale / Barratt Impulsiveness Scale	Zhu 2017	Rest-FC	51 adults	DN, CON, SAL, SUB

References

Cai, H., Chen, J., Liu, S., Zhu, J. and Yu, Y., 2020. Brain functional connectome-based prediction of individual decision impulsivity. *cortex*, 125, pp.288-298.

Chen, Z., Guo, Y., Suo, T. and Feng, T., 2018. Coupling and segregation of large-scale brain networks predict individual differences in delay discounting. *Biological psychology*, 133, pp.63-71.

Herman, A.M., Critchley, H.D. and Duka, T., 2020. Trait Impulsivity Associated With Altered Resting-State Functional Connectivity Within the Somatomotor Network. *Frontiers in Behavioral Neuroscience*, 14, p.111.

Hobkirk, A.L., Bell, R.P., Utevsky, A.V., Huettel, S. and Meade, C.S., 2019. Reward and executive control network resting-state functional connectivity is associated with impulsivity during reward-based decision making for cocaine users. *Drug and alcohol dependence*, 194, pp.32-39.

Zhu, X., Cortes, C.R., Mathur, K., Tomasi, D. and Momenan, R., 2017. Model-free functional connectivity and impulsivity correlates of alcohol dependence: a resting-state study. *Addiction biology*, 22(1), pp.206-217.

Interestingly, default, control and salience/ventral attention networks are most often reported in studies of cognition and mental health. We have updated the manuscript to reflect this literature search.

FC studies of cognition (Finn et al. 2015, Greene et al. 2018, Sripada et al. 2019) and mental disorders (Xia et al. 2018, Baker et al. 2019, Chen et al. 2021, Tang et al. 2020) have also suggested the importance of default, control and salience/ventral attention networks. Therefore, an alternate hypothesis is that a common set of predictive network features might explain individual differences in cognition, or even across cognition and mental health. To systematically investigate the two hypotheses, we considered the prediction of a variety of behavioral measures.

pg 24 (Discussion)

We found that cognitive performance was predicted by distributed network features with default C, control A, salience/ventral attention A, and somatomotor B networks being particularly prominent. The overrepresentation of default, control and salience/ventral attention networks were consistent with previous studies (Rosenberg et al. 2016, Dubois et al. 2018, Jiang et al. 2020). The prominent role of the somatomotor network was more surprising, although somatomotor regions have been associated with fluid intelligence (Greene et al. 2018), attention (Rosenberg et al. 2016), and general cognitive dysfunction (Kebets et al. 2019).

Mental health measures were predicted by distributed network features with default A/B and control A networks being particularly prominent. Previous studies have linked the default and control networks to multiple psychiatric disorders and symptom profiles (Elliott et al. 2018, Baker et al. 2019, Sripada et al. 2020). Interestingly, while the predictive network features for cognition and mental health both involved the default network, different subnetworks were involved in the two behavioral domains: default network C in cognition and default networks B/C in mental health.

Finally, (impulsivity-related) personality measures were predicted by distributed network features with default A/B and dorsal attention A/B networks being particularly prominent. Most impulsivity studies have typically highlighted fronto-striatal circuits (Jentsch and Taylor 1999, Beck et al. 2009, Buckholtz et al. 2010, Balodis et al. 2012, Hawes et al. 2021). However, very few studies have investigated functional connectivity at the whole-brain level across all networks and these studies only focused on adults (Cai et al. 2020). Therefore, our study complements the literature by providing evidence for the importance of default and attentional networks in predicting impulsivity in children.

(R1Q3) Apologies if I overlooked a detail, but I couldn't find any information about how task regressors were accounted for in the analysis. This could be an important factor in determining the task-based functional connectivity measures, though I do appreciate that it's also important to ensure that measures calculated during rest and task were of equivalent character.

We apologize for not explicitly mentioning this in the original manuscript. Task activations were not regressed from the task fMRI because we felt that it was important to match the processing across all brain states. However, this could potentially affect the interpretation of the results. We have now explicitly mentioned this point in the discussion.

pg 26 (Discussion)

Furthermore, to match processing across resting and task states, task activations were not regressed from the task-state data. Therefore, the prediction improvement of certain task-FC over resting-FC in the cognitive domain might be partially due to task activation (Cole et al. 2019). However, we note that a previous study has suggested that task-induced changes in FC might predict phenotypes independent of task activation (Greene et al. 2020). Future work will benefit from further differentiation of task activation from task-induced FC changes.

Reviewer #2:

(R2Q1) The manuscript “Shared and unique brain network features predict cognition, personality and mental health in childhood”, by Chen et al., seeks to provide a comprehensive evaluation of the extent to which functional connectivity (FC) can predict a variety of traits at the individual level in generally healthy adolescents across 3 broad domains – namely cognitive, impulsivity-related personality, and mental health behavioral measures. It does this using FC derived in both resting and task-states, which additionally allows the manuscript to probe the impact of “brain-state” on the connectivity/behavioral relationships. The data is from the large-scale ABCD Study. Overall, the manuscript is well-written and builds nicely on previous (and rigorous) predictive work by this group in other datasets. The extension of that work to a broad variety of behavioral measures from ABCD is timely and highly relevant to the important goal of understanding the ability of FC to predict individual behavioral differences. As such, this is likely to be an important and impactful manuscript in the rapidly expanding literature on predictive modeling using FC.

We thank the reviewer for the positive comments.

(R2Q2) However, I have a couple of concerns about the manuscript and analysis in its present form, especially (i) the highly selected subset of ABCD participants that made it into the analysis cohort and the potential impact of that selection on the results, and (ii) the impact of computing FC from task data with no attempt to remove the direct effect of BOLD activation from the data. Starting from the full ABCD cohort of 11875 individuals, the final study sample is drastically reduced to just 1858 individuals. The number of individuals excluded for each of the various criteria is nicely (and transparently) laid out in Figure 1, which makes clear that nearly 70% of the excluded individuals were excluded due to not passing fMRI “quality control” (6935 of 10017). Based on the Methods, this exclusion presumably arises from fMRI runs that had over half of their volumes “censored” (removed) due to the chosen thresholds for movement (FD_rms) and DVARS. However, no information is provided about the specific distribution of quality control problems (FD_rms or DVARS), or critically, the extent to which those variables (e.g., high motion) themselves are correlated with the behavioral variables under investigation.

We have now provided the distribution of QC problems and the correlation between QC measures and behavioral scores. Consistent with previous studies (Siegel et al., 2017), there were correlations between QC variables and various behavioral measures. We have performed a control analysis showing that relaxing the QC thresholds still lead to similar predictive network features (see R2Q3).

The manuscript has been updated as follows.

pg 27 (Methods)

Figure S20 shows the distribution of censored frames and runs.

pg 28 (Methods)

Table S3 shows the demographic information of included and excluded participants. Table S4 shows the means and standard deviations of the behavioral measures of included and excluded participants. Tables S5 to S7 show the correlation between QC measures and behavioral scores in the included participants. Consistent with previous studies (Siegel et al., 2017), there were associations between behavioral measures and various QC measures. For example, participants with worse cognition had lower quality data.

Figure S20. Distribution of QC metrics. BBR refers to boundary-based registration. A larger BBR cost indicates worse alignment between anatomical and functional images. Dotted red lines show the QC cut-off. Runs with more than 50% frames censored, BBR cost > 0.6 or max FD > 5mm were removed. Note that the rest runs were from 10,277 participants, while the task runs were from 4,506 participants (Figure 1A).

Table S4. Behavioral distribution of included and excluded participants in ABCD 2.0.1. 32 (marked in red) of the 36 behavioral measures were statistically different ($p < 0.05$, uncorrected) between included and excluded participants.

	Included (1858)	Excluded (10017)
Vocabulary	86.45(7.75)	84.08(8.13)
Attention	95.62(7.98)	93.7(9.31)
Working memory	99.28(10.86)	96.14(12.24)
Executive function	94.84(8.09)	92.08(9.69)

Processing speed	90.80(13.98)	87.54(14.64)
Episodic memory	105.35(11.84)	102.33(12.06)
Reading	92.52(6.53)	90.54(6.93)
Fluid cognition	94.84(9.40)	90.92(10.77)
Crystallized cognition	88.33(6.61)	85.99(7.09)
Overall cognition	89.33(8.02)	85.63(9.22)
Short delay recall	10.22(2.83)	9.55(3.08)
Long delay recall	9.74(2.99)	9.06(3.23)
Fluid intelligence	18.81(3.56)	17.73(3.86)
Visuospatial accuracy	0.63(0.17)	0.58(0.17)
Visuospatial reaction time	2725.04(447.01)	2649.83(474.02)
Visuospatial efficiency	2.4e-4(7.3e-5)	2.3e-4(6.9e-5)
Negative urgency	8.31(2.51)	8.52(2.67)
Lack of planning	7.57(2.18)	7.77(2.41)
Sensation seeking	9.74(2.65)	9.77(2.69)
Positive urgency	7.63(2.77)	8.06(2.99)
Lack perseverance	6.81(2.12)	7.09(2.27)
Behavioral inhibition	9.46(3.62)	9.52(3.78)
Reward responsiveness	10.73(2.89)	11.05(2.92)
Drive	3.69(2.81)	4.22(3.10)
Fun seeking	5.57(2.48)	5.73(2.67)
Anxious depressed	2.51(2.93)	2.52(3.09)
Withdrawn depressed	0.95(1.60)	1.05(1.73)
Somatic complaints	1.51(1.92)	1.49(1.96)
Social problems	1.36(1.97)	1.67(2.33)
Thought problems	1.49(1.98)	1.64(2.23)
Attention problems	2.39(3.11)	3.09(3.55)
Rule-breaking behavior	0.94(1.51)	1.24(1.91)
Aggressive behavior	2.82(3.74)	3.35(4.45)
Total psychosis symptoms	2.08(3.18)	2.74(3.62)

Psychosis severity	4.83(8.91)	6.60(10.87)
Mania	0.99(2.22)	1.36(2.86)

Table S5. Correlation between behavioral scores and mean framewise displacement (FD) across the included participants (n=1858).

	FD (Rest)	FD (MID)	FD (SST)	FD (N-back)
Vocabulary	-0.107	-0.010	-0.097	-0.100
Attention	-0.045	-0.056	-0.046	-0.053
Working memory	-0.080	-0.051	-0.025	-0.071
Executive function	-0.110	-0.069	-0.068	-0.075
Processing speed	-0.137	-0.097	-0.084	-0.084
Episodic memory	-0.089	-0.052	-0.036	-0.105
Reading	-0.103	-0.071	-0.089	-0.079
Fluid cognition	-0.151	-0.105	-0.082	-0.126
Crystallized cognition	-0.120	-0.096	-0.105	-0.101
Overall cognition	-0.164	-0.120	-0.109	-0.136
Short delay recall	-0.042	-0.045	-0.064	-0.066
Long delay recall	-0.055	-0.049	-0.060	-0.079
Fluid intelligence	-0.100	-0.077	-0.086	-0.093
Visuospatial accuracy	-0.121	-0.087	-0.067	-0.090
Visuospatial reaction time	0.025	0	0.016	0.005
Visuospatial efficiency	-0.120	-0.081	-0.068	-0.085
Negative urgency	0.046	0.043	0.028	0.036
Lack of planning	0.027	0.028	0.024	0.033
Sensation seeking	-0.009	0.002	-0.039	0.004
Positive urgency	0.104	0.101	0.069	0.091
Lack perseverance	-0.004	0.007	0.012	0.014
Behavioral inhibition	0.011	-0.008	0	0.047
Reward responsiveness	0.039	0.030	0.018	0.049
Drive	0.064	0.072	0.041	0.085
Fun seeking	0.087	0.081	0.046	0.063
Anxious depressed	-0.013	-0.016	-0.020	-0.007

Withdrawn depressed	-0.017	-0.015	-0.026	-0.005
Somatic complaints	-0.017	-0.040	-0.021	-0.005
Social problems	0.013	0.018	0.012	0.047
Thought problems	-0.004	0.007	-0.011	0.016
Attention problems	0.032	0.006	-0.011	0.042
Rule-breaking behavior	0.032	0.008	0.016	0.041
Aggressive behavior	0.010	-0.010	-0.003	0.017
Total psychosis symptoms	0.068	0.079	0.077	0.095
Psychosis severity	0.065	0.063	0.065	0.081
Mania	0.018	-0.006	-0.012	0.009

Table S6. Correlation between behavioral scores and mean DVARS across the included participants (n=1858).

	DVARS (Rest)	DVARS (MID)	DVARS (SST)	DVARS (N-back)
Vocabulary	-0.077	-0.029	-0.040	-0.063
Attention	-0.0060	0.0010	-0.017	-0.025
Working memory	-0.023	0.0080	0.014	-0.025
Executive function	-0.029	0.019	-0.0070	-0.011
Processing speed	-0.044	0.044	0.014	0.013
Episodic memory	-0.064	-0.022	-0.0080	-0.067
Reading	-0.067	0.0050	-0.029	-0.042
Fluid cognition	-0.057	0.018	0.0020	-0.035
Crystallized cognition	-0.083	-0.016	-0.041	-0.061
Overall cognition	-0.081	0.0040	-0.020	-0.054
Short delay recall	-0.025	-0.0060	-0.028	-0.043
Long delay recall	-0.040	-0.0060	-0.029	-0.052
Fluid intelligence	-0.026	0.011	-0.012	-0.035
Visuospatial accuracy	-0.053	-0.0010	-0.013	-0.033
Visuospatial reaction time	0.0020	-0.0020	0.0080	-0.0090
Visuospatial efficiency	-0.048	-0.0040	-0.016	-0.026
Negative urgency	0.063	0.040	0.050	0.039

Lack of planning	0.0070	-0.012	-0.0020	-0.022
Sensation seeking	-0.018	-0.033	-0.050	-0.043
Positive urgency	0.089	0.079	0.072	0.079
Lack perseverance	0.067	0.059	0.054	0.059
Behavioral inhibition	0.037	0.049	0.030	0.061
Reward responsiveness	0.035	0.036	0.0030	0.023
Drive	0.036	0.039	0.017	0.053
Fun seeking	0.060	0.039	0.011	0.019
Anxious depressed	0.022	0.025	0.026	0.024
Withdrawn depressed	0.041	0.033	0.014	0.043
Somatic complaints	-0.0090	-0.020	-0.016	0.0030
Social problems	0.050	0.028	0.028	0.053
Thought problems	0.053	0.033	0.015	0.031
Attention problems	0.106	0.055	0.049	0.079
Rule-breaking behavior	0.050	0.011	0.012	0.028
Aggressive behavior	0.055	0.016	0.033	0.024
Total psychosis symptoms	0.041	0.0050	0.014	0.036
Psychosis severity	0.034	0.0080	0.012	0.037
Mania	0.053	0.021	0.0070	0.024

Table S7. Correlation between behavioral scores and mean boundary-based registration (BBR) cost across the included participants (n=1858). A larger BBR value indicates worse alignment between the anatomical and functional images.

	BBR (REST)	BBR (MID)	BBR (SST)	BBR (N-back)
Vocabulary	0.033	0.003	-0.003	0.009
Attention	-0.002	-0.014	-0.003	0.017
Working memory	0.015	-0.014	0.006	-0.002
Executive function	-0.004	-0.014	0	0.009
Processing speed	-0.014	-0.022	-0.017	-0.012
Episodic memory	-0.021	-0.033	-0.031	-0.028
Reading	-0.007	-0.041	-0.019	-0.018
Fluid cognition	-0.009	-0.031	-0.017	-0.008

Crystallized cognition	0.016	-0.021	-0.012	-0.004
Overall cognition	0	-0.033	-0.018	-0.010
Short delay recall	0.006	-0.003	0.001	-0.016
Long delay recall	0.019	0.015	0.010	0.001
Fluid intelligence	-0.007	-0.028	-0.025	-0.013
Visuospatial accuracy	-0.047	-0.043	-0.045	-0.046
Visuospatial reaction time	0.002	0.010	0.002	0.009
Visuospatial efficiency	-0.042	-0.044	-0.038	-0.048
Negative urgency	-0.019	-0.010	-0.029	-0.024
Lack of planning	-0.014	-0.007	-0.010	0.004
Sensation seeking	-0.034	-0.040	-0.030	-0.035
Positive urgency	-0.039	-0.002	-0.030	-0.029
Lack perseverance	0.024	0.020	0.001	0.011
Behavioral inhibition	-0.051	-0.062	-0.053	-0.051
Reward responsiveness	-0.044	-0.043	-0.037	-0.023
Drive	-0.026	-0.032	-0.031	-0.023
Fun seeking	-0.021	-0.005	-0.005	-0.012
Anxious depressed	0.015	0.002	0.003	0
Withdrawn depressed	-0.024	-0.015	-0.028	-0.026
Somatic complaints	-0.003	-0.002	0.023	-0.003
Social problems	-0.019	-0.011	-0.017	-0.010
Thought problems	-0.026	-0.010	-0.026	-0.023
Attention problems	0.008	0.020	-0.007	0.007
Rule-breaking behavior	0.002	0.022	-0.012	0.024
Aggressive behavior	0.016	0.017	-0.011	0.020
Total psychosis symptoms	-0.031	-0.007	-0.028	-0.024
Psychosis severity	-0.021	-0.009	-0.029	-0.021
Mania	-0.03	0.003	-0.026	-0.022

(R2Q3) I have two particular concerns about this:

A) As a likely influential study using ABCD data, this study has the potential to establish an analysis template in which it is considered appropriate and acceptable to work with just a highly restricted subset of ABCD participants.

B) More importantly, it is very likely that a number of the behavioral variables are themselves correlated with the quality control metrics, and thus the QC/exclusion process may be fundamentally biasing the reported relationships between FC and behavior. It seems likely that with 70% of individuals excluded due to fMRI QC, the resulting subset is no longer reflective of a normal, healthy pre-adolescent population, but rather one that is highly-sampled to remove individuals prone to move in the scanner. The result is likely a much more homogenous sample, which doesn't fully sample the behavioral variance of traits that happen to be correlated with motion. Evidence of the potential biasing effect of the exclusion criteria is evident in Table S3, which shows that the included individuals are appreciably more White, female, and from a family with higher income than the excluded individuals. Thus, I think it is important that some additional analyses be performed to get a better handle on the extent to which the fMRI QC exclusion criteria may be biasing the results. One option would be to repeat the analyses using much more liberal exclusion criteria for FD_{rms} and DVARS, so that only a small subset of subjects with the worst data problems (i.e., consistently very high motion) are excluded. A second option would be to clean the data using a "softer" approach such as ICA-FIX or AROMA that adapts to structured artifacts in the data in a continuous fashion, and thus does not require an arbitrary threshold for data inclusion vs. exclusion.

We thank the reviewer for this very important point. As a control analysis, we have repeated the analyses with more liberal QC criteria and the predictive network features were similar to the main results. The manuscript has been updated as follows.

pg 20 (Results)

Fourth, many participants were excluded due to image quality issues (Figure 1A). The resulting sample had a higher proportion of female and white participants (Table S3). Image quality issues were also correlated with various behavioral measures (Tables S4 to S7). To increase confidence that our results are applicable to the broader population, we performed two analyses. In the first analysis, we applied a bandstop filter to remove respiratory pseudo-motion, thus retaining 21.7% more participants (N = 2262). In the second analysis, we additionally loosened the image QC criteria, thus retaining double the participants (N = 3744). The predictive-feature matrices remained highly similar to the original matrices: $r = 0.93$ (first analysis; Figure S21) and $r = 0.85$ (second analysis; Figure S22).

pg 31 (Methods)

In another set of analysis, we investigated whether our results are robust to the QC criteria. In the first analysis, we used the same QC thresholds as the original preprocessing, but applied a bandstop filter (0.31Hz to 0.43Hz) to remove respiratory pseudo-motion from the motion estimates (Power et al. 2019, Fair et al. 2020, Gratton et al. 2020), yielding 21.7% more participants (N = 2262). In the second analysis, in addition to respiratory pseudo-motion filtering, we loosened the motion thresholds so only frames with $FD > 0.5$ and $DVARS > 75$ were censored. We also loosened the boundary-based registration (BBR) QC criterion, so only runs with BBR cost > 0.7 were discarded. The resulting sample size was doubled (N = 3744). For both analyses, we compared the predictive-feature matrices with the matrices from the main analysis.

Figure S21. Brain network features that support individual-level prediction of cognition, personality and mental health. This figure is the same as Figure 8 but using data processed with respiratory pseudo-motion filtering ($N = 2262$). (A) Predictive-feature matrices averaged across brain states, considering only within-network and between-network blocks that were significant across all four brain states (Rest, MID, SST, N-Back). (B) Predictive network connections obtained by averaging the matrices in panel (A) within each between-network and within-network block. (C) Positive predictive features obtained by summing positive predictive-feature values across the rows of panel (A). A higher value for a brain region

indicates that stronger connectivity yielded a higher prediction for the behavioral measure. (D) Negative predictive features obtained by summing negative predictive-feature values across the rows of panel (A). A higher value for a brain region indicates that weaker connectivity yielded a greater prediction for the behavioral measure. In both panels C and D, the color of each parcel corresponds to the percentile of predictive-feature values among 400 parcels. For visualization, the values within each predictive-feature matrix in panel A were divided by their standard deviations across all entries in the predictive-feature matrix. Conclusions were highly similar using data from original processing pipeline (Figure 8).

Figure S22. Brain network features that support individual-level prediction of cognition, personality and mental health. This figure is the same as Figure 8 but using data processed with respiratory pseudo-motion filtering and more liberal quality control thresholds (N = 3744). (A) Predictive-feature matrices averaged across brain states, considering only within-network and between-network blocks that were significant across all four brain states (Rest, MID, SST, N-Back). (B) Predictive network connections obtained by averaging the matrices in panel (A) within each between-network and within-network block. (C) Positive predictive features obtained by summing positive predictive-feature values across the rows of panel (A). A higher value for a brain region indicates that stronger connectivity yielded a higher prediction for the behavioral measure. (D) Negative predictive features obtained by summing negative predictive-feature values across the rows of panel (A). A higher value for a brain region indicates that weaker connectivity yielded a greater prediction for the behavioral measure. In both panels C and D, the color of each parcel corresponds to the percentile of predictive-feature values among 400 parcels. For visualization, the values within each predictive-feature matrix in panel A were divided by their standard deviations across all entries in the predictive-feature matrix. Conclusions were highly similar using data from original processing pipeline (Figure 8).

(R2Q4) Since it wasn't mentioned, I assume that a model of the effect of the task on the BOLD activation was not regressed out of the task data prior to computing their FC matrices. In my mind, this results in a serious interpretational problem for "functional connectivity" in task-states (which is present also in some other literature on this topic), as no attempt is made to remove the direct effect of the task itself from the FC estimate. In that regard, it doesn't seem all that surprising that FC computed during a cognitive task would be more predictive of cognitive behavioral variables. This issue should be addressed explicitly, including at least a discussion of the issue and its potential impact on the results. (Preferably, the task FC results would be computed on residualized task data for comparison; however, if such data is not readily available, a highlighting and discussion of the issue would be sufficient for this particular manuscript).

Task activations were not regressed from the task fMRI because we felt that it was important to match the processing across all brain states. However, we agree with the reviewer that this could potentially affect the interpretation of the results. Therefore, we have now explicitly mentioned this point in the discussion.

pg 26 (Discussion)

Furthermore, to match processing across resting and task states, task activations were not regressed from the task-state data. Therefore, the prediction improvement of certain task-FC over resting-FC in the cognitive domain might be partially due to task activation (Cole et al. 2019). However, we note that a previous study has suggested that task-induced changes in FC might predict phenotypes independent of task activation (Greene et al. 2020). Future work will benefit from further differentiation of task activation from task-induced FC changes.

(R2Q5) The Abstract and Discussion express “surprise” about the “strong similarity” of the predictive network features within a given behavioral domain. However, even for the cognitive domain, which had the highest average correlation between measures of its predictive-feature matrices, the average correlation was “only” 0.68, which leaves plenty of variance to potentially explain some of the differences within a behavioral domain. As a hypothetical, what if the average correlation for cognition was 0.5 – would that still qualify as strongly similar? Given that there is no generative model or quantitative hypotheses that would make explicit predictions about the degree of similarity, the framing of the results in this fashion seems somewhat arbitrary. As a further example, to my eyes, the predictive-feature matrices in Figure S7 actually differ in appreciable ways across brain-states within a given behavioral domain. Are those matrices “strongly similar”? In some respects they are, and in others, they are clearly not (e.g., within-network SomMot, to give just one example).

We thank the reviewer for this comment.

First, we have added further quantification of what constituted “similarity”. Intuitively, when we visually compare two predictive-feature matrices, we look at each network block (e.g., within-network SomMot as mentioned by the reviewer) and check whether the predictive features are both positive (red) or negative (blue). To encode this intuition when comparing a given pair of predictive-feature matrices, we computed the proportion of network blocks for which the predictive network features exhibited consistent directionality (positive or negative) between the pair of predictive-feature matrices. Among all predictive feature matrices, 49.3% of network blocks were positive while 50.7% of network blocks were negative, so the chance level of the predictive-feature values having the same sign was 50.01% (note that this does not assume the blocks are independent). Statistical significance was tested using the corrected resampled t-test against the chance level of 50.01% (Nadeau and Bengio 2003, Bouckaert and Frank 2004).

We found that within each behavioral domain, the proportion of consistent predictive network features across behavioral measures was significantly better than chance. Furthermore, within each behavioral domain, the proportion of consistent predictive network across brain states was significantly better than chance. The manuscript has been updated as follows.

pg 11 (Results)

For each pair of behavioral measures, we computed the proportion of network blocks for which the predictive network features exhibited consistent directionality (positive or negative) across the pair of behavioral measures (Figure S7). Among all predictive feature matrices, 49.3% of network blocks were positive while 50.7% of network blocks were negative, so the chance level of the predictive-feature values having the same sign was 50.01%. Within each behavioral domain, the proportion of consistent predictive network features across behavioral measures was significantly greater than chance: 74% for cognition ($p=6e-45$), 58% for personality ($p=1e-10$) and 67% for mental health ($p=4e-14$). Each within-domain proportion was also significantly greater than the corresponding between-domain proportions ($p < 0.015$). The sole exception was the relatively high between-domain proportion for mental health and personality, consistent with Figure 4A.

Figure S7. Proportion of network blocks with consistent predictive network features. For each pair of behavioral measures, we computed the proportion of network blocks for which the predictive network features exhibited consistent directionality (positive or negative) across the two behavioral measures. Green indicates consistency greater than 50%, while brown indicates consistency less than 50%. Within each behavioral domain, the proportion of consistent predictive network features was significantly better than chance: 74% for cognition ($p=6e-45$), 58% for personality ($p=1e-10$) and 67% for mental health ($p=4e-14$). Each within-domain proportion was also significantly greater than the corresponding between-domain proportions ($p < 0.015$). The sole exception was the relatively high between-domain proportion for mental health and personality, consistent with Figure 4A.

pg 15 (Results)

For each pair of brain states, we computed the proportion of network blocks for which the predictive network features exhibited consistent directionality (positive or negative) across the pair of brain states. Within each behavioral domain, the proportion of consistent predictive network features across brain states was significantly greater than chance: 63% for cognition ($p=8e-32$), 70% for personality ($p=3e-43$) and 68% for mental health ($p=4e-35$).

pg 30 (Methods)

To compare similarity between a given pair of predictive-feature matrices, we computed the proportion of network blocks for which the predictive network features exhibited consistent directionality (positive or negative) between the pair of predictive-feature matrices. More specifically, for each predictive-feature matrix, predictive-feature values were averaged within and between 18 groups comprising 17 cortical networks and subcortical structures (Figure 1B), yielding an 18 x 18 symmetric matrix. For each of 18 x 18 network blocks and a given pair of predictive-feature matrices, the predictive-feature values were considered consistent if they were both positive or both negative. Among all predictive feature matrices, 49.3% of network blocks were positive while 50.7% of network blocks were negative, so the chance level of the predictive-feature values having the same sign was 50.01%. The proportion of consistent predictive-network blocks were computed for each of 120 cross-validation folds. Statistical significance was tested using the corrected resampled t-test against the chance level of 50.01% (Nadeau and Bengio 2003, Bouckaert and Frank 2004).

Second, we agree that there are notable differences in predictive network features across behaviors within the same domain and across different brain states. We have highlighted examples of such differences in the updated manuscript.

pg 11 (Results)

The predictive features were similar within each behavioral domain, but there was a number of notable exceptions. For example, within the somatomotor network, resting-state predictive network features were positive for reward responsiveness, but negative for sensation seeking.

pg 15 (Results)

The predictive features were similar across different brain states, but there was a number of notable exceptions. For example, within the somatomotor network, predictive network features were negative for cognition in the resting-state, but positive for cognition in the N-back condition.

Finally, thanks to reviewer 3's suggestion, we have performed further analyses showing that behaviors could be predicted by prediction models of **other** behaviors within the same behavioral domain. Furthermore, behaviors could be predicted by using the top predictive network features of **other** behaviors within the same behavioral domain. For more details, please see our responses R3Q3.

(R2Q6) Similarly, the interpretation of the topography of the predictive network features feels somewhat arbitrary in that certain within- and between-network connections have been highlighted, while others are ignored.

The reviewer is correct that we have previously highlighted certain networks by “eyeballing” Figure 8. We have now averaged the absolute predictive network values from Figures 8C & 8D, allowing us to more objectively highlight networks with the strongest predictive-feature values in the main text.

pg 19 (Results)

Cognitive performance of individual participants was predicted by a distributed set of large-scale network features (Figures 8A & 8B) with default C, control A, somatomotor B and salience/ventral attention A networks being particularly prominent (Figures 8C, 8D & S14).

Personality measures of individual participants were predicted by a distributed set of large-scale network features (Figures 8A & 8B) with default A/B and dorsal attentional A/B networks being particularly prominent (Figures 8C, 8D & S14).

Mental health of individual participants was predicted by a distributed set of large-scale network features (Figures 8A & 8B) with default A/B and control A networks being particularly prominent (Figures 8C, 8D & S14).

Figure S14. Bar plots showing average predictive-network feature values for the hypothesis-driven domains of (A) cognition, (B) personality and (C) personality behaviors. For each

network, average predictive-network feature values were obtained by averaging the absolute predictive-network feature values of all brain regions within the network in Figures 8C and 8D. Color in the bar plots corresponds to network color (Figure 1B).

(R2Q7) Given that the manuscript broadly covers a wide range of behavioral variables, I think it would be beneficial if a portion of the Discussion was dedicated to the issue of what sort of role FC can broadly play in predicting behavior (at least in adolescents), especially for personality and mental health measures, for which the predictive power of FC was rather minimal. What are the implications of these generally small effect sizes for the goal of using FC in personalized medicine?

We thank the reviewer for the suggestion. We have added the following discussion to the manuscript.

pg 26 (Discussion)

Finally, although most behavioral measures were predicted better than chance, the prediction accuracies were low, especially for personality and mental health measures (Marek et al. 2020). Further improvement will be necessary for clinical utility. From the imaging perspective, improved imaging acquisition and modelling, as well as aggregation across larger sample size might improve prediction performance (Schulz et al. 2020, Elliott et al. 2021, Kong et al. 2021). From the behavioral perspective, improvement might be achieved by using composite measures (Dubois et al. 2018) or more innovative digital approaches allowing for greater sampling frequency and thus better reliability (Ho et al. 2020, Kiang et al. 2021, Ong et al. 2021). Lastly, the use of non-imaging features could further enrich our predictive models (Eickhoff and Langner 2019).

(R2Q8) Will the predictive-feature matrices be made publicly available? It is great that they are displayed in the Supplement for visual comparison, but they contain a lot of potentially valuable detailed information that can only be harnessed if the matrices themselves are publicly available.

We thank the reviewer for this excellent suggestion. We have created a NIMH Data Archive (NDA) study and uploaded our processed data there, including the predictive network features and FC matrices. Researchers with access to the ABCD data will be able to download the processed data.

We note that the NDA link (<https://nda.nih.gov/study.html?id=824>) is not currently active. As recommended by NDA, the link will only be public after the manuscript is published, since we will not be able to change the relevant information (e.g. reference to this study) after the link becomes public.

pg 31 (Methods)

The ABCD data are publicly available via the NIMH Data Archive (NDA). Processed data from this study (including the predictive network features and FC matrices) have been uploaded to the NDA. Researchers with access to the ABCD data will be able to download the data: <https://nda.nih.gov/study.html?id=824>.

(R2Q9) It would be helpful if the caption for Figure 2 included some indication of what the cognition, personality, and mental health domains represent, since at that point in the manuscript, it isn't yet clear. i.e., the average correlation of X, Y, Z individual measures for each domain.

Thank you for the suggestion. We have updated the captions.

pg 7 (Figure 2 captions)

The boxplots show the average accuracy within each behavioral domain across 120 replications. The cognition domain comprises measures such as fluid cognition and working memory. The (impulsivity-related) personality domain comprises measures such as sensation seeking and behavioral inhibition. The mental health domain comprises measures such as thought problems and psychosis severity.

(R2Q10) Given that boxplots can use different algorithms for setting the whisker length and defining outliers, it would be helpful if the details for these particular boxplots were defined somewhere in the manuscript.

Thank you for the reminder. Details have been added.

pg 7 (Figure 2 captions)

For each boxplot, the horizontal line indicates the median and the circle indicates the mean. The bottom and top edges of the box indicate the 25th and 75th percentiles respectively. Outliers are defined as data points beyond 1.5 times the interquartile range. The whiskers extend to the most extreme data points not considered outliers.

pg 10 (Figure 3 captions)

Boxplot convention is the same as Figure 2.

(R2Q11) What was the basis for choosing positive and negative urgency as the behavioral measures for which to display the predictive features matrix in the main text, especially given that personality was rather poorly predicted as a whole? Why not two of the cognitive measures instead (such as crystallized cognition and visuospatial reaction time – i.e., the two cognitive measures with the highest and lowest predictive accuracy)?

We appreciate this reviewer's question. Given the large number of predictive-feature matrices, we could not show all of them in the main text. Therefore, we selected two predictive feature matrices as illustrative examples to highlight similarities and differences between the matrices. However, the reviewer is correct that this is arbitrary. In the revised manuscript, we have replaced the figure with Figure S7 (in the original manuscript), which shows the predictive feature matrices averaged across all behavioral measures within a behavioral domain. Furthermore, this figure is moved to later in the manuscript as Figure 6. The full set of predictive feature matrices are found in Figures S3 to S6.

pg 15 (Results)

Given that predictive network features were similar within a behavioral domain, the predictive-feature matrices were averaged across behaviors, yielding a predictive-feature matrix for each behavioral domain and each brain state (Figure 6). The predictive features were similar across different brain states, but there was a number of notable exceptions. For example, within the somatomotor network, predictive network features were negative for cognition in the resting-state, but positive for cognition in the N-back condition.

Figure 6. Predictive-feature matrices for each brain state (Rest, MID, SST, N-Back) averaged across all behavioral measures within each hypothesis-driven behavioral domain (cognition, personality, mental health). For visualization, the values within each matrix were divided by their standard deviations (across all entries in the matrix).

(R2Q12) The text related to Figure 5B implies that the 12 predictive-feature matrices after averaging within each behavioral domain (Figure S7) were correlated both across behavioral domains and brain states. The expected result for Figure 5B would be a 12 x 12 correlation matrix. Instead, Figure 5B just shows 3 separate (4 x 4) correlation matrices for each behavioral domain. The full 12 x 12 matrix would be more informative, and more analogous to Figure 5A.

Thank you for the suggestion. We have updated Figures 4 and 5 to show the full 12 x 12 matrices. Note that Figures 4 and 5 corresponded to Figures 5 and 6 in the original manuscript.

Figure 4. Predictive network features are similar within hypothesis-driven behavioral

domains and across brain states. (A) Correlations of predictive-feature matrices (Figures S3 to S6) across behavioral measures. The predictive-feature matrices were concatenated across brain states and correlated across behavioral measures. If a pair of behavioral measures exhibited a high value (green), then this indicates that the two behavioral measures are predicted by highly similar network features. (B) Correlations of predictive-feature matrices across brain states. Predictive-feature matrices were averaged within each behavioral domain and correlated across brain states. The behavioral measures were ordered and categorized based on ABCD's classification of these measures into cognition, personality and mental health behavioral domains, so we referred to this ordering as "hypothesis-driven". Figure S10 shows the analogue of this figure, but without collapsing across either dimension of brain state or behavior. MID: monetary incentive delay; SST: stop signal task.

Figure 5. Predictive network features are similar within data-driven behavioral domains and across brain states. Both panels (A) and (B) are the same as Figure 4, except that behavioral measures are ordered and categorized based on the data-driven clusters of cognition, personality and mental health. These data-driven clusters were obtained by hierarchical clustering of the predictive-feature matrices (Figures S3 to S6) as indicated by the dendrogram in panel A. Clustering was performed using hierarchical agglomerative average linkage (UPGMA) clustering as implemented in *scipy* 1.2.1 (Virtanen et al. 2020).

Figure S11 shows the analogue of this figure, but without collapsing across either dimension of brain state or behavior. MID: monetary incentive delay; SST: stop signal task.

(R2Q13) The text indicates that the average correlations between the hypothesis-driven and data-driven predictive-feature matrices were high (p. 15). Were those correlations computed before or after thresholding the matrices for significant blocks? If after, why? (Using the thresholded matrices would likely inflate the correlations, if those are what was used).

The correlations were computed before thresholding. We have updated the manuscript to make it clearer.

pg 20 (Results)

Average correlations between the hypothesis-driven and data-driven predictive-feature matrices (before thresholding for significant network blocks) were $r = 0.99$ (cognition), 0.84 (personality) and 0.90 (mental health).

(R2Q14) Several places in the manuscript refer to the importance of inverting the regression models (Haufe et al., 2014) to derive the predictive-feature matrices. It would be helpful if some intuition could be provided as to why the model inversion proposed by Haufe et al. is the “right” thing to do.

We thank the reviewer for this suggestion. We have updated the manuscript as follows:

pg 11 (Results)

Most previous studies have interpreted the regression weights (Sripada et al. 2019, Jiang et al. 2020) or selected features (Finn et al. 2015, Greene et al. 2018) of predictive models, which can be highly misleading (Haufe et al. 2014). For example, suppose we seek to predict the target variable y (e.g., fluid cognition) from the FC of two edges ($FC1$ and $FC2$). In this hypothetical example, let us further assume that $FC1 = y - motion$, and $FC2 = motion$. Then a prediction model with 100% accuracy would be $1 \times FC1 + 1 \times FC2$. The regression weights of this model are both one for $FC1$ and $FC2$. Therefore, if we interpreted the weights of the regression model, we would conclude that both $FC1$ and $FC2$ are strongly related to the target variable y . Haufe’s inversion resolves this issue by computing the covariance between the predicted target variable and the FC of the two edges. Using the Haufe approach, $FC2$ will be assigned a weight of zero, consistent with the intuition that $FC2$ is not related to the target variable even though it is helpful for predicting the target variable. As will be shown in additional control analyses, the predictive features were more robust across regression models with Haufe’s approach, further underlining the importance of this inversion process.

(R2Q15) The scaling/normalization of the predictive-feature matrices would benefit from better explanation. It is unclear in S4 whether each subject’s FC matrix is individually normalized and the resulting values extracted to form $FC_{\{ab\}}$ for each edge, or whether the normalization occurs on $FC_{\{ab\}}$ itself. Relatedly, the equation for $p_{\{ab\}}$ in S4 indicates that $\text{variance}(y_{\text{hat}})$ was used a normalizing factor, which makes sense given that the numerator is a covariance. But this doesn’t seem consistent with the statement in various places that the predictive-feature matrices were divided by their standard deviations for visualization. I suspect these are two different normalizations, but this is confusing and would benefit from clarification. Also, in Figure 7, feature predictability is specified on both a normalized and percentage scale. For the normalized scale (“divided by their standard deviations”), what was the std computed across (the matrix)? As for the percentage scale, no explanation is provided as to what that represents.

We apologize for being unclear. The reviewer is correct that there are two different normalizations. The first normalization is applied to the FC matrices before computing the Haufe inversion (Supplementary Methods S4). The second normalization is applied to the predictive-feature matrices themselves for the purpose of visualization.

We have updated the manuscript to provide the details of the normalizations, so it should hopefully be clear now that the normalizations were performed on different “objects” (FC versus predictive-feature matrices).

pg 5 (Supplementary methods S4)

The FC of each subject was normalized to achieve zero mean and unit norm across all edges of the subject. This normalization arises from our choice of the correlation metric in the kernel ridge regression model. Not performing the normalization results in highly similar predictive network feature matrices (not shown).

pg 22 (Figure 8 captions; original Figure 7 now corresponds to Figure 8)

In both panels C and D, the color of each parcel corresponds to the percentile of predictive-feature values among 400 parcels... For visualization, the values within each predictive-feature matrix in panel A were divided by their standard deviations across all entries in the predictive-feature matrix.

(R2Q16) The manuscript provides the average correlation of the predictive-feature matrices for cognition (0.68), but not for personality and mental health. For completeness, the average for those should be provided as well.

We have now provided these numbers in the text.

pg 15 (Results)

As another example, the average correlations of predictive network features within the cognition, personality and mental health domains were 0.68 ± 0.19 (mean \pm std), 0.21 ± 0.33 and 0.46 ± 0.27 respectively.

(R2Q17) The length of the fMRI runs should be provided as part of the manuscript. Given that the MID, SST, and N-Back all involve similar runs lengths, and that Rest involves twice as much data (in a complete subject), it seems likely that the Rest results in Figure 2 would have been worse if using a matched amount of data.

We fully agree and have now reported the length of fMRI runs. The reviewer is correct that resting-FC had the most amount of timepoints. Therefore, if we matched the amount of data, resting-FC might perform worse. Interestingly, N-back-FC had the least amount of timepoints, but performed the best. We have updated the manuscript as follows.

pg 6 (Results)

SST-FC had worse performance than resting-FC ($p = 0.0082$), but we note that resting-FC had about 50% more timepoints than MID-FC, which could explain the difference. Interestingly, N-back-FC performed the best with the least amount of timepoints.

pg 27 (Methods)

For each participant, there were four resting-fMRI runs. Each resting-fMRI run was 300 secs long. For each participant, there were two runs for each fMRI task. Each MID run was 322.4 secs long. Each N-Back run was 289.6 secs long. Each SST run was 349.6 secs long.

pg 28 (Methods)

Finally, by excluding siblings, the main analysis utilized data from 1858 unrelated children (Figure 1A). For these 1858 children, the length of uncensored fMRI data was 897 ± 232 secs (mean \pm std) for resting-state, 531 ± 107 secs for MID, 579 ± 126 secs for SST, and 482 ± 86 secs for N-back.

(R2Q18) It is fairly standard to use a Fisher r-to-z transform prior to averaging correlations, which is not mentioned anywhere in the manuscript, and thus presumably wasn't done for either averaging the FC matrices across runs, or when averaging the predictive correlations across measures within a behavioral domain (e.g., in Figure 2). Some explanation and justification for the choice to average correlation values directly should be provided.

We used the Fisher r-to-z transformation when averaging FC matrices across runs. However, we did not use Fisher r-to-z transformation when averaging predictive performance (correlations) across behaviors within a domain.

As requested by the reviewer, we have now performed Fisher r-to-z transformation before averaging the predictive performance (correlations). The numbers were very similar to before with relative change less than 2%.

We have updated Figures 2, S16, S17 and S18. Given the figures look almost identical to before, we did not show them in this response to avoid clutter.

pg 29 (Methods)

When averaging prediction accuracies (correlations) across behavioral measures, the correlations underwent Fisher-r-to-z transformation before averaging.

(R2Q19) It is unclear to me, are the p-values reported in the manuscript "FDR-corrected p-values" (i.e., the minimum q-level that would still reject the null-hypothesis), or are they the nominal p-value (with the actual p-value threshold that satisfies $q=0.05$ not reported)?

We apologize for being unclear. Nominal p-values were reported. We have now elaborated on the FDR correction procedure and thresholds in the manuscript.

pg 30 (Methods)

We performed FDR correction on all statistical tests that did not involve predictive feature matrices (i.e., statistical tests for results in Figures 1-3) to determine which behaviors were significantly predicted and would therefore be utilized in subsequent analyses. There was a total of 105 tests and the p value threshold was 0.041. A separate FDR correction was performed on all remaining tests (i.e., statistical tests for results from Figure 4 onwards). There was a total of 4149 tests and the p value threshold was 0.029. When reporting p values, the nominal p values were reported.

(R2Q20) Given the possibility of site-specific influences on the behavioral measures, did the permutation testing shuffle behavioral measures only within site, or did it shuffle randomly across all sites? If the latter, justification for that choice should be provided.

We thank the reviewer for this important note. We previously ignored sites in our permutation testing. We have now repeated all permutation tests by restricting permutation to within the same site. With this new permutation scheme, one mental health behavioral measure (somatic complaints) is no longer significant. Therefore, we have also repeated all subsequent analyses using 31 behavioral measures and updated all the figures. The conclusions of the paper remained unchanged.

Figure 3 (below) has been updated by removing * from somatic complaints. The updated Figures 4 and 5 can be found in our previous response to R2Q12. For your convenience, the updated Figures 8 and S15 are shown below.

(A) Cognition

(B) Personality

(C) Mental Health

Figure 3. Cross-validated prediction performance (Pearson's correlation between observed and predicted values) using multi-kernel ridge regression by exploiting resting-FC, MID-FC, SST-FC and N-back-FC jointly. (A) Cognitive measures. (B) (Impulsivity-related) Personality measures. (C) Mental health measures. * denotes above chance prediction after correcting for multiple comparisons (FDR $q < 0.05$). The boxplots show the average accuracy across 120 replications. Note the different scales across the three panels. The same set of behavioral measures were predicted better than chance when using coefficient of determination (COD) instead of Pearson's correlation as a measure of prediction performance (Figure S2). Boxplot convention is the same as Figure 2.

Figure 8. Brain network features that support individual-level prediction of cognition, personality and mental health. (A) Predictive-feature matrices averaged across brain states, considering only within-network and between-network blocks that were significant across all four brain states (Rest, MID, SST, N-Back). (B) Predictive network connections obtained by averaging the matrices in panel (A) within each between-network and within-network block. (C) Positive predictive features obtained by summing positive predictive-feature values across the rows of panel (A). A higher value for a brain region indicates that stronger connectivity yielded a higher prediction for the behavioral measure. (D) Negative predictive features obtained by summing negative predictive-feature values across the rows of panel (A). A higher value for a brain region indicates that weaker connectivity yielded a greater prediction for the behavioral measure. In both panels C and D, the color of each parcel corresponds to the percentile of predictive-feature values among 400 parcels. See Figure S13A for the subcortical maps. For visualization, the values within each predictive-feature matrix in panel A were divided by their standard deviations across all entries in the predictive-feature matrix. The current figure utilized hypothesis-driven behavioral domains. Conclusions were highly similar using data-driven behavioral clusters (Figures S15 & S16).

Figure S16. Predictive brain network features for predicting cognition, personality and mental health. This figure is the same as Figure 8 but using data-driven behavioral clusters, instead of hypothesis-driven behavioral domains. (A) Predictive-feature matrices averaged across brain states, considering only within-network and between-network blocks that were significant across all four brain states (Rest, MID, SST, N-Back). (B) Predictive network connections obtained by averaging the matrices in panel (A) within each between-network and within-network block. (C) Positive predictive features obtained by summing positive predictive-feature values across the rows of panel (A). A higher value for a brain region indicates that stronger connectivity yielded a higher prediction for the behavioral measure. (D) Negative predictive features obtained by summing negative predictive-feature values across the rows of panel (A). A higher value for a brain region indicates that weaker connectivity yielded a greater prediction for the behavioral measure. Conclusions were highly similar using hypothesis-driven behavioral domains (Figure 8).

(R2Q21) Section S4 contains an incomplete sentence: "A positive value (or negative) predictive-feature value for an edge, indicating that higher FC between brain regions and was associated with predicting greater (or lower) behavioral values."

We have fixed this sentence as follow.

pg 5 (Supplementary methods S4)

A positive value (or negative) predictive-feature value for an edge indicates that higher FC between brain regions a and b is associated with predicting greater (or lower) behavioral values.

Reviewer #3:

(R3Q1) In this manuscript, Chen and colleagues analyzed data from 1858 children ages 9-10 from the ABCD dataset. The authors examine whole-brain functional connectivity and use these measures to predict individual differences in a host of behavioral measures focused on cognition, personality (impulsivity dimensions), and mental health. The authors contrast prediction from resting and task state data and across different behavioral measures to look for commonalities and differences. Major findings include that (a) there was significant prediction of all three classes of measures, with cognitive measures predicting best, (b) prediction and predictive features were similar for task and rest states, with some evidence that task states (or a combo of task and rest) slightly outperformed rest state predictions for the cognitive measures and some benefit to combining data across states, and (c) predictive features were very similar across behavioral measures within a domain although they differed across domains.

This manuscript adds an important contribution to a growing literature on predicting behavioral measures from fMRI functional connectivity data. The paper was well written and easy to follow. The analyses were logical and thoughtfully implemented. The broad scope of this paper, in terms of both behavioral measures analyzed and whole-brain functional connectivity approach, was impressive and directly resulted in findings that would have difficult to appreciate in papers with a narrower approach, i.e., regarding the general similarity of predictive features across many measures. I have a few questions about methodological controls and the interpretation of these findings, outlined below.

We thank the reviewer for the positive comments.

(R3Q2) One of the most novel findings of this manuscript is that predictive features were very similar across diverse measures within a behavioral domain. The authors discuss that this may have been related to predicting common factors across cognition or psychopathology (e.g., g or p), but also point out that the neural measures were significantly more correlated than the behavioral measures themselves. How do the authors reconcile/interpret this observation?

Our original phrasing might have suggested a potential tension between the two points. However, we believe the two points are not necessarily contradictory.

The actual behavioral measures are probably supported by shared and distinct factors, e.g., shared cognitive factors might be related to the g-factor, while shared mental health factors might be related to the p-factor. On the other hand, FC might be more sensitive to certain shared/distinct factors and less sensitive to other shared/distinct factors.

The correlational strengths among behavioral measures depend on the relative weights of common and distinct factors, regardless of whether FC is sensitive to these factors. On the other hand, because we are using FC to predict behavioral measures, the resulting predictive models will only account for factors that FC is sensitive to. Therefore, the correlational strengths among predictive network features depend on the relative weights of common and distinct factors that FC is sensitive to.

Here, we found that the predictive network features were more strongly correlated (within a behavioral domain) than the behavioral measures themselves, suggesting that within the same behavioral domain, the relative contributions of shared FC-sensitive factors were larger than shared non-FC-sensitive factors.

We have modified the text to clarify this point.

Indeed, behavioral measures are thought to be supported by a combination of shared and distinct factors (Mesulam 1990, Miyake et al. 2000, Poldrack 2006, Barrett & Satpute, 2013). However, it is unlikely that FC is equally sensitive to all behavioral factors. In particular, FC might be more sensitive to certain shared and distinct factors, while being insensitive to other shared and distinct factors (Wu et al. 2020). Given that the predictive network features were more strongly correlated (within a behavioral domain) than the behavioral measures themselves, this suggests that the relative contributions of shared FC-sensitive factors were larger than shared non-FC-sensitive factors.

(R3Q3) Moreover, the interpretation of features in prediction is quite fraught (as the authors themselves acknowledge). This may be especially true when features are correlated amongst themselves as they are likely to be in functional connectivity data. I think the arguments made here would be greatly strengthened if the authors included a few stronger tests of the commonalities across feature sets: e.g., could the authors use the model based on one behavioral measure to predict a different behavioral measure within the same domain? Is prediction from these top features sufficient/necessary for prediction for all behaviors within a domain? What about across domains?

We thank the reviewer for the excellent suggestion. We have performed the suggested analyses and found that predictive models and predictive network features successfully generalized to other behavioral measures within the same behavioral domain. Furthermore, within-domain generalizations were better than between-domain generalizations.

The manuscript has been updated as follows.

pg 17 (Results)

Predictive models and predictive network features generalize to other behavioral measures within the same behavioral domain

Given that predictive network features were similar within behavioral domains, we further performed a cross-behavior prediction analysis where each of the 31 significantly predicted behaviors was predicted using the prediction models of other behavioral measures from the same behavioral domain (or different domains). We found that cross-behavior predictions for all behavioral domains were significantly better than chance using models from the same domain ($p < 0.0005$ for all 3 behavioral domains; Figure 7A). Within-domain cognitive and mental health models also predicted better than personality models applied to either cognitive or mental health measures ($p < 0.011$). There was no significant difference between other within-domain and between-domain predictions. Similar conclusions were obtained using data-driven behavioral domains, although we note that this analysis is circular since data-driven behavioral domains were defined based on the predictive-feature matrices (Figure 7B).

In a second analysis, for each behavior, we selected the top predictive features of other behavioral measures from the same behavioral domain (or different domains). The top FC edges were then aggregated and used to predict the behavior. As shown in Figure 7C, we found that top features from the same behavioral domain significantly predicted cognition ($p < 0.0005$), personality ($p = 0.002$), and mental health ($p < 0.0005$). When predicting cognition, within-domain cognitive features performed better than features from personality or mental health models ($p < 2e-6$). There was no significant difference between other within-domain and between-domain predictions. Similar conclusions were obtained using data-driven behavioral domains, although we note that this analysis is circular since data-driven behavioral domains were defined based on the predictive-feature matrices (Figure 7D).

Overall, we found that predictive models and predictive network features generalized to other behavioral measures within the same behavioral domain. Within-domain generalizations were often significantly better than between-domain generalizations.

pg 29 (Methods)

Model generalization across behaviors

We tested whether prediction models could be generalized across behavioral measures. More specifically, for each of the 31 significantly predicted behavioral measures (Figure 3), we averaged the predictions of the multi-kernel FC models from all other behavioral measures from the same behavioral domain (or different behavioral domain). The average prediction was compared with the actual behavioral values to compute the cross-behavior prediction accuracy. We emphasize that the average prediction did not include the target behavioral variable and was performed independently for each of the 120 cross-validation folds.

Feature generalization across behaviors

We tested whether top predictive-network features could be generalized across behavioral measures. More specifically, for each of the 31 significantly predicted behavioral measures, we averaged the predictive feature matrices across all other behavioral measures from the same behavioral domain (or different behavioral domain) and across all brain states. The top 10% of edges with the highest absolute values in the average predictive feature matrices were then selected. For each subject, an aggregate prediction was computed as the average FC of selected edges with positive predictive-feature values minus the average FC of selected edges with negative predictive feature values. The aggregate prediction was compared with the actual behavioral values to compute the cross-behavior prediction accuracy. We emphasize that the selection procedure did not include the target behavioral variable, was performed independently for each of the 120 cross-validation folds and no test data was used for the selection.

Figure 7. Predictive models and predictive network features generalize to other behavioral measures within the same behavioral domain. (A) Cross-behavior predictive performance averaged within each behavioral domain of cognition, personality, and mental health (MH). Each behavioral measure was predicted by *averaging predictive models* of other behavioral measures in the same domain (diagonal cells) or different domains (off-diagonal cells). (B) Same as (A), but using data-driven behavioral domains (Figure 6). (C) Cross-behavior predictive performance averaged within each behavioral domain of cognition, personality, and mental health (MH). Each behavioral measure was predicted by *averaging top predictive network features* of other behavioral measures in the same domain (diagonal cells) or different domains (off-diagonal cells). (D) Same as (C), but using data-driven behavioral domains.

(R3Q4) The authors conclude that task data (or a combination of task and rest data) may be better than rest data at predicting cognitive measures, but it was not clear to me that these measures were well matched. First, as far as I could tell, task measures still included task activations which may inflate functional connectivity and cross-subject correspondence (e.g., Cole et al., 2019). This may be acceptable from the perspective of clinical applications, but should be accounted for in the interpretation of the data (i.e., the interpretation could be that task activations help to improve predictions, rather than, or in addition to, variations in task functional connectivity). Secondly, the amount of fMRI data that went into any given correlation matrix may vary substantially across different tasks and rest (as well as across subjects). This will be exacerbated in the multi-task comparisons which contains matrices based on ~4x as much data as any single measure. Given that functional connectivity reliability increases systematically with data quantity (e.g., Laumann et al., 2015; Elliott et al., 2019; Noble et al., 2017), these differences may be important and underlie improvements in prediction. If I understood correctly, the authors included a control for the additional features of the multi-kernel FC by averaging the correlation matrices from different states into a single final matrix – but this still does not address the issue that this final matrix is now based on a very different quantity of fMRI data than the matrices from individual tasks. The authors could address this issue by sub-sampling the fMRI data from each state to match the amount of time that is used for functional connectivity analysis and then re-conduct their analyses.

Task activations were not regressed from the task fMRI because we felt that it was important to match the processing across all brain states. However, we agree with the reviewer that this could potentially affect the interpretation of the results. Therefore, we have explicitly mentioned this point in the discussion.

pg 26 (Discussion)

Furthermore, to match processing across resting and task states, task activations were not regressed from the task-state data. Therefore, the prediction improvement of certain task-FC over resting-FC in the cognitive domain might be partially due to task activation (Cole et al. 2019). However, we note that a previous study has suggested that task-induced changes in FC might predict phenotypes independent of task activation (Greene et al. 2020). Future work will benefit from further differentiation of task activation from task-induced FC changes.

The reviewer is also correct that the amount of data varied across conditions. In the revised manuscript, we have now explicitly mentioned how this could affect the interpretation of the single-kernel prediction results.

pg 6 (Results)

SST-FC had worse performance than resting-FC ($p = 0.0082$), but we note that resting-FC had about 50% more timepoints than MID-FC, which could explain the difference. Interestingly, N-back-FC performed the best with the least amount of timepoints.

Finally, the reviewer was specifically concerned that multi-kernel prediction utilized more data than single-kernel prediction. Although we also showed that multi-kernel regression performed better than simply averaging FC across task states, the reviewer was concerned that both multi-kernel FC and mean-FC still utilized more data than the single-state regression.

We agree with the reviewer that the additional data probably does play a role in the prediction performance. However, we neglected to mention in the original manuscript that mean-FC was also numerically worse than n-back-FC for predicting cognition. This suggests that more data alone (e.g., mean-FC) did not yield better prediction performance. Instead,

the multi-kernel approach was able to make better use of more data across different brain states to improve behavioral prediction. We have modified the manuscript as follows:

pg 20 (Results)

We found that mean-FC yielded statistically worse prediction performance for cognition compared with multi-kernel regression (Figure S18), but not personality and mental health. Interestingly, mean-FC was also numerically (but not statistically) worse than n-back-FC for predicting cognition ($r = 0.28$ versus $r = 0.29$). Overall, this suggests that although more data probably contributed to better prediction, more data alone did not improve prediction performance. Instead, the multi-kernel approach was able to make better use of more data across different brain states to improve behavioral prediction.

(R3Q5) Related to pt. 3 above, I did not see mention of the length of the task fMRI scans in the methods. It would be helpful to include this information, along with information about how much data was retained (average and range) for participants after frame censoring in each scan type.

We apologize for the omission. We have now provided information about the amount of fMRI data before and after preprocessing.

pg 27 in Methods

For each participant, there were four resting-fMRI runs. Each resting-fMRI run was 300 secs long. For each participant, there were two runs for each fMRI task. Each MID run was 322.4 secs long. Each N-Back run was 289.6 secs long. Each SST run was 349.6 secs long.

pg 28 in Methods

Finally, by excluding siblings, the main analysis utilized data from 1858 unrelated children (Figure 1A). For these 1858 children, the length of uncensored fMRI data was 897 ± 232 secs (mean \pm std) for resting-state, 531 ± 107 secs for MID, 579 ± 126 secs for SST, and 482 ± 86 secs for N-back.

(R3Q6) Many subjects (~80%) were excluded from the current analyses, most due to not passing task or resting fMRI QC. Can the authors say a little more on which specific QC criteria led to exclusion in these datasets? Was there any difference in behavioral measures between included and excluded participants? I was surprised, given the multiband nature of this dataset, that the FD criteria was not filtered to address respiratory artifacts (e.g., Fair et al., 2019; Power et al., 2019). Gratton et al.

The main QC criteria were censoring frames with $FD > 0.3$ mm or $DVARS > 50$, along with one volume before and two volumes after. Uncensored segments of data containing fewer than five contiguous volumes were also censored. Functional runs with over half of their volumes censored and/or max $FD > 5$ mm were removed. We also excluded individuals who did not have at least 4 minutes for each fMRI state. Functional runs with boundary-based registration (BBR) costs greater than 0.6 were excluded. A larger BBR cost indicates poorer alignment between T1 and fMRI data.

We have now provided the distribution of QC problems and the correlation between QC measures and behavioral scores (Figure S20, Tables S4 to S7). Consistent with previous studies (Siegel et al., 2017), there were correlations between QC variables and various behavioral measures. To keep this document from becoming too long, changes to the manuscript together with the new Figure S20 and Tables S4 to S7 can be found in our response to R2Q2.

Furthermore, as suggested by the reviewer, we performed a control analysis to filter respiratory pseudo-motion, yielding 21.7% more participants (N = 2262). In addition, as requested by R2Q3, in a second control analysis, we loosened the QC criteria, yielding double the participants (N = 3744). The predictive-feature matrices remained highly similar to the original matrices: $r = 0.93$ (first analysis; Figure S21) and $r = 0.85$ (second analysis; Figure S22). Figures S21 and S22 are shown below for the reviewer's convenience

Figure S21. Brain network features that support individual-level prediction of cognition, personality and mental health. This figure is the same as Figure 8 but using data processed with respiratory pseudo-motion filtering (N = 2262)...

Figure S22. Brain network features that support individual-level prediction of cognition, personality and mental health. This figure is the same as Figure 8 but using data processed with respiratory pseudo-motion filtering and more liberal quality control thresholds (N = 3744)...

(R3Q7) While prediction was significant, it was still fairly low (correlations between true and predicted measures peaked around 0.3 for the cognitive domain, and around 0.1 for personality and mental health domains). I think it would be helpful to discuss this point in the limitation section, as this level of prediction dampens enthusiasm for using these methods in clinical domains (at least at present/with these exact methods). Perhaps the authors could (briefly) acknowledge potential imaging or behavioral factors that may limit prediction.

We agree with the reviewer. We have added the following discussion to the manuscript.

pg 26 (Discussion)

Finally, although most behavioral measures were predicted better than chance, the prediction accuracies were low, especially for personality and mental health measures (Marek et al. 2020). Further improvement will be necessary for clinical utility. From the imaging perspective, improved imaging acquisition and modelling, as well as aggregation across larger sample size might improve prediction performance (Schulz et al. 2020, Elliott et al. 2021, Kong et al. 2021). From the behavioral perspective, improvement might be achieved by using composite measures (Dubois et al. 2018) or more innovative digital approaches allowing for greater sampling frequency and thus better reliability (Ho et al. 2020, Kiang et al. 2021, Ong et al. 2021). Lastly, the use of non-imaging features could further enrich our predictive models (Eickhoff and Langner 2019).

(R3Q8) Pg. 4, first paragraph includes a typo: "...distinct cognitive processes such as attention, language or attention..."

Thank you. The typo has been fixed.

pg 3 (Introduction)

For example, specialized brain networks support distinct cognitive processes, such as attention, language or episodic memory (Corbetta and Shulman, 2002; Fedorenko and Thompson-Schill 2014; DiNicola et al. 2020).

(R3Q9) Typo (maybe?) on pg 15: "Mental health of individual participants was predicted by a distributed set of large-scale network features (Figures 7A & 7B) with default and frontoparietal control networks being particularly prominent (Figures 7C & 7D). For example, greater connectivity between default networks A/B and dorsal attention networks A/B were predictive of larger mental health scores (i.e., worse mental health)." Did the authors mean default and frontoparietal? It looks like both are implicated.

We thank the reviewer for pointing out the typo. We have fixed it in the revised manuscript.

pg 19 (Results)

For example, greater connectivity between default network B and control network A was predictive of larger mental health scores (i.e., worse mental health).

REVIEWER COMMENTS

Reviewer #1 (Remarks to the Author):

The authors should be commended for their extensive and thorough revision of all of the reviewers comments.

Reviewer #2 (Remarks to the Author):

The manuscript by Chen et al. is appreciably improved, with the authors having thoughtfully considered the reviewers' critiques and suggestions. The manuscript is even more technically sound than it was initially. Unfortunately, however, the revision didn't go far enough in my opinion to address concerns regarding the critical issue of sample selection within the ABCD Study, and its potential impact on the results.

My most fundamental concern from the initial submission remains, which is the highly selected subset of ABCD participants used for the analysis. To attempt to address this, the revised manuscript performs a 'control analysis' in which "more liberal QC criteria" were adopted. This analysis fully doubled number of included participants (from 1858 to 3744). But relative to the number of participants that have resting-fMRI and "Passed Freesurfer recon-all" (n=10277), even the more 'liberal' 3744 sample represents less than 50% of available ABCD participants (from non-Philips scanners), and thus is still highly exclusionary and non-representative of the population that ABCD is intended to sample. Thus, what's shown in this manuscript is not a "large sample of typically developing children", but rather a "large sample of children selected for low-motion, and which are as a consequence more female, from higher income families, considerably less racially diverse, with higher cognitive performance and fewer mental health problems" (which is all, to the manuscript's credit, now explicitly shown in Tables S3 and S4).

Importantly, another recent high-profile paper on the ABCD Task data (Chaarani et al, Nature Neuroscience, 2021), which also used the Release 2.0.1 data, had to grapple with these same issues. Their selection criteria were not nearly as strict, such that 5547-6657 participants were maintained for each of the 3 fMRI tasks (Table 1), although even then there were differences between their included and excluded participants on the aforementioned demographic variables (Table 2).

To me, a truly 'liberal' analysis sample in the current manuscript would have adopted a radically different perspective on the issue of sample selection, and instead of tacking on just another quintile (20%) to the sample, instead included everyone *except* the worst 10-20% on motion. [This would be consistent with my previous suggestion that the analyses be repeated "using a much more liberal exclusion criteria for FD_rms and DVARS, so that only a small subset of subjects with the worst data problems (i.e., consistently very high motion) are excluded"]. Or, at least aim to get in the N=6000 range, similar to what the Chaarani analysis ended up using.

I maintain that such an analysis of behavioral prediction in a truly broad ABCD sample is both necessary

and important, and both the manuscript and scientific community will be better off if it is performed as part of this particular manuscript. If the results from the 'broad' and 'narrow' samples differ considerably (as I suspect they might, given that the jump from the N=1858 to N=3744 sample already decreased the correlation of the predictive feature matrices to $r = 0.85$), then at least the results from a true contrast of 'broad' and 'narrow' samples of ABCD participants will be available. Ideally, the 'broad' analysis would become the basis for the manuscript, and the 'narrow' analysis would become the supplemental supporting material. (Using the narrow N=1858 analysis as the "main" analysis forms a potentially bad precedent for the general use of ABCD data).

With that said, if an analysis of a truly broad sample is not in the cards, then I think there needs to be a dedicated paragraph in the 'Limitations' section of the Discussion highlighting (1) the highly selected nature of the sample, (2) how the 'control' analysis, as implemented, does not fully probe the extent to which the results may change if participants from the "other 50%" of the ABCD sample were included, (3) an explanation of why an analysis using a truly broad sample of ABCD participants was not performed, and (4) caution against other studies duplicating this sample-selection approach if the goal is to make conclusions about a representative sample of developing children.

The following are some more minor items that could be improved or clarified, mostly related to sections/text added as part of the revision (and thus not previously reviewed):

Generalization analysis: Ultimately, I'm not sure how meaningful these particular set of analyses are, given the high within-domain correlation of the predictive-feature matrices. (The basic result here seems readily predicted, even "foreordained", from the predictive-feature correlations). That said, the Results around the generalization findings are framed entirely in terms of p-values, with no mentioning of effect sizes. Figure 7 does include average prediction accuracy, but without some context for what the within-domain *non-transfer* average prediction performance was (on the same color scale), it is difficult to ascertain how much prediction accuracy was lost from the transfer models, which seems relevant to know.

L176-178: Resting-FC having 50% more timepoints than MID-FC doesn't really provide an "explanation" for the difference between RS-FC and MID-FC in prediction performance. Rather it argues against the better performance of MID-FC being due to an increased amount of data. This should be worded more precisely. (Or, were you intending to refer to SST-FC, not MID-FC?)

L486-488: Similar comment applies to the addition here. Namely, if mean-FC based prediction was marginally worse than N-Back-FC, that runs counter to the statement that "more data probably contributed to better prediction" (at least in the context of the specific data included there).

L295-6 and Figure S7: Given that there were only 3 "between-domain" comparisons available, it seems a stretch to describe 1 out of 3 as being a "sole exception".

Figure 2: Just to confirm this isn't a reporting error: Somehow for Cognition, the difference in prediction performance for Rest vs. SST was significantly different, but the (larger) difference between SST and

MID/N-Back was not?

Figure 3 caption: If I understand correctly, these boxplots (of individual behavioral measures) do not represent the “*average* accuracy across 120 replications”, but rather simply the “accuracy across 120 replications”.

Figure 7B and D: If this analysis is “circular”, what is the point of including it?

Figure 8D: The label of the color bar appears to be missing (presumably “negative network feature predictability”).

Tables S1 and S2: It would be helpful if these two tables additionally made clear which measures belonged to which behavioral domain.

Figure S1: As the COD analog to Figure 1, this figure should either include similar statistical comparisons, or at least a note in the caption that statistical comparison was not performed (and thus its absence should not be inferred as a lack of statistical differences).

In the Supplement, it sometimes gets confusing to associate captions with figures, due to the caption appearing on a page with the wrong figure. E.g., p. 32 has the caption for S16, but the figure shown on that page is S17.

L578, L638: In-line reference formatting is not entirely consistent – some references with 3 or more authors list the first two authors (rather than just the first) prior to “et al”.

Some items related to some of the Methods additions:

The run durations listed for each task are not consistent with the actual protocol.

The fact that task activity was not regressed should be mentioned under the “Functional connectivity” section of the Methods as well. Also, whether or not a Fisher’s transform was applied prior to averaging the FC matrices from different runs.

L782: It would be clearer if the amount of available data was framed in terms of data “remaining after censoring”, rather than “length of uncensored data”.

L824-5: It is probably worth mentioning that after averaging, you (presumably) inverted the Fisher transform (z-to-r) so that the reported values in all the figures are back in standard correlation “units”.

L842 (Model generalization across behaviors): The methodological description here seems incomplete. In particular, the text jumps straight to “we averaged the predictions”, without first detailing exactly what was done. For example, were the *predictive-feature* matrices of the other ‘reference’ behaviors first averaged, to generate a model, from which the target behavior could be predicted? Or, were the features of each individual ‘reference’ behavior used to separately predict the ‘target’ behavior, and

then those predicted values averaged across all the other ‘reference’ behaviors?

L850 (Feature generalization across behaviors): If I’m reading this correctly, the FC information was averaged across the selected edges, to yield a single “average FC” per individual. In which case, how was the prediction computed? Was it still run through the kernel regression? If so, how was the similarity term defined?

L897 (Multiple comparison correction): L899 refers to “statistical test for results in Figure 1-3”, but Figure 1 doesn’t involve any statistical tests. It’s unclear what the “105 tests” refers to. Similarly, it’s unclear what constitutes the “4149 tests” for results from Figure 4 onward. In general, while the use of FDR is a positive, it’s hard to assess what specific comparison FDR was applied over.

Reviewer #3 (Remarks to the Author):

The authors have submitted a very impressive revision that addressed all of my comments thoroughly. I am content with the manuscript in its current form and look forward to seeing this addition to the literature.

Response to reviewers NCOMMS-20-20465-A “Shared and unique brain network features predict cognition, personality and mental health in childhood”

We are pleased that reviewers 1 and 3 have accepted our study. Reviewer 2 continued to have reservations about the sample selection. We have considered the comments of Reviewer 2 carefully and have performed an additional control analysis to address Reviewer 2’s concerns. We believe the current manuscript is even better than the previous revision. We hope the manuscript is now suitable for publication.

Please find detailed responses (in blue) to the reviewer comments (in italics). For convenience, changes to the manuscript are quoted verbatim (normal font) when appropriate.

Reviewer #1:

(R1Q1) The authors should be commended for their extensive and thorough revision of all of the reviewers comments.

We thank the reviewer for the positive comments.

Reviewer #2:

(R2Q1) The manuscript by Chen et al. is appreciably improved, with the authors having thoughtfully considered the reviewers’ critiques and suggestions. The manuscript is even more technically sound than it was initially. Unfortunately, however, the revision didn’t go far enough in my opinion to address concerns regarding the critical issue of sample selection within the ABCD Study, and its potential impact on the results.

My most fundamental concern from the initial submission remains, which is the highly selected subset of ABCD participants used for the analysis. To attempt to address this, the revised manuscript performs a ‘control analysis’ in which “more liberal QC criteria” were adopted. This analysis fully doubled number of included participants (from 1858 to 3744). But relative to the number of participants that have resting-fMRI and “Passed Freesurfer recon-all” (n=10277), even the more ‘liberal’ 3744 sample represents less than 50% of available ABCD participants (from non-Philips scanners), and thus is still highly exclusionary and non-representative of the population that ABCD is intended to sample. Thus, what’s shown in this manuscript is not a “large sample of typically developing children”, but rather a “large sample of children selected for low-motion, and which are as a consequence more female, from higher income families, considerably less racially diverse, with higher cognitive performance and fewer mental health problems” (which is all, to the manuscript’s credit, now explicitly shown in Tables S3 and S4).

Importantly, another recent high-profile paper on the ABCD Task data (Chaarani et al, Nature Neuroscience, 2021), which also used the Release 2.0.1 data, had to grapple with these same issues. Their selection criteria were not nearly as strict, such that 5547-6657 participants were maintained for each of the 3 fMRI tasks (Table 1), although even then there were differences between their included and excluded participants on the aforementioned demographic variables (Table 2).

*To me, a truly ‘liberal’ analysis sample in the current manuscript would have adopted a radically different perspective on the issue of sample selection, and instead of tacking on just another quintile (20%) to the sample, instead included everyone *except* the worst 10-20%*

on motion. [This would be consistent with my previous suggestion that the analyses be repeated “using a much more liberal exclusion criteria for FD_rms and DVARS, so that only a small subset of subjects with the worst data problems (i.e., consistently very high motion) are excluded”]. Or, at least aim to get in the N=6000 range, similar to what the Chaarani analysis ended up using.

I maintain that such an analysis of behavioral prediction in a truly broad ABCD sample is both necessary and important, and both the manuscript and scientific community will be better off if it is performed as part of this particular manuscript. If the results from the ‘broad’ and ‘narrow’ samples differ considerably (as I suspect they might, given that the jump from the N=1858 to N=3744 sample already decreased the correlation of the predictive feature matrices to $r = 0.85$), then at least the results from a true contrast of ‘broad’ and ‘narrow’ samples of ABCD participants will be available. Ideally, the ‘broad’ analysis would become the basis for the manuscript, and the ‘narrow’ analysis would become the supplemental supporting material. (Using the narrow N=1858 analysis as the “main” analysis forms a potentially bad precedent for the general use of ABCD data).

With that said, if an analysis of a truly broad sample is not in the cards, then I think there needs to be a dedicated paragraph in the ‘Limitations’ section of the Discussion highlighting (1) the highly selected nature of the sample, (2) how the ‘control’ analysis, as implemented, does not fully probe the extent to which the results may change if participants from the “other 50%” of the ABCD sample were included, (3) an explanation of why an analysis using a truly broad sample of ABCD participants was not performed, and (4) caution against other studies duplicating this sample-selection approach if the goal is to make conclusions about a representative sample of developing children.

We thank the reviewer for the thoughtful perspective on this issue. We agree that the sample utilized in the main analyses (N = 1858) and control analysis (N = 3744) were not fully representative of the full sample. However, pioneering studies have shown that small amount of motion can result in systematic artifacts in FC, which is especially problematic in studies involving children (Power et al., 2012; Satterthwaite et al., 2012). Therefore, we hope the reviewer recognized the difficult tradeoff between data quality and representativeness with potentially no perfect solution for this conundrum.

We agree with the reviewer that loosening the QC thresholds beyond our current control analysis (N = 3744) might lead to even larger difference with the main results. However, this larger difference could be a result of greater motion/imaging artifacts. Therefore, it is hard to conclude whether the difference is due to the more representative sample or greater motion/imaging artifacts or both.

After significant struggling with this conundrum (data quality versus representativeness), we have devised a control analysis that simultaneously addresses issues about data quality and representativeness of the sample. More specifically, we selected subsets of participants from the main sample (N = 1858) that matched the age, sex, income, racial composition, and behavioral distributions of the full ABCD sample. This matching procedure was performed separately for each behavioral measure, yielding an average of 921 participants for each behavioral phenotype. As shown in Table S9 (replicated below for your convenience), there was no statistical difference between the included and excluded participants for the 5 characteristics (age, sex, income, racial composition, and behavioral distributions) for the 36 behavioral measures after correction for multiple comparisons with FDR $q < 0.05$.

By construction, these participants have low motion/imaging artifacts (since participants are a subset of the main sample) and are representative of the entire ABCD sample (due to our matching procedure). We found that the resulting predictive network features were again highly similar ($r = 0.85$) to the main analysis, suggesting the robustness of our results.

We have also carefully considered the reviewer’s suggestion of taking inspiration from the high-profile ABCD study (Chaarani et al., 2021). However, application of Chaarani’s QC criteria to our study will not yield anywhere close to 6000 participants. The reason is that Chaarani was considering each task separately, yielding 5547 participants for the SST task, 6657 participants for the MID task and 6009 participants for the N-back task. In our study, we are using resting-FC and task-FC to jointly predict behavioral measures, so a conjunction of participants across resting and task states is necessary, resulting in much fewer subjects. Table 1 (below) summarizes the number of participants if we used Chaarani’s QC criteria. Here are some observations:

- Note that the table below did not include several Chaarani’s QC criteria related to task activation analysis (e.g., beta weights outlier detection). Therefore, the resulting sample size for each modality (e.g., $N = 6503$ for SST task) is larger than Chaarani’s study. Yet, after conjunction across resting and task states, we are left with 4187 participants, which is only 11% more than our control analysis ($N = 3744$).
- The set of QC measures used in the table does not include typical functional connectivity QC in the literature, so using this sample will incur significant concerns about data quality.
- Differences in results between this enlarged sample ($N = 4187$) and the main sample ($N = 1858$) could be due to motion/imaging artifacts and/or the sample being more representative, so it’s hard to draw any conclusion about the difference.
- The 4187 participants still have a greater proportion of white and female participants than the full ABCD sample, so the diversity issue is still not fully resolved.
- To obtain a sample size close to 6000 would require QC criteria much weaker than Chaarani’s study, which would lead even greater concerns about data quality.

Table 1. Sample selection using Chaarani’s QC criteria, excluding task activation QC criteria (e.g., beta weights outlier detection). Note that the QC criteria below exclude typical functional connectivity QC used in the literature.

	REST	MID	N-Back	SST	Conjunction
Pass T1 recon-all qc	11075				
Not Phillips scanner	9640				
>= 2 Runs passed MRI QC	9217	8281	8141	8147	7495
Mean FD < 0.9 for both runs	8769	7271	6644	6908	5615
Having at least one minimally processed run released by ABCD	8451	6896	6287	6503	5014
No Missing behavior					4664
No Sibling					4187

For the reasons above, we do not believe that it makes sense to relax the QC even further than our previous control analysis ($N = 3744$). Instead, we hope the reviewer is satisfied with our

new control analysis, which in our opinion addresses both issues of data quality and representativeness.

The manuscript has been updated as follows:

Pg 20 (Results)

Fourth, many participants were excluded due to image quality issues (Figure 1A). The resulting sample had a higher proportion of female and white participants (Table S3). Image quality issues were also correlated with various behavioral measures (Tables S4 to S7). To increase confidence that our results were applicable to the broader population, we performed three analyses. In the first analysis, we applied a bandstop filter to remove respiratory pseudo-motion, thus retaining 21.7% more participants ($N = 2262$). In the second analysis, we additionally loosened the image QC criteria, thus retaining double the participants ($N = 3744$). In the third analysis, for each behavioral measure, we selected subsets of participants from the main sample ($N = 1858$) that matched the age, sex, household income, racial composition and behavioral distributions of the full ABCD sample, yielding an average of 921 participants (min = 563, max = 1073) for each behavioral measure (Table S9). There was no statistical difference between the included and excluded participants for the 5 characteristics (age, sex, household income, racial composition, and behavioral distributions) for the 36 behavioral measures after correction for multiple comparisons with FDR $q < 0.05$. By construction, these participants have low motion/imaging artifacts (since participants are a subset of the main sample) and are representative of the entire ABCD sample (due to our matching procedure). The predictive-feature matrices remained highly similar to the original matrices: $r = 0.93$ (first analysis; Figure S21), $r = 0.85$ (second analysis; Figure S22), and $r = 0.85$ (third analysis; Figure S23).

Pg 27 (Discussion)

In our study, we used resting-FC and task-FC to jointly predict behavioral measures. Therefore, a participant was included only if the imaging data of every brain state survived QC. This resulted in a much smaller sample size than if we analyzed each brain state independently (Chaarani et al., 2021). The resulting main sample ($N = 1858$) was also less representative of the full sample in terms of age, sex, racial composition and household income. In a control analysis, we subsampled participants from the main sample to match characteristics of the full sample, thus simultaneously addressing issues about data quality and representativeness of the sample. However, this control analysis further reduced sample sizes. Thus, it remains an open question how to maximize sample sizes, while maintaining data quality and representativeness.

Pg 32 (Methods)

In the third analysis, we performed stratified subsampling to obtain subsets of participants from the main sample ($N = 1858$) that matched the age, sex, household income, racial composition, and behavioral distributions of the full ABCD sample. This procedure was performed for each behavioral measure separately. Briefly, we divided each characteristic (age, sex, household income, racial composition, and behavioral distribution) into K bins. For example, if $K = 3$, then there were $3 \times 3 \times 3 \times 3 \times 3 = 243$ bins. Note that K was not the same for all characteristics, e.g., sex only had 2 bins corresponding to males and females. The target percentage of participants in a bin was the percentage of participants from the full sample in the bin. We also set a target number of participants M . For example, suppose $M = 1000$, target percentage

for a bin is 12% and there were 200 (of the 1858) participants in the bin. Then we randomly selected 12% x 1000 = 120 participants out of the 200 participants. The procedure was repeated for all bins. However, the procedure might fail if M was too big, so the target number of participants for a bin was much greater than the subset of 1858 participants in the bin. In this scenario, we altered the number of bins and/or change the target M.

Table S9A. Distributions of age, sex, and behavioral measures of included and excluded subjects after matching. There is no significant difference in terms of age, sex and behavioral measures between included and excluded subjects after multiple comparisons correction with FDR $q < 0.05$.

Behavior being matched	N matched	Included			Excluded		
		Age	Female	Behavior	Age	Female	Behavior
Vocabulary	924	119.33(7.44)	50.00	85.10(7.91)	118.90(7.46)	47.61	84.40(8.13)
Attention	910	119.15(7.47)	50.44	94.52(8.48)	118.92(7.46)	47.54	93.96(9.19)
Working memory	847	119.30(7.64)	50.65	97.39(11.70)	118.91(7.45)	47.55	96.59(12.11)
Executive function	563	119.45(7.61)	50.09	93.28(8.88)	118.91(7.46)	47.71	92.48(9.54)
Processing speed	998	119.34(7.46)	49.90	89.15(13.89)	118.90(7.46)	47.55	87.95(14.64)
Episodic memory	1002	119.36(7.51)	50.80	103.66(11.74)	118.90(7.46)	47.60	102.74(12.10)
Reading	1014	119.41(7.58)	50.30	91.54(6.73)	118.89(7.45)	47.60	90.79(6.92)
Fluid cognition	687	119.27(7.56)	50.80	92.40(10.57)	118.92(7.46)	47.61	91.50(10.66)
Crystallized cognition	748	119.23(7.66)	49.33	86.97(7.04)	118.92(7.45)	47.60	86.32(7.06)
Overall cognition	856	119.29(7.56)	50.70	87.17(8.30)	118.92(7.45)	47.56	86.15(9.20)
Short delay recall	1008	119.30(7.53)	50.99	9.76(2.83)	118.92(7.46)	47.57	9.65(3.07)
Long delay recall	924	119.40(7.46)	50.00	9.23(3.19)	118.92(7.46)	47.49	9.17(3.20)
Fluid intelligence	1073	119.40(7.50)	51.44	18.20(3.70)	118.90(7.46)	47.50	17.88(3.85)
Visuospatial accuracy	924	119.32(7.51)	50.32	0.60(0.17)	118.90(7.46)	47.50	0.59(0.17)
Visuospatial reaction time	836	119.38(7.42)	50.24	2685.85(477.99)	118.90(7.47)	47.72	2660.13(470.00)
Visuospatial efficiency	1005	119.25(7.42)	50.25	2.3e-4(7.4e-4)	118.91(7.47)	47.41	2.3e-4(6.9e-4)
Negative urgency	926	119.49(7.50)	50.11	8.47(2.56)	118.90(7.46)	47.58	8.49(2.65)
Lack of planning	847	119.40(7.46)	49.82	7.62(2.23)	118.91(7.46)	47.57	7.75(2.39)
Sensation seeking	769	119.44(7.45)	49.93	9.73(2.73)	118.91(7.46)	47.50	9.77(2.68)
Positive urgency	1004	119.40(7.59)	50.50	7.87(2.82)	118.91(7.45)	47.46	8.00(2.97)
Lack perseverance	1011	119.35(7.44)	50.45	7.05(2.21)	118.91(7.46)	47.49	7.04(2.26)
Behavioral inhibition	1002	119.42(7.59)	50.50	9.47(3.63)	118.91(7.45)	47.42	9.51(3.77)
Reward responsiveness	850	119.37(7.48)	50.35	10.95(2.90)	118.92(7.46)	47.48	11.01(2.92)
Drive	995	119.43(7.59)	51.06	3.96(2.93)	118.90(7.45)	47.47	4.16(3.07)
Fun seeking	844	119.43(7.58)	50.12	5.61(2.50)	118.91(7.45)	47.46	5.71(2.65)
Anxious depressed	996	119.33(7.59)	50.20	2.61(3.08)	118.91(7.45)	47.41	2.51(3.06)
Withdrawn depressed	928	119.26(7.54)	50.22	1.12(1.88)	118.92(7.46)	47.49	1.03(1.69)
Somatic complaints	997	119.38(7.56)	50.35	1.54(2.05)	118.90(7.45)	47.45	1.49(1.95)
Social problems	911	119.33(7.39)	49.73	1.59(2.27)	118.91(7.47)	47.44	1.63(2.28)
Thought problems	992	119.33(7.57)	50.30	1.60(2.21)	118.91(7.45)	47.45	1.62(2.19)
Attention problems	921	119.27(7.49)	50.05	2.92(3.58)	118.91(7.46)	47.42	2.98(3.49)
Rule-breaking behavior	924	119.38(7.66)	49.68	1.09(1.62)	118.90(7.44)	47.56	1.20(1.88)
Aggressive behavior	918	119.35(7.53)	50.11	3.17(4.12)	118.91(7.46)	47.47	3.27(4.37)
Total psychosis	996	119.33(7.48)	50.80	2.43(3.51)	118.91(7.46)	47.49	2.65(3.56)

symptoms							
Psychosis severity	1004	119.35(7.50)	50.20	5.81(9.73)	118.91(7.46)	47.51	6.37(10.69)
Mania	997	119.38(7.51)	50.25	1.24(2.65)	118.90(7.46)	47.54	1.31(2.79)

Table S9B. Continuation of Table S9. Distribution of racial composition of included and excluded subjects after matching. There is no significant difference in terms of racial composition between included and excluded subjects after multiple comparisons correction with FDR $q < 0.05$.

	Include					Excluded				
	Asian	Black	Hispanic	White	Other	Asian	Black	Hispanic	White	Other
Vocabulary	3.25	12.34	20.45	52.71	11.26	2.06	15.11	20.28	52.12	10.39
Attention	3.08	12.31	20.22	53.08	11.32	2.08	15.11	20.30	52.03	10.47
Working memory	3.19	13.11	20.54	52.07	11.10	2.08	15.02	20.26	52.12	10.43
Executive function	3.91	14.56	20.25	48.85	12.43	2.07	14.92	20.31	52.30	10.34
Processing speed	3.01	11.62	20.14	54.11	11.12	2.07	15.19	20.31	51.96	10.47
Episodic memory	3.19	11.98	20.16	53.79	10.88	2.06	15.17	20.30	52.02	10.47
Reading	3.35	11.83	19.72	54.04	11.05	2.04	15.19	20.36	52.01	10.39
Fluid cognition	3.78	13.10	21.11	49.93	12.08	2.06	14.97	20.21	52.42	10.35
Crystallized cognition	3.48	13.37	20.45	51.34	11.36	2.06	15.01	20.31	52.05	10.35
Overall cognition	3.50	12.27	20.56	51.87	11.80	2.05	15.07	20.24	52.37	10.36
Short delay recall	3.08	12.00	20.34	53.77	10.81	2.07	15.27	20.13	52.11	10.43
Long delay recall	3.03	20.78	53.03	12.34	10.82	2.09	20.15	52.12	15.34	10.43
Fluid intelligence	3.08	11.56	20.32	53.87	11.18	2.04	15.39	20.18	52.20	10.29
Visuospatial accuracy	3.03	12.45	20.56	52.81	11.15	2.03	15.25	20.26	51.86	10.45
Visuospatial reaction time	3.23	13.04	20.81	51.91	11.00	2.02	15.18	20.25	51.93	10.42
Visuospatial efficiency	2.89	12.54	20.50	53.13	10.95	2.04	15.26	20.27	51.65	10.41
Negative urgency	3.35	12.20	20.52	52.70	11.23	2.03	15.23	20.24	51.98	10.49
Lack of planning	3.19	12.75	20.66	51.59	11.81	2.05	15.16	20.23	52.02	10.45
Sensation seeking	3.25	12.87	21.46	50.98	11.44	2.05	15.14	20.18	51.79	10.52
Positive urgency	3.09	11.85	20.12	53.78	11.16	2.04	15.28	20.28	51.69	10.47
Lack perseverance	2.97	11.97	20.67	53.21	11.18	2.05	15.27	20.23	51.93	10.46
Behavioral inhibition	3.09	12.08	20.06	53.69	11.08	2.03	15.27	20.28	51.91	10.47
Reward responsiveness	3.18	12.71	20.35	52.00	11.76	2.04	15.18	20.25	51.80	10.50
Drive	3.12	12.16	19.60	53.57	11.56	2.03	15.26	20.32	51.84	10.50
Fun seeking	3.32	12.91	20.62	51.90	11.26	2.03	15.16	20.23	51.85	10.51
Anxious depressed	2.71	12.25	20.58	53.51	10.94	2.06	15.27	20.26	51.64	10.44
Withdrawn depressed	3.02	12.50	20.80	52.37	11.31	2.04	15.23	20.25	51.73	10.43
Somatic complaints	2.61	12.14	20.46	53.56	11.23	2.07	15.28	20.27	51.78	10.43
Social problems	2.85	12.18	20.64	52.91	11.42	2.06	15.25	20.26	51.82	10.47
Thought problems	2.62	12.00	20.06	54.13	11.19	2.07	15.29	20.31	51.56	10.45
Attention problems	2.82	12.38	20.41	52.99	11.40	2.06	15.24	20.28	51.72	10.49
Rule-breaking behavior	3.03	12.45	20.67	52.71	11.15	2.04	15.23	20.26	51.73	10.48
Aggressive behavior	2.83	12.42	20.70	52.72	11.33	2.06	15.23	20.25	51.80	10.44
Total psychosis symptoms	2.71	12.15	19.98	54.02	11.14	2.06	15.26	20.32	51.92	10.34
Psychosis severity	2.89	12.35	20.52	53.09	11.16	2.05	15.24	20.27	51.82	10.40
Mania	2.61	12.04	20.66	53.46	11.23	2.08	15.27	20.27	51.83	10.45

Table S9C. Continuation of Table S9. Distribution of family income of included and excluded subjects after matching. There is no significant difference in terms of family income between included and excluded subjects after multiple comparisons correction with FDR $q < 0.05$.

Behaviors being matched	Included				Excluded			
	< 50K	>= 50K & < 100K	>= 100K	Unknown	< 50K	>= 50K & < 100K	>= 100K	Unknown
Vocabulary	26.41	26.62	38.74	8.23	27.10	25.85	38.46	8.59
Attention	25.16	27.14	39.45	8.24	27.20	25.80	38.42	8.58
Working memory	25.97	26.33	39.08	8.62	27.13	25.89	38.48	8.50
Executive function	26.64	25.75	37.30	10.3	27.09	25.91	38.54	8.47
Processing speed	25.05	27.15	39.58	8.22	27.24	25.81	38.37	8.58
Episodic memory	25.75	26.75	39.52	7.98	27.16	25.83	38.40	8.61
Reading	25.74	26.92	38.95	8.38	27.17	25.83	38.44	8.56
Fluid cognition	26.20	26.49	37.55	9.75	27.07	25.89	38.63	8.42
Crystallized cognition	26.87	26.07	38.24	8.82	27.07	25.91	38.50	8.52
Overall cognition	25.93	27.22	37.73	9.11	27.10	25.83	38.61	8.45
Short delay recall	25.79	26.79	38.99	8.43	27.22	25.78	38.49	8.51
Long delay recall	26.19	26.30	38.74	8.77	27.16	25.83	38.54	8.46
Fluid intelligence	25.07	27.12	40.26	7.55	27.34	25.75	38.28	8.62
Visuospatial accuracy	25.97	26.19	39.61	8.23	26.99	25.89	38.51	8.61
Visuospatial reaction time	26.20	26.56	38.64	8.61	26.98	25.86	38.59	8.57
Visuospatial efficiency	25.37	26.57	39.60	8.46	27.07	25.85	38.49	8.58
Negative urgency	26.13	26.57	38.66	8.64	27.20	25.85	38.43	8.52
Lack of planning	26.09	25.97	38.72	9.21	27.20	25.90	38.43	8.47
Sensation seeking	26.66	26.27	37.71	9.36	27.15	25.88	38.50	8.47
Positive urgency	26.29	26.69	39.14	7.87	27.19	25.84	38.38	8.59
Lack perseverance	25.62	27.20	39.07	8.11	27.26	25.79	38.39	8.57
Behavioral inhibition	25.55	26.75	39.32	8.38	27.26	25.83	38.36	8.55
Reward responsiveness	26.59	26.59	38.12	8.71	27.16	25.86	38.47	8.51
Drive	26.13	26.33	39.40	8.14	27.21	25.87	38.36	8.56
Fun seeking	26.42	25.95	38.51	9.12	27.17	25.91	38.44	8.48
Anxious depressed	26.00	26.71	38.76	8.53	27.26	25.81	38.40	8.53
Withdrawn depressed	26.19	26.40	38.58	8.84	27.23	25.84	38.42	8.51
Somatic complaints	26.18	26.48	38.72	8.63	27.24	25.83	38.41	8.52
Social problems	26.02	26.45	38.86	8.67	27.24	25.84	38.40	8.52
Thought problems	26.21	26.31	39.21	8.27	27.24	25.85	38.36	8.56
Attention problems	26.60	26.17	38.98	8.25	27.20	25.86	38.39	8.56
Rule-breaking behavior	26.19	27.06	38.74	8.01	27.23	25.78	38.41	8.58
Aggressive behavior	25.93	26.36	39.32	8.39	27.25	25.84	38.36	8.54
Total psychosis symptoms	25.50	27.01	39.26	8.23	27.30	25.79	38.36	8.54
Psychosis severity	26.20	26.49	39.24	8.07	27.24	25.83	38.37	8.56
Mania	25.98	26.48	39.12	8.43	27.28	25.82	38.37	8.54

Figure S23. Brain network features that support individual-level prediction of cognition, personality and mental health. This figure is the same as Figure 8 but using subjects matched for age, sex, race, family income, and behavior with the overall population (Table S9). (A) Predictive-feature matrices averaged across brain states, considering only within-network and between-network blocks that were significant across all four brain states (Rest, MID, SST, N-Back). (B) Predictive network connections obtained by averaging the matrices in panel (A) within each between-network and within-network block. (C) Positive predictive features obtained by summing positive predictive-feature values across the rows of panel (A). A higher value for a brain region indicates that stronger connectivity yielded a higher prediction for the behavioral measure. (D) Negative predictive features obtained by summing negative predictive-feature values across the rows of panel (A). A higher value for a brain region indicates that weaker connectivity yielded a greater prediction for the behavioral measure. In both panels C and D, the color of each parcel corresponds to the percentile of predictive-feature values among 400 parcels. For visualization, the values within each predictive-feature matrix in panel A were divided by their standard deviations across all entries in the predictive-feature matrix. Conclusions were highly similar using data from original analysis (Figure 8).

*(R2Q2) Generalization analysis: Ultimately, I'm not sure how meaningful these particular set of analyses are, given the high within-domain correlation of the predictive-feature matrices. (The basic result here seems readily predicted, even "foreordained", from the predictive-feature correlations). That said, the Results around the generalization findings are framed entirely in terms of p-values, with no mentioning of effect sizes. Figure 7 does include average prediction accuracy, but without some context for what the within-domain *non-transfer* average prediction performance was (on the same color scale), it is difficult to ascertain how much prediction accuracy was lost from the transfer models, which seems relevant to know.*

The reviewer is correct that the high within-domain correlation of predictive-feature matrices suggests that within-domain generalization should be possible. However, we do expect some performance decrease, so it remains unclear whether the generalization would yield better-than-chance prediction performance. In addition, we note that this analysis was requested by reviewer 3 in the previous round of revision.

As requested by the reviewer, we have added within-domain non-transfer average prediction performance as an additional row in each panel in Figure 7. We have also added the prediction performance values in each cell in Figure 7 to provide context of effect sizes. As expected, generalization performance was lower than the original prediction performance, but the generalization was better than chance.

Figure 7. Predictive models and predictive network features generalize to other behavioral measures within the same behavioral domain. (A) Cross-behavior predictive performance averaged within each behavioral domain of cognition, personality, and mental health (MH) are shown in the first three rows. As a reference, prediction performance using the original multi-kernel FC models (Figure 2) was shown in the fourth row. Each behavioral measure was predicted by *averaging predictive models* of other behavioral measures in the same domain (diagonal cells) or different domains (off-diagonal cells). (B) Same as (A), but using data-driven behavioral domains (Figure 6). (C) Cross-behavior predictive performance averaged within each behavioral domain of cognition, personality, and mental health (MH) are shown in the first three rows. As a reference, prediction performance using the original multi-kernel FC models (Figure 2) was shown in the fourth row. Each behavioral measure was predicted by *averaging top predictive network features* of other behavioral measures in the same domain (diagonal cells) or different domains (off-diagonal cells). (D) Same as (C), but using data-driven behavioral domains.

(R2Q3) L176-178: Resting-FC having 50% more timepoints than MID-FC doesn't really provide an "explanation" for the difference between RS-FC and MID-FC in prediction performance. Rather it argues against the better performance of MID-FC being due to an increased amount of data. This should be worded more precisely. (Or, were you intending to refer to SST-FC, not MID-FC?)

Thank you for spotting the typo. We have updated the manuscript.

Pg 6 (Results)

SST-FC had worse performance than resting-FC ($p = 0.0082$), but we note that resting-FC had about 50% more timepoints than SST-FC, which could explain the difference.

(R2Q4) L486-488: Similar comment applies to the addition here. Namely, if mean-FC based prediction was marginally worse than N-Back-FC, that runs counter to the statement that "more data probably contributed to better prediction" (at least in the context of the specific data included there).

The reviewer is correct. We have improved the phrasing of the sentence.

Pg 20 (Results)

Overall, this suggests that although multi-kernel FC approach might have benefited from more data per subject, more data in itself did not improve prediction performance.

(R2Q5) L295-6 and Figure S7: Given that there were only 3 "between-domain" comparisons available, it seems a stretch to describe 1 out of 3 as being a "sole exception".

The reviewer is correct. We have deleted "sole" from the manuscript.

Pg 11 (Results)

The exception was the relatively high between-domain proportion for mental health and personality, consistent with Figure 4A.

(R2Q6) Figure 2: Just to confirm this isn't a reporting error: Somehow for Cognition, the difference in prediction performance for Rest vs. SST was significantly different, but the (larger) difference between SST and MID/N-Back was not?

We apologize for not being clear. In Figure 2, for the single-kernel results, we only compared rest-FC with task-FC (i.e., rest vs MID, rest vs SST and rest vs N-back). We have modified the figure captions to be clearer. Please also see our response to R2Q20 where we have listed all the performed statistical tests.

Pg 7 (Figure 2 captions)

^ denotes statistically significant difference between approaches after correction for multiple comparisons ($FDR q < 0.05$). Note that we only compared multi-kernel FC with N-back, the best single-kernel regression performer. For the single-kernel results, we only compared resting-state with each of the three task-states.

(R2Q7) Figure 3 caption: If I understand correctly, these boxplots (of individual behavioral measures) do not represent the “*average* accuracy across 120 replications”, but rather simply the “accuracy across 120 replications”.

We thank the reviewer for the careful read. We have corrected the caption.

Pg 10 (Figure 3 caption)

The boxplots show the accuracy across 120 replications.

(R2Q8) Figure 7B and D: If this analysis is “circular”, what is the point of including it?

While Figures 7B and 7D might *overestimate* the domain-specificity of feature/model transferability, we believe Figures 7A and 7C might *underestimate* the potential of the feature/model transferability because these domains were not data-driven. Therefore, we believe that providing both hypothesis-driven and data-driven results make sense.

Pg 17 (Results)

Similar conclusions were obtained using data-driven behavioral domains (Figures 7B & 7D). We note that this analysis might overestimate the domain-specificity of feature/model transferability because the data-driven behavioral domains were defined based on the predictive-feature matrices. However, we note that the hypothesis-driven results might underestimate the potential of feature/model transferability, so both hypothesis-driven and data-driven results were shown for completeness.

(R2Q9) Figure 8D: The label of the color bar appears to be missing (presumably “negative network feature predictability”).

Thank you for catching this omission. We have fixed Figure 8D (shown below for your convenience).

(R2Q10) Tables S1 and S2: It would be helpful if these two tables additionally made clear which measures belonged to which behavioral domain.

We have added the domain information by coloring the measures based on its behavioral domain.

Table S1. Behavioral measures used in this study. Behavioral measures are color coded by behavioral domain (red = cognition, blue = personality, green = mental health).

Scale	Subscale/Measure
NIH Toolbox (Hodes et al. 2013)	Flanker (attention) List sorting working memory (working memory) Dimensional change card sort (executive function) Oral reading recognition (reading) Pattern comparison processing speed (processing speed) Picture sequence memory test (episodic memory) Picture vocabulary test (vocabulary) Cognition fluid composite (fluid cognition) Crystallized composite (crystallized cognition) Cognition total composite (total cognition)
Rey Auditory Verbal Learning Test (RAVLT) (Strauss et al. 2006)	Short delay recall Long delay recall
Little Man Task (Acker and Acker 1982)	Accuracy (visuospatial accuracy) Reaction time - correct responses (visuospatial reaction time) Efficiency (visuospatial efficiency)
Wechsler Intelligence Scale for Children-V (WISC-V) (Wechsler 2014)	Matrix reasoning (fluid intelligence)
Modified UPPS-P for Children from PhenX (Lynam 2013)	Negative urgency Positive urgency Lack of planning Lack of perseverance Sensation seeking
Behavioral Inhibition & Activation (Pagliaccio et al. 2016)	Behavioral inhibition sum Reward responsiveness Drive Fun seeking
Achenbach Child Behavior Check List (Achenbach and Rescorla 2013)	Anxious/Depressed Withdrawn/Depressed Somatic complaints Social problems Thought problems Attention problems Rule-breaking behavior Aggressive behavior
Parent General Behavior Inventory (Youngstrom et al. 2013)	Mania
Pediatric Psychosis Questionnaire - Brief Version (Loewy et al. 2012)	Total number of psychosis symptoms Symptom severity score

Table S2. Lookup table showing the original ABCD variable names with the corresponding descriptive labels used in the manuscript. More details of the behavioral measures can be found in the ABCD data dictionary. Behavioral measures are color coded by behavioral domain (red = cognition, blue = personality, green = mental health).

Description	ABCD field	ABCD file
Vocabulary	nihtbx_picvocab_uncorrected	abcd_tbss01.txt
Attention	nihtbx_flanker_uncorrected	abcd_tbss01.txt
Working memory	nihtbx_list_uncorrected	abcd_tbss01.txt
Executive function	nihtbx_cardsort_uncorrected	abcd_tbss01.txt
Processing speed	nihtbx_pattern_uncorrected	abcd_tbss01.txt
Episodic memory	nihtbx_picture_uncorrected	abcd_tbss01.txt
Reading	nihtbx_reading_uncorrected	abcd_tbss01.txt
Fluid cognition	nihtbx_fluidcomp_uncorrected	abcd_tbss01.txt
Crystallized cognition	nihtbx_cryst_uncorrected	abcd_tbss01.txt
Overall cognition	nihtbx_totalcomp_uncorrected	abcd_tbss01.txt
Short delay recall	pea_ravlt_sd_trial_vi_tc	abcd_ps01.txt
Long delay recall	pea_ravlt_ld_trial_vii_tc	abcd_ps01.txt
Fluid intelligence	pea_wiscv_trs	abcd_ps01.txt
Visuospatial accuracy	lmt_scr_perc_correct	lmt201.txt
Visuospatial reaction time	lmt_scr_rt_correct	lmt201.txt
Visuospatial efficiency	lmt_scr_efficiency	lmt201.txt
Negative urgency	upps_y_ss_negative_urgency	abcd_mhy02.txt
Lack of planning	upps_y_ss_lack_of_planning	abcd_mhy02.txt
Sensation seeking	upps_y_ss_sensation_seeking	abcd_mhy02.txt
Positive urgency	upps_y_ss_positive_urgency	abcd_mhy02.txt
Lack perseverance	upps_y_lack_of_perseverance	abcd_mhy02.txt
Behavioral inhibition	bis_y_ss_bis_sum	abcd_mhy02.txt
Reward responsiveness	bis_y_ss_bas_rr	abcd_mhy02.txt
Drive	bis_y_ss_bas_drive	abcd_mhy02.txt
Fun seeking	bis_y_ss_bas_fs	abcd_mhy02.txt

Anxious depressed	cbcl_scr_syn_anxdep_r	abcd_cbcls01.txt
Withdrawn depressed	cbcl_scr_syn_withdep_r	abcd_cbcls01.txt
Somatic complaints	cbcl_scr_syn_somatic_r	abcd_cbcls01.txt
Social problems	cbcl_scr_syn_social_r	abcd_cbcls01.txt
Thought problems	cbcl_scr_syn_thought_r	abcd_cbcls01.txt
Attention problems	cbcl_scr_syn_attention_r	abcd_cbcls01.txt
Rule-breaking behavior	cbcl_scr_syn_rulebreak_r	abcd_cbcls01.txt
Aggressive behavior	cbcl_scr_syn_aggressive_r	abcd_cbcls01.txt
Total psychosis symptoms	pps_y_ss_number	abcd_mhy02.txt
Psychosis severity	pps_y_ss_severity_score	abcd_mhy02.txt
Mania	pgbi_p_ss_score	abcd_mhp02.txt

(R2Q11) Figure S1: As the COD analog to Figure 1, this figure should either include similar statistical comparisons, or at least a note in the caption that statistical comparison was not performed (and thus its absence should not be inferred as a lack of statistical differences).

We apologize for being unclear. We did not perform any statistical test for Figure S1. We had now added this information in the caption.

Figure S1 caption:

No statistical test was performed here.

(R2Q12) In the Supplement, it sometimes gets confusing to associate captions with figures, due to the caption appearing on a page with the wrong figure. E.g., p. 32 has the caption for S16, but the figure shown on that page is S17.

We thank the reviewer for pointing out this issue. We have adjusted the supplementary figures and caption positions to ensure that captions are not on the same page with other figures.

(R2Q13) L578, L638: In-line reference formatting is not entirely consistent – some references with 3 or more authors list the first two authors (rather than just the first) prior to “et al”.

Our reference management software adds two authors to distinguish papers published in the same year with the same first author. We note that Nature Communications uses numeric in-text reference style, so this issue will be fixed if (or when) the study is published.

(R2Q14) Some items related to some of the Methods additions: The run durations listed for each task are not consistent with the actual protocol.

The reviewer raised a very good point. We note that there is a discrepancy between Hagler et al. (2019) and Casey et al. (2018).

In particular, Hagler2019 (Image acquisition parameters in Appendix) stated that “For fMRI, acquisition time varies by scan type: rsfMRI (5’0”), MID (5’22”), nBack (4’50”), SST (5’50”).”

On the other hand, Casey2018 (Section 3.3.3.1 Monetary Incentive Delay Task) stated that “Each run consists of 50 contiguous trials (10 per trial type) presented in pseudorandom order and lasts 5:42.”

In our study, we calculated the actual task duration based on the minimally processed data and reported the results of our calculations, which turned out to be consistent with Hagler et al. (2019).

Pg 28 (Methods)

Each MID run was 322.4 secs long. Each N-Back run was 289.6 secs long. Each SST run was 349.6 secs long.

(R2Q15) The fact that task activity was not regressed should be mentioned under the “Functional connectivity” section of the Methods as well. Also, whether or not a Fisher’s transform was applied prior to averaging the FC matrices from different runs.

We thank the reviewer for these suggestions. The manuscript has been updated as follows.

Pg 29 (Methods)

We note that correlation values were converted to z-scores using Fisher’s r-to-z transformation prior to averaging and converted back to correlation values after averaging... To match processing across resting and task states, task activations were not regressed from the task-state data.

(R2Q16) L782: It would be clearer if the amount of available data was framed in terms of data “remaining after censoring”, rather than “length of uncensored data”.

We thank the reviewer for this suggestion. We have updated the manuscript as follows.

Pg 29 (Methods)

For these 1858 children, the length of fMRI data remaining after censoring was 897 ± 232 secs (mean \pm std) for resting-state, 531 ± 107 secs for MID, 579 ± 126 secs for SST, and 482 ± 86 secs for N-back.

(R2Q17) L824-5: It is probably worth mentioning that after averaging, you (presumably) inverted the Fisher transform (z-to-r) so that the reported values in all the figures are back in standard correlation “units”.

The reviewer is correct. We have now made this clear in the manuscript.

Pg 30 (Methods)

When averaging prediction accuracies (correlations) across behavioral measures, the correlations underwent Fisher-r-to-z transformation before averaging and converted back to correlation values after averaging.

*(R2Q18) L842 (Model generalization across behaviors): The methodological description here seems incomplete. In particular, the text jumps straight to “we averaged the predictions”, without first detailing exactly what was done. For example, were the *predictive-feature* matrices of the other ‘reference’ behaviors first averaged, to generate a model, from which the target behavior could be predicted? Or, were the features of each individual ‘reference’ behavior used to separately predict the ‘target’ behavior, and then those predicted values averaged across all the other ‘reference’ behaviors?*

We apologize for being unclear. We have added a specific example of how the prediction was done. Hopefully it should be clear now.

Pg XXX (Methods)

For example, in the case of generalization within the same behavioral domain, a participant’s fluid cognition was predicted as follows. The resting-FC and task-FC of the participant were fed into the 15 multi-kernel predictive models (of the other 15 cognitive measures), yielding 15 prediction values. These 15 prediction values were averaged yielding a final prediction of fluid cognition for the participant.

(R2Q19) L850 (Feature generalization across behaviors): If I’m reading this correctly, the FC information was averaged across the selected edges, to yield a single “average FC” per individual. In which case, how was the prediction computed? Was it still run through the kernel regression? If so, how was the similarity term defined?

We apologize for being unclear. We did not run kernel regression again. Instead, for simplicity, this single “average FC” was directly used as the prediction. We note this is not an issue because scale differences between FC and the behavioral measure would not affect the prediction performance metric (Pearson’s correlation). We could have done something “fancier” like fitting a univariate general linear model based on this single “average FC” value in the training data and then performing prediction in the test data, but this would not affect the Pearson’s correlation metric.

We have clarified this in the manuscript.

Pg 31 (Methods)

For each participant, an aggregate FC value was computed as the average FC of selected edges with positive predictive-feature values minus the average FC of selected edges with negative predictive feature values. The aggregate FC value was used as the prediction for the participant. The prediction was compared with the actual behavioral value to compute cross-behavior prediction performance. We note that scale differences between the aggregate FC value and behavioral measures were not an issue since Pearson’s correlation is invariant to scale.

(R2Q20) L897 (Multiple comparison correction): L899 refers to “statistical test for results in Figure 1-3”, but Figure 1 doesn’t involve any statistical tests. It’s unclear what the “105 tests” refers to. Similarly, it’s unclear what constitutes the “4149 tests” for results from Figure 4 onward. In general, while the use of FDR is a positive, it’s hard to assess what specific comparison FDR was applied over.

We apologize for being unclear. The reviewer is correct that Figure 1 did not involve any statistical test. That was a typo. We have modified the manuscript to make it clearer.

Pg 32 (Methods)

We performed FDR correction ($q < 0.05$) on all statistical tests that did not involve predictive feature matrices to determine which behaviors were significantly predicted and would therefore be utilized in subsequent analyses. There was a total of 105 tests and the p value threshold was 0.041. A separate FDR correction ($q < 0.05$) was performed on all tests involving predictive network features. There was a total of 4149 tests and the p value threshold was 0.029. FDR correction was also performed separately for Table S4 and Table S9. Table S10 provides more details about the statistical tests performed. When reporting p values, the nominal p values were reported.

First set of statistical corrections (total of 105 statistical tests)	
Figure 2	Compare all prediction models against chance-level prediction (5 models * 3 behavioral domains = 15 tests) Compare prediction of resting-FC against each task-FC (3 tasks * 3 behavioral domains = 9 tests) Compare multi-kernel FC against the best single kernel (3 behavioral domains = 3 tests)
Figure 3	Compare predictions against chance-level prediction (36 tests)
Figure S2	Compare predictions against chance-level prediction (36 tests)
Figure S18	Compare multi-kernel FC against mean-FC (3 behavioral domains = 3 tests) Compare mean-FC model against the best single-kernel model (3 behavioral domains = 3 tests)
Second set of statistical corrections (total of 4149 statistical tests)	
Figure 5	Compare within- and between- domain network overlap against chance-level overlap (6 tests) Compare within-domain overlap and between-domain overlap (6 tests) Compare within brain state network overlap against chance-level overlap (3 tests)
Figure 7a	Compare model-transfer accuracies against chance-level prediction (9 tests) Compare within-domain model-transfer against between-domain model transfer (6 tests)
Figure 7c	Compare feature-transfer accuracies against chance-level prediction (9 tests) Compare within-domain feature-transfer against between-domain feature transfer (6 tests)
Figure 8	Compare average PNF value of each network block for each brain state against chance-level PNF values (171 unique blocks * 4 brain states * 3 behavioral domain = 2052 tests)
Figure S14	Compare average PNF value of each network block for each brain state against chance-level PNF values (171 unique blocks * 4 brain states * 3 behavioral domain = 2052 tests)
Third set of statistical corrections (total of 36 statistical tests)	
Table S4	Compare behavioral distributions of included and excluded subjects (36 tests)
Fourth set of statistical corrections (total of 180 statistical tests)	
Table S9	Compare distributions of age, sex, race, income, and behavior between included and excluded subjects (5 variables matched * 36 behaviors = 180 tests)

Reviewer #3:

(R3Q1) The authors have submitted a very impressive revision that addressed all of my comments thoroughly. I am content with the manuscript in its current form and look forward to seeing this addition to the literature.

We thank the reviewer for the positive comments.

References

Casey BJ et al. 2018. The Adolescent Brain Cognitive Development (ABCD) study: Imaging acquisition across 21 sites. *Developmental cognitive neuroscience*. 32, 43–54.

Chaarani B et al. 2021. Baseline brain function in the preadolescents of the ABCD Study. *Nature Neuroscience*. Jun 7:1-1.

Hagler, D.J. et al. 2019. Image processing and analysis methods for the Adolescent Brain Cognitive Development Study. *Neuroimage*. 2019 Nov 15;202:116091.

Power, J.D., Barnes, K.A., Snyder, A.Z., Schlaggar, B.L., and Petersen, S.E., 2012. Spurious but systematic correlations in functional connectivity MRI networks arise from subject motion. *NeuroImage*, 59 (3), 2142–2154.

Satterthwaite TD, Wolf DH, Loughead J, Ruparel K, Elliott MA, Hakonarson H, Gur RC, Gur RE. 2012. Impact of in-scanner head motion on multiple measures of functional connectivity: relevance for studies of neurodevelopment in youth. *Neuroimage*. 60(1):623-32.

REVIEWERS' COMMENTS

Reviewer #2 (Remarks to the Author):

The authors have done a commendable job of wrestling with the issue of data quality vs. sample representativeness. I am now satisfied that the results seem robust. This will be an impactful and valuable addition to the literature.

Some minor suggestions and items for clarification related to the specific additions in this revision follow:

Regarding the 4th row additions to Figure 7 (prediction performance using the original multi-kernel FC models), the caption refers back to Figure 2 as the source of that data, but Figure 2 clearly shows that the mean prediction performance of Personality and Mental Health were both approximately 0.10. So, where does the value of 0.132 for Mental Health in Figure 7 come from?

L510 should mention that the main sample also had higher household income than the excluded sample (in addition to higher proportion of female and white participants).

The results in Tables S9[A,B,C] should include an uncorrected p-value of the comparison between the 'Included' and 'Excluded' sample for each "Behavior being matched" and each matching variable. Simply stating that there was no significant difference after FDR (across 180 comparisons) is too stringent in this case. For example, if more than 10% of the samples differed at $p < 0.1$ (uncorrected) that's meaningful information to know.

Figure 8 caption: It would be helpful to readers if the last sentence of the caption also mentioned the similarity to the control analyses, and pointed to Figures S21, S22, and S23.

L979: It seems to me that the description of the stratified subsampling could be clearer. For example, why use "3 x 3 x 3 x 3 x 3" as an example of the number of bins, rather than just being precise about the number of bins used for each characteristic? For the categorical characteristics (sex, racial composition, and presumably income), didn't the number of bins need to be fixed? As for the continuous variables (age and behavior), how about reporting the min, max and median of the number of bins used? This takes almost no additional space, and has the benefit of being precise.

p. 44 in the Supplement is blank.

The Table in the author's response contains useful information regarding the N available if one requires a conjunction of REST, MID, N-Back, and SST, as well as no missing behavior. I would suggest the authors considering including that as another supplemental table, and refer to it at L711.

(R2Q1) The authors have done a commendable job of wrestling with the issue of data quality vs. sample representativeness. I am now satisfied that the results seem robust. This will be an impactful and valuable addition to the literature.

We thank the reviewer for the positive comments

(R2Q2) Regarding the 4th row additions to Figure 7 (prediction performance using the original multi-kernel FC models), the caption refers back to Figure 2 as the source of that data, but Figure 2 clearly shows that the mean prediction performance of Personality and Mental Health were both approximately 0.10. So, where does the value of 0.132 for Mental Health in Figure 7 come from?

The accuracy value of 0.132 in Figure 7 is the average accuracies of all significantly predicted mental health, so it is higher the number of 0.1 in Figure 2, which is the average accuracy of all mental health behaviors. We have modified the caption of Figure 7 to make it clearer.

pg 18 (Figure 7 caption)

As a reference, average prediction performance of behavioral measures (predicted better than chance) from the original multi-kernel FC models (Figure 3) is shown in the fourth row.

(R2Q3) L510 should mention that the main sample also had higher household income than the excluded sample (in addition to higher proportion of female and white participants).

We have added this in the text.

pg 20 (Results)

The resulting sample had a higher proportion of female and white participants with higher family income than excluded participants (Table S3).

(R2Q4) The results in Tables S9[A,B,C] should include an uncorrected p-value of the comparison between the 'Included' and 'Excluded' sample for each "Behavior being matched" and each matching variable. Simply stating that there was no significant difference after FDR (across 180 comparisons) is too stringent in this case. For example, if more than 10% of the samples differed at $p < 0.1$ (uncorrected) that's meaningful information to know.

We have added the uncorrected p-values in Table S9D.

Table S9D. Continuation of Table S9. Uncorrected p values of differences in characteristics between included and excluded participants after matching. There is no significant difference after multiple comparisons correction with FDR $q < 0.05$.

	Age	Sex	Income	Race	Behavior
Vocabulary	0.10	0.18	0.92	0.03	0.01
Attention	0.38	0.11	0.53	0.06	0.07

Working memory	0.15	0.10	0.91	0.15	0.06
Executive function	0.10	0.29	0.50	0.02	0.05
Processing speed	0.07	0.19	0.43	0.01	0.01
Episodic memory	0.06	0.06	0.64	0.01	0.02
Reading	0.04	0.11	0.75	0.004	0.001
Fluid cognition	0.24	0.12	0.61	0.01	0.03
Crystallized cognition	0.28	0.41	0.99	0.08	0.01
Overall cognition	0.16	0.08	0.67	0.01	0.008
Short delay recall	0.12	0.04	0.77	0.02	0.24
Long delay recall	0.06	0.17	0.93	0.07	0.59
Fluid intelligence	0.03	0.01	0.18	0.004	0.01
Visuospatial accuracy	0.11	0.13	0.86	0.06	0.08
Visuospatial reaction time	0.08	0.16	0.96	0.09	0.13
Visuospatial efficiency	0.16	0.12	0.69	0.09	0.05
Negative urgency	0.02	0.16	0.91	0.01	0.78
Lack of planning	0.07	0.24	0.83	0.05	0.14
Sensation seeking	0.06	0.24	0.83	0.07	0.65
Positive urgency	0.04	0.08	0.75	0.01	0.16
Lack perseverance	0.08	0.09	0.58	0.02	0.94
Behavioral inhibition	0.04	0.08	0.67	0.02	0.73
Reward responsiveness	0.09	0.13	0.96	0.05	0.60
Drive	0.03	0.04	0.82	0.01	0.05
Fun seeking	0.05	0.18	0.91	0.05	0.26
Anxious depressed	0.08	0.12	0.84	0.09	0.29
Withdrawn depressed	0.18	0.14	0.90	0.06	0.13
Somatic complaints	0.05	0.10	0.91	0.08	0.44
Social problems	0.11	0.24	0.88	0.07	0.62
Thought problems	0.09	0.11	0.88	0.06	0.76
Attention problems	0.16	0.17	0.96	0.10	0.59

Rule-breaking behavior	0.06	0.25	0.75	0.06	0.09
Aggressive behavior	0.09	0.16	0.83	0.10	0.51
Total psychosis symptoms	0.08	0.06	0.59	0.06	0.06
Psychosis severity	0.07	0.13	0.81	0.06	0.11
Mania	0.05	0.12	0.83	0.07	0.49

(R2Q5) Figure 8 caption: It would be helpful to readers if the last sentence of the caption also mentioned the similarity to the control analyses, and pointed to Figures S21, S22, and S23.

We thank the reviewer for this great suggestion. We have updated as follows:

pg 23 (Figure 8 captions)

Conclusions were highly similar using data-driven behavioral clusters (Figures S15 & S16), as well as other control analyses (Figures S21 to S23).

(R2Q6) L979: It seems to me that the description of the stratified subsampling could be clearer. For example, why use “3 x 3 x 3 x 3 x 3” as an example of the number of bins, rather than just being precise about the number of bins used for each characteristic? For the categorical characteristics (sex, racial composition, and presumably income), didn’t the number of bins need to be fixed? As for the continuous variables (age and behavior), how about reporting the min, max and median of the number of bins used? This takes almost no additional space, and has the benefit of being precise.

We have modified this part to be more specific.

pg 33 (Methods)

Briefly, we divided each characteristic (age, sex, household income, racial composition, and behavioral distribution) into a number of bins. There were 2 bins for sexes, 5 bins for racial groups, and 4 bins for income levels. In the case of age and behavioral measures, the number of bins varied depending on the behavioral measures. On average, there were 2.14 bins for age (min = 2, max = 3, median = 2) and 3.64 bins for behavioral measures (min = 3, max = 7, median = 3). For example, if there were 2 bins for age and 3 bins for a particular behavioral measure, then there were 2 (sex) x 5 (race) x 4 (income) x 2 (age) x 3 (behavior) = 240 bins.

(R2Q7) p. 44 in the Supplement is blank.

We have deleted the blank page.

(R2Q8) The Table in the author’s response contains useful information regarding the N available if one requires a conjunction of REST, MID, N-Back, and SST, as well as

no missing behavior. I would suggest the authors considering including that as another supplemental table, and refer to it at L711.

We have included this result as a supplementary figure.

pg 27 (Discussion)

This resulted in a much smaller sample size than if we analyzed each brain state independently (Chaarani et al., 2021; Figure S24).

	REST	MID	N-Back	SST	Conjunction
Pass T1 recon-all qc	11075				
Not Phillips scanner	9640				
>= 2 Runs passed MRI QC	9217	8281	8141	8147	7495
Mean FD < 0.9 for both runs	8769	7271	6644	6908	5615
Having at least one minimally processed run released by ABCD	8451	6896	6287	6503	5014
No Missing behavior					4664
No Sibling					4187

Figure S24. Number of participants if we used quality control (QC) criteria from Chaarani and colleagues (Chaarani et al., 2021) excluding task activation QC criteria (e.g., beta weights outlier detection). Because we excluded the task activation QC criteria in this table, the resulting sample size for each modality (e.g., N = 6503 for SST task) was larger than Chaarani's study. Yet, after conjunction across resting and task states, we were left with 4187 participants, which was only 11% more than our control analysis (N = 3744). Furthermore, we note that the QC criteria in this Figure excluded typical functional connectivity QC used in the literature.